# DexNDM: Closing the Reality Gap for Dexterous In-Hand Rotation via Joint-wise Neural Dynamics Model

**Xueyi Liu**[1,3], **He Wang**[2,4], **Li Yi**[1,3]
[1]Tsinghua University  [2]Peking University  [3]Shanghai Qi Zhi Institute  [4]Galbot
Project website: meowuu7.github.io/DexNDM

**(A) Challenging Geometries**

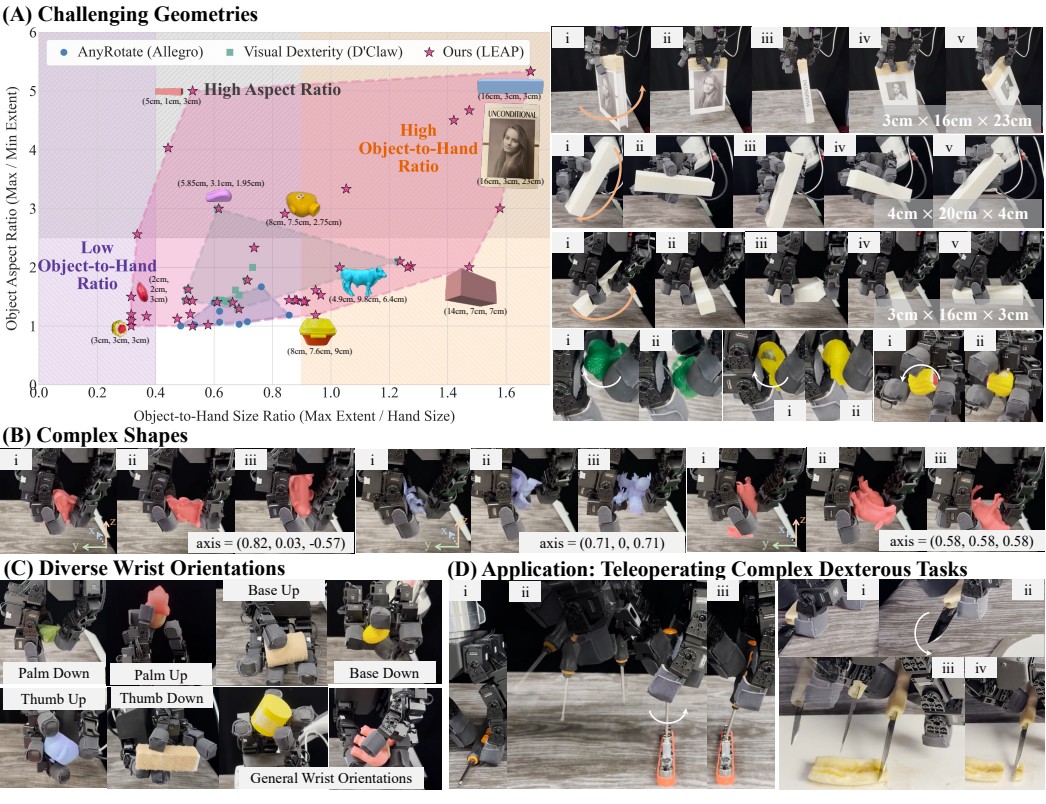

Figure 1: We introduce DexNDM, a sim-to-real approach that enables unprecedented in-hand rotation in the real world. We master a wide object distribution, including **(A)** challenging geometries and **(B)** complex shapes, across **(C)** rich wrist orientations. **(D)** A teleoperation application. Videos in website.

## Abstract

Achieving generalized in-hand object rotation remains a significant challenge in robotics, largely due to the difficulty of transferring policies from simulation to the real world. The complex, contact-rich dynamics of dexterous manipulation create a "reality gap" that has limited prior work to constrained scenarios involving simple geometries, limited object sizes and aspect ratios, constrained wrist poses, or customized hands. We address this sim-to-real challenge with a novel framework that enables a single policy, trained in simulation, to generalize to a wide variety of objects and conditions in the real world. The core of our method is a joint-wise dynamics model that learns to bridge the reality gap by effectively fitting limited amount of real-world collected data and then adapting the sim policy's actions accordingly. The model is highly data-efficient and generalizable across different whole-hand interaction distributions by factorizing dynamics across joints, compressing system-wide influences into low-dimensional variables, and learning each joint's evolution from its own dynamic profile, implicitly capturing these net effects. We pair this with a fully autonomous data collection strategy that gathers diverse, real-world interaction data with minimal human intervention. Our com-

plete pipeline demonstrates unprecedented generality: a single policy successfully rotates challenging objects with complex shapes (*e.g.*, animals), high aspect ratios (up to 5.33), and small sizes, all while handling diverse wrist orientations and rotation axes. Comprehensive real-world evaluations and a teleoperation application for complex tasks validate the effectiveness and robustness of our approach.

# 1 INTRODUCTION

Advancing dexterous manipulation is essential to achieving highly capable embodied intelligence. A fundamental yet challenging skill in this domain is in-hand object rotation. The long-standing goal, which we also pursue in this work, is to develop a general-purpose policy that can rotate a broad distribution of objects across diverse wrist orientations and rotation axes in the real world.

Despite recent progress, the community has yet to achieve this level of generality. Existing methods (Chen et al., 2022; Yang et al., 2024; Qi et al., 2023; Wang et al., 2024; Zhao et al., 2025; Yuan et al., 2023) are often constrained to specific scenarios: some assume a consistently up-facing hand, others handle only a limited set of simple, regular-sized objects, and many rely on expensive, customized hardware with sophisticated tactile sensing. While some approaches (Yang et al., 2024) show generality in one dimension, such as rotation axes, they are limited in others, like object complexity. To our knowledge, no prior work demonstrates robust, in-the-air rotation for a wide spectrum of objects—including complex shapes, high aspect ratios, and varied sizes—under diverse wrist orientations and rotation axes.

The primary barrier to this goal is the formidable "sim-to-real gap", due to the difficulty in modeling the complex interaction dynamics marked by rich, rapidly varying, and load-dependent contacts. This undermines both model-based (Pang & Tedrake, 2021; Pang et al., 2023; Suh et al., 2025) and model-free (Qi et al., 2023; Chen et al., 2022; Yang et al., 2024) approaches. A promising idea for sim-to-real transfer is learning a neural dynamics model from real-world data (He et al., 2025; bin Shi et al., 2024). This approach has proven effective in locomotion, where relatively easier failure recovery and readily observable states permit efficient collection of distributionally relevant task data. This success, however, does not easily translate to general-purpose manipulation, where the requirements for data volume and distributional relevance create an inescapable conflict. The need for generality demands massive data to cover diverse objects. Yet, ensuring this data is distributionally relevant is sometimes impossible and operationally far more complex: suboptimal deployable policy cannot manipulate hard objects (*e.g.*, long); catastrophic failures (*i.e.*, dropping the object) necessitates frequent human intervention for resets; severe hand-induced occlusions complicate accurately tracking states of diverse objects. This conflict creates a critical bottleneck for the field.

To overcome these challenges, we introduce a framework that breaks this inescapable conflict by fundamentally rethinking both the model and the data. Our central insight is to factorize the learning problem through a more generalizable dynamics model, which in turn enables a more scalable data collection strategy. First, instead of modeling the high-dimensional hand-object system as a whole (bin Shi et al., 2024), we learn a joint-wise neural dynamics model. This model factorizes the system and predicts the evolution of each joint using only its own proprioceptive history, generalizing the idea of RMA (Kumar et al., 2021). This design directly confronts the challenges: it is inherently immune to object state estimation difficulty, and by distilling system-wide influences—self-actuation, inter-joint couplings and object loads—into low-dimensional and task-sufficient net effects with reduced nuisance variability, the model becomes highly sample-efficient and generalizable without sacrificing expressivity as evidenced by experiments. This enhanced generalizability is the key that unlocks our second innovation: a fully autonomous data collection strategy. By applying randomized loads to the hand in a task-agnostic manner, we gather data while eliminating catastrophic failures and the need for human resets. This allows us to learn a dynamics model generalizing well to our task of interest from cheap and scalable data, which we then use to train a residual policy that adapts a simulation-trained base policy to the real world, achieving broad generality. We attain the base policy via a specialist-to-generalist pipeline: train category-specific experts on data spanning aspect ratios and geometric complexities, then distill them into a unified policy.

We validate our method in both the simulation and the real world. In simulation, our base policy generalizes to novel, complex shapes, outperforming strong baselines by 37%–81%. In real world, our sim-to-real method significantly and consistently improves rotation performance, enabling versatile rotation across diverse wrist orientations and rotation axes on a broad object distribution—including

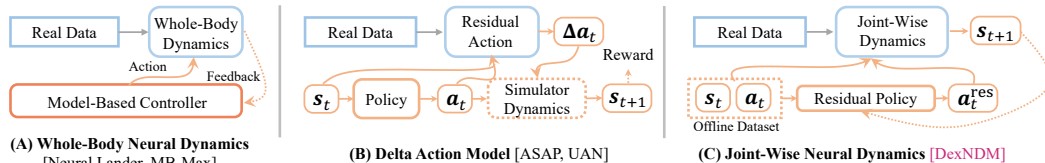

**Figure 2: Learning from Real-World Data for Control.** (A) Learn a whole-body dynamics model from real-world data for policy tuning or model-based control. (B) Learn a residual action model to finetune a base policy. (C) Learn joint-wise dynamics and a residual policy to adapt the base policy.

complex geometries (*e.g.,* animal models), aspect ratios up to 5.33, and object-to-hand ratios of 0.31–1.68 (Fig. 1; videos on our website). Notably, in a challenging downward-facing hand configuration, we are, to our knowledge, the first to rotate long objects (10–16 cm) around their long axis for about one full circle in the air. Compared to Visual Dexterity (Chen et al., 2022) on a large, customized D'Claw, our smaller LEAP hand matches or surpasses performance and succeeds on shapes it struggles with (*e.g.*, elephant, bunny, teapot). We also generalize to a much broader, more challenging object distribution than the prior multi-wrist SOTA (Yang et al., 2024). Moreover, we showcase an application enabled by our general rotation policy: building a teleoperation system to perform complex dexterous tasks, such as tool-using (*e.g.,* screwdriver, knife) and assembly (Heo et al., 2023). A systematic ablation study validates the crucial role of our key design choices in both the dynamics model and the data collection strategy. Our main contributions are four-fold:

- A novel sim-to-real framework for dexterous in-hand rotation, built on a joint-wise neural dynamics model and autonomous data collection to tackle the core challenges of learning complex interaction dynamics and acquiring real-world interaction data.
- An in-hand object rotation policy that achieves unprecedented generality in rotating challenging objects (high-aspect-ratio, complex shapes, small sizes) under difficult wrist orientations.
- An in-depth analysis of the rationale, advantages, and scope of effectiveness of the joint-wise neural dynamics model from both theoretical and empirical perspectives.
- A demonstration of a practical application in teleoperation for complex dexterous tasks.

## 2 RELATED WORK

Our work is broadly related to two research topics: in-hand object rotation and sim-to-real strategies. **In-hand rotation** is an important yet challenging robitc task. Despite advances, prior methods still (i) assume an up-facing hand (Qi et al., 2022; Wang et al., 2024; Yuan et al., 2023; Zhao et al., 2025), (ii) handle only normal-sized objects with limited geometric diversity (Qi et al., 2023; Röstel et al., 2025; Pitz et al., 2024a;b; Yang et al., 2024), or (iii) rely on expensive hardware and sophisticated tactile sensing (Yang et al., 2024; Wang et al., 2024; Qi et al., 2023). AnyRotate (Yang et al., 2024) achieves axis and wrist generality, but only on normal-sized regular objects in the real world. Visual Dexterity (Chen et al., 2022) rotates complex shapes in the air, yet performance on small or high-aspect-ratio objects is unverified. DexterityGen (Yin et al., 2025) showcases a meaningful application of in-hand reorientation policies in teleoperation, but its rotation capability does not exceed the upper bound of prior work. We aim to achieve generality in rotating challenging (*e.g.,* long, small) and complex objects across diverse wrist orientations and rotation axes. A central obstacle to realizing this is the **sim-to-real gap**: mismatched parameters, model discrepancies, and unmodeled effects derail transfer of simulation-trained policies. Existing approaches include: (1) Domain Randomization (DR), which broadens training distributions (Loquercio et al., 2019; Peng et al., 2017; Tan et al., 2018; Yu et al., 2019; Mozifian et al., 2019; Siekmann et al., 2020); (2) System Identification (SysID), which fits simulator parameters from real data (An et al., 1985; Mayeda et al., 1988; Lee et al., 2023; Sobanbabu et al., 2025); (3) online adaptive policies (Kumar et al., 2021; Qi et al., 2022); and (4) neural modeling of real dynamics to guide transfer (He et al., 2025; Fey et al., 2025; Hwangbo et al., 2019). DR relies on heuristic ranges; SysID is bounded by its parameterization; and online adaptation typically depends on dynamics coverage in training. Learning real dynamics offers the highest ceiling: A classical line in neural control learns residual or full models for the whole system for model-based control (Fig. 2 (A), *e.g.,* Neural Lander (Shi et al., 2018), MB-Max (bin Shi et al., 2024)). As the task complexity increases, learning globally accurate, physically plausible dynamics that is super robust to support policy tuning or controller development is difficult (Shi, 2025). Therefore, another trend of methods proposed in sim-to-real RL (*e.g.,* UAN (Fey et al., 2025) and ASAP (He et al., 2025)) learn sim-real delta actions and fine-tune policies based on that to bridge the dynamics gap (Fig. 2 (B)). Success hinges on collecting enough real-world data that is distributionally relevant to the task or can offer a comprehensive coverage—a minor issue in locomotion and static-contact tasks, but a major bottleneck in dexterous manipulation. We address

this with a generalizable joint-wise neural dynamics model that relaxes the training data distribution requirement, followed by a residual policy to bridge the reality gap (Fig. 2 (C)).

# 3 METHODOLOGY

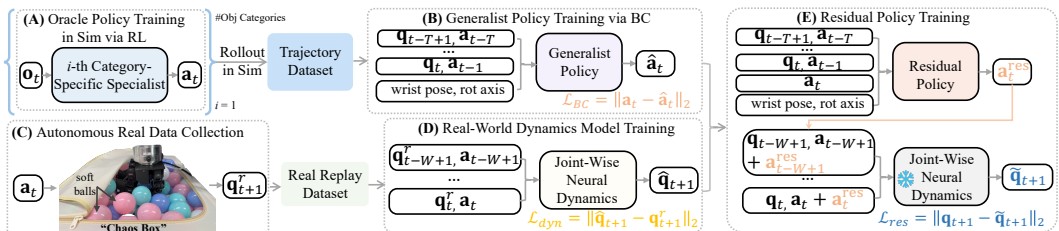

Figure 3: **Method Overview. (A)** RL-train object category-specific rotation specialists. **(B)** Distill them into a single generalist via BC. **(C-E)** Neural sim-to-real: autonomously collect real-world transitions with stochastic contact conditions **(C)**, learn a joint-wise neural dynamics model **(D)**, and train a residual to bridge the reality gap **(E)**. Deploy the base generalist **(B)** augmented with the residual **(E)**.

Our goal is a generalist policy that can rotate a wide variety of objects under various conditions in the real world. We adopt a model-free RL approach. Key challenges are the pronounced sim-to-real dynamics gap in contact-rich dexterous manipulation and the need for broad object generalization. To achieve broad generalization, we adopt a specialist-to-generalist approach: we first train category-specific oracle policies across curated object categories (Sec. 3.1), and then distill them into a single generalist policy (Sec. 3.2). We address the sim-to-real challenge via a neural sim-to-real strategy centered on an expressive, data-efficient, and generalizable joint-wise dynamics model (Sec. 3.3). This model can generalize to the target rotation task by training on imperfectly aligned data, enabling a fully autonomous data-collection system in which real-world interaction data is gathered by applying stochastic contacts to the robotic hand. With the neural dynamics model learned from this autonomously collected data, we train a residual policy that compensates for the base generalist policy's actions to close the sim-to-real dynamics gap. Workflow illustrated in Figure 3.

## 3.1 MULTI-WRIST-ORIENTATION IN-HAND OBJECT ROTATION ACROSS MULTI-AXIS

We formulate in-hand rotation as a finite-horizon Partially Observable Markov Decision Process (POMDP), $M = (\mathcal{S}, \mathcal{A}, \mathcal{O}, \mathcal{P}, \mathcal{R})$, with state, action, and observation spaces $(\mathcal{S}, \mathcal{A}, \mathcal{O})$, transition dynamics $\mathcal{P}$, and reward $\mathcal{R}$. We train a neural policy $\pi : \mathcal{O} \rightarrow \mathcal{A}$ with RL to maximize expected cumulative return over horizon $N$: $\pi^* = \arg\max_\pi \mathbb{E}_{\tau \sim p_\pi(\tau)}[\sum_{t=1}^N r(\mathbf{s}_t, \mathbf{a}_t)]$.

**Observations and Actions.** At timestep $t$, the policy receives $\mathbf{o}_t$: a short history of proprioception, fingertip and object states, per-joint/per-finger force measurements, binary contact signals, wrist orientation, and the target rotation axis (Sec. A.1). The policy outputs a distribution over relative target position. We sample $\Delta\mathbf{a}_t \sim \pi(\mathbf{o}_t)$ and update the joint target $\mathbf{a}_t = \mathbf{a}_{t-1} + \alpha\,\Delta\mathbf{a}_t$ with $\alpha = 1/24$. $\mathbf{a}_t$ is converted to torques via a PD controller and executed on the robot.

**Reward Function.** The reward consists of three weighted components $r = \alpha_{\text{rot}}r_{\text{rot}} + \alpha_{\text{goal}}r_{\text{goal}} + \alpha_{\text{penalty}}r_{\text{penalty}}$, with $r_{\text{rot}}$ and $r_{\text{penalty}}$ following RotateIt (Qi et al., 2023). The rotation term $r_{\text{rot}}$ encourages rotation about the target axis. The penalty $r_{\text{penalty}}$ discourages off-axis angular velocity, deviation from a canonical hand pose, object linear velocity, and joint work/torque. Since these rewards alone struggle on hard cases (*e.g.*, rotating long objects), we add an intermediate goal-pose reward, $r_{\text{goal}}$, that guides the object to a waypoint on the target rotation axis. Details in Sec. A.1

## 3.2 GENERALIST POLICY TRAINING VIA BEHAVIOUR CLONING

Having obtained the oracle policy with rich privileged observations for each object category, we use Behavior Cloning (BC) to train the unified, real-world deployable, multi-geometry generalist policy. Although DAgger-style distillation has been effective in prior work, in our early experiments on deploying single-object-category, downward-facing hand rotation policies, we observe that they either fail to optimize in simulation or collapse in the real world. We attribute this to high task difficulty. We therefore switched to BC and discovered that the resulting proprioception-only specialist policy behaves more stably than the DAgger policy and can rotate simple cylinder objects. We continue to use BC to train the generalist, real-world deployable policy: roll out all oracle policies, aggregate only successful trajectories, and train a generalist via supervised learning. This approach works well on hardware. We hypothesize that its success stems from imitating only high-quality

oracle behavior. The observation $\mathbf{o}_t^{\text{gene}}$ of the generalist policy contains a history of proprioception $\{(\mathbf{q}_k, \mathbf{a}_{k-1})\}_{k=t-T+1}^{t}$, wrist orientation and rotation axis. We use $T = 10$ and implement the policy as a residual MLP (He et al., 2015).

### 3.3 CLOSING THE REALITY GAP VIA JOINT-WISE NEURAL DYNAMICS

While the generalist policy is already real-world deployable, a persistent sim-to-real gap—caused by mismatched physical dynamics and unmodeled effects—prevents it from mastering challenging object interactions. We bridge this gap with a novel neural sim-to-real strategy that effectively learns complex, real-world dynamics model.

The central challenge is to acquire useful and sufficient volume of real data so that the learned dynamics model can help sim-to-real transfer. For dexterous manipulation, prior data acquisition methods (Hwangbo et al., 2019; He et al., 2025; Fey et al., 2025; bin Shi et al., 2024) are often impractical. Rolling out a base policy (He et al., 2025; bin Shi et al., 2024) or executing wave actions (Fey et al., 2025) frequently fails on diverse and complex objects, requiring constant human intervention, while imperfect state estimators introduce heavy noise. This leads to real datasets that are small, biased, and insufficient in coverage and quality. We address these challenges by rethinking both model and data. We propose a joint-wise neural dynamics model that dramatically improves sample efficiency and generalizability while preserving expressivity by learning from a low-dimensional, information-contractive, task-sufficient representation of the system dynamics. This allows for an autonomous data collection strategy that gathers diverse, large-scale real-world data by applying stochastic contact, eliminating the need for task-specific rollouts and human resets.

**Joint-Wise Neural Dynamics.** To model the system's dynamics without relying on noisy and limited object-state estimation, one way is to learn a "whole-hand" neural model. This model predicts the hand's next state from its length-$W$ state–action history, $\mathbf{q}^{t+1} = f_\theta(H_t)$ with $H_t = \{\mathbf{q}_j, \mathbf{a}_j\}_{j=t-W+1}^{t}$, thereby implicitly capturing the whole system dynamics, including external forces from the object (Qi et al., 2022). However, this approach remains data-hungry, inheriting the other data acquisition challenges described above.

Our solution is to factorize the problem. We introduce joint-wise neural dynamics where the dynamics of each joint $i$ are modeled as $\mathbf{H}_t^{\text{eff}} \ddot{\mathbf{q}}_t^i + \mathbf{G}_t^{\text{eff}} = \tau_t^i$, where $\mathbf{H}_t^{\text{eff}}, \mathbf{G}_t^{\text{eff}} \in \mathbb{R}$ are low-dimensional effective terms that distill high-dimensional, system-wide influences such as inter-joint coupling, actuation, and object-induced effects. The neural model then predicts the next state of each joint $i$ from its own $W$-step state–action history: $\mathbf{q}_{t+1}^i = f_{\psi_i}(h_t^i)$ with $h_t^i = \{\mathbf{q}_j^i, \mathbf{a}_j^i\}_{j=t-W+1}^{t}$. This factorization is effective as it acts as an information bottleneck, forcing the model to discard spurious correlations and learn only the essential dynamics of each joint. This projected history is sufficiently informative with enough information to accurately predict the joint's next state (Sec. 4.2, A.3). At the same time, it is also robustly simple as it is too low-dimensional to permit the reconstruction of the original high-dimensional system-wide influences, thus avoiding the need to model irrelevant complexity (Sec. A.4). The direct consequence is a model that is highly sample-efficient and generalizes broadly across interactions, yet retains expressivity (Sec. 4.2). We now provide a theoretical analysis to formalize why this simplification leads to better generalization.

**Theoretical Rationale: Generalization via Information Contraction.** We write the whole-hand model as $f_\theta = \{f_\theta^i\}$ with $\mathbf{q}_{t+1}^i = f_\theta^i(H_t)$, and the joint-wise model as $\mathbf{q}_{t+1}^i = f_{\psi_i}^i(h_t^i)$. Let $\mathcal{P}$ be the target distribution for $(H_t, \mathbf{q}_{t+1}^i)$ (e.g., formed by task of our interest); consider a different distribution $\mathcal{Q}$ and the projection $g : (H_t, \mathbf{q}_{t+1}^i) \mapsto (h_t^i, \mathbf{q}_{t+1}^i)$, i.e., $g : \mathbb{R}^{2Wd} \times \mathbb{R} \to \mathbb{R}^{2W} \times \mathbb{R}$. We compare the prediction error of joint $i$ on the target distribution $\mathcal{P}$ achieved by these two types of model, i.e., $f_\theta^i$ and $f_{\psi_i}^i$, to support the generalization benefit:

**Claim 3.1** *Under assumptions typical of our setting, $\forall 1 \leq i \leq d$, the joint-wise model $f_{\psi_i}^i$ trained on $g(\mathcal{Q})$ generalizes to $g(\mathcal{P})$ better than the whole-hand model $f_\theta^i$ trained on $\mathcal{Q}$ generalizes to $\mathcal{P}$.*

We first show that, under mild assumptions typically satisfied in our setting, the projection $g$ contracts distribution shift: $\text{KL}(g(\mathcal{P}) \| g(\mathcal{Q})) < \text{KL}(\mathcal{P} \| \mathcal{Q})$ (Theorem 3.1, proof deferred to Sec. A.2).

**Theorem 3.1 (Data Processing Inequality for KL (strict form))** *Let $\mathcal{P}$ and $\mathcal{Q}$ be probability distributions on $\mathbb{R}^n \times \mathbb{R}$ with densities $P$ and $Q$ with respect to a common base measure. Let $g : X \in \mathbb{R}^n \times \mathbb{R} \to Y \in \mathbb{R}^m \times \mathbb{R}$ be measurable, $m \leq n$, and denote the pushforwards by $g(\mathcal{P})$ and $g(\mathcal{Q})$. Then $\text{KL}(\mathcal{P} \| \mathcal{Q}) \geq \text{KL}(g(\mathcal{P}) \| g(\mathcal{Q}))$. Moreover, the inequality is strict if $g$ is non-injective in a way that merges points where $\mathcal{P}$ and $\mathcal{Q}$ have a different relative structure. More*

*concretely, it indicates that if there* $\exists y_0 \in \mathbb{R}^m, P(Y = y_0) > 0, P(X|Y = y_0) \neq Q(X|Y = y_0),$ *then* $\mathrm{KL}(\mathcal{P} \parallel \mathcal{Q}) > \mathrm{KL}\big(g(\mathcal{P}) \parallel g(\mathcal{Q})\big).$

The contraction of divergence implies tighter generalization guarantees (Theorem 3.2, proof in A.2):

**Theorem 3.2 (Generalization Gap Contraction)** *Let* $(X, Y) \in \mathbb{R}^n \times \mathbb{R}$ *and* $g(X, Y) = (g_X(X), Y)$ *with* $g_X : \mathbb{R}^n \to \mathbb{R}^m$, $m < n$. *Let* $\mathcal{P}, \mathcal{Q}$ *be distributions on* $(X, Y)$ *satisfying covariate shift, i.e.,* $\mathcal{P}(Y \mid X) = \mathcal{Q}(Y \mid X)$. *Let* $L$ *be a loss bounded by* $B$, *and define* $R_{\mathcal{P}}(h) = \mathbb{E}_{(X,Y) \sim \mathcal{P}}[L(h(X), Y)]$. *If* $\mathrm{KL}\big(g(\mathcal{P})\|g(\mathcal{Q})\big) < \mathrm{KL}(\mathcal{P}\|\mathcal{Q})$, *then for function* $f_1 : X \to Y$ *and* $f_2 : g_X(X) \to Y$: $\sup|R_{\mathcal{P}}(f_2 \circ g_X) - R_{\mathcal{Q}}(f_2 \circ g_X)| < \sup|R_{\mathcal{P}}(f_1) - R_{\mathcal{Q}}(f_1)|.$

Assuming $f_2 \circ g_X$ is sufficiently expressive and a relatively large domain shift from $\mathcal{Q}$ to $\mathcal{P}$ (typical of our setting), $f_2 \circ g_X$ has lower prediction error than $f_1$ on target domain $\mathcal{P}$, establishing Claim 3.1. See Sec. A.2 for details. In practice, we pretrain the model on simulation data for initialization.

**Autonomous Data Collection.** Our model's ability to generalize from distributionally different data motivates our second innovation: a low-cost, autonomous data collection strategy. This approach, which we call the "Chaos Box" (Fig. 3(C)), embodies four principles: (i) policy-awareness (to roughly align the distribution), (ii) object-loaded interaction, (iii) broad coverage, and (iv) scalability. The implementation is simple: the robotic hand is placed in a container of soft balls. We then open-loop replay actions from the simulated base policy, which provides a coarse distributional prior (i). The hand's interaction with the balls imposes rich, stochastic contacts (ii-iii). With probability 0.5, we add Gaussian noise ($\sigma$=0.01) to each action to broaden coverage (iii). This entire process is fully autonomous, hardware-safe, and requires no human resets (iv). Fig. 4 supports our model and data designs: I/O histories of a joint cover the task-relevant distribution, whereas histories of the whole hand do not.

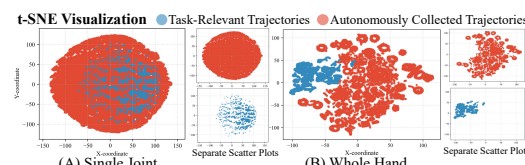

Figure 4: **State-Action History Distribution.**

**Bridging the Dynamics Gap via a Residual Policy.** Using the learned dynamics $f_\psi$, we train a residual policy $\pi^{\mathrm{res}}$ that compensates the base policy's actions to bridge the dynamics gap (Fig. 3(E)). By adding a corrective term to the base action, it wishes that the resulting state transition of the interaction system in the real world closely matches what would occur in simulation. Concretely, given the base policy's observation $\mathbf{o}_t^{\mathrm{gene}}$ and base action $\mathbf{a}_t$, $\pi^{\mathrm{res}}$ outputs a correction $\mathbf{a}_t^{\mathrm{res}}$, and to match the simulator's next state $\mathbf{q}_{t+1}$, we solve $\pi^{\mathrm{res}*} = \arg\min_{\pi^{\mathrm{res}}} \mathbb{E}_{\tau \sim p_{\pi^*}(\tau)} \sum_{t=1}^{N-1} \left\| \mathbf{q}_{t+1} - f_\psi\big(\{\mathbf{q}_j, \mathbf{a}_j + \pi^{\mathrm{res}}(\mathbf{o}_j^{\mathrm{gene}}, \mathbf{a}_j)\}_{j=t-W+1}^t\big) \right\|$. We solve it by training $\pi^{\mathrm{res}}$ in a supervised manner on the trajectory dataset used to train the base policy. At deployment, we execute $\mathbf{a}_t + \mathbf{a}_t^{\mathrm{res}}$. We opt not to use $f_\psi$ for policy training or finetuning the policy directly, which would require globally accurate, penetration-free, contact-rich dynamics (including the object) and super robustness to out-of-distribution exploration (Shi, 2025). See Sec. B.4 for a discussion on residual policy vs. direct finetuning.

## 4 EXPERIMENTS

We extensively evaluate our method in simulation and real world against strong baselines (Sec. 4.1). In simulation, our generalist policy generalizes to unseen geometries for multi-wrist poses, multi-axis rotation. On hardware, it achieves unprecedented in-air rotation with a LEAP hand (Shaw et al., 2023) under challenging wrist poses on difficult objects, including long (13.5-20cm), small (2-3cm) objects, and complex animal shapes (Sec. 4.2). We also show a teleoperation setup that pairs the policy with VR to perform complex dexterous tasks (Sec. 4.2), such as tool-using and assembly.

### 4.1 EXPERIMENTAL SETTINGS

**Training and Evaluation Protocols.** We create an object dataset spanning aspect ratios, sizes, and complexity with randomized physical properties for training. We split objects into five categories and train an oracle policy for each with PPO (Schulman et al., 2017) in Isaac Gym (Makoviychuk et al., 2021). We use objects from ContactDB (Brahmbhatt et al., 2019) as the test set in simulation to evaluate the generalization ability to shape variations. We evaluate rotation across randomized wrist orientations and four rotation-axis groups: $\pm x, \pm y, \pm z$, and a general axis set with 26 axes. We evaluate on three object sets in the real world (Fig. 5): (1) regular objects (including a high-aspect-ratio cuboid); (2) small objects; and (3) normal-sized irregular objects. Objects shown in purple

and all small objects are unseen. We evaluate on three principle axis sets and a cubic-diagonal set: (1,1,1), (1,0,1), (1,1,0), (0,1,1). Results are averaged over objects and reported as mean ± standard deviation across three independent evaluations. Details in Sec. C.

**Baselines.** We compare against in-hand rotation/reorientation baselines—AnyRotate (Yang et al., 2024) and Visual Dexterity (VD) (Chen et al., 2022)—and sim-to-real methods UAN (Fey et al., 2025) and ASAP (He et al., 2025). AnyRotate's code is

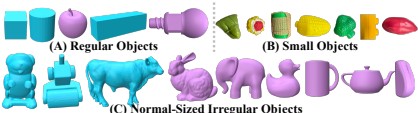

Figure 5: **Objects for Real Experiment.**

unavailable and relies on specialized tactile sensing, so we use our re-implementation in simulation; on hardware, we evaluate on their replicable objects and compare to their reported performance. A direct comparison to VD is impractical: adapting their D'Claw code to LEAP failed to behave well in simulation, so we compare to their qualitative results (link). UAN and ASAP, designed for arms/legged robots and not modeling objects, are adapted by training compensators on object-free transitions; making them object-aware is nontrivial (see Sec. D).

**Metrics.** We evaluate using RotateIt metrics (Qi et al., 2023), plus a goal-oriented success: *Time-to-Fall (TTF)*—duration until termination; in simulation, episodes are capped at 400 steps (20s) and TTF is normalized by 20s, while in the real world we report raw time; *Rotation Reward (RotR)*—episode sum of $\omega \cdot \mathbf{k}$ (simulation only); *Rotation Penalty (RotP)*—per-step average $\omega \times \mathbf{k}$ (simulation only); *Radians Rotated (Rot)*—total radians rotated in the real world; *Goal-Oriented Success (GO Succ.)* following Visual Dexterity: sample a goal pose; set the target axis to the relative rotation axis; count success if the orientation is within $0.1\pi$ of the goal (simulation only).

## 4.2 IN-HAND ROTATION RESULTS AND ANALYSIS

**Simulation Results.** Our policy generalizes to unseen objects and outperforms our re-implemented baseline (Table 1). Among all settings, rotating along the gravity direction ($\pm z$ axis) is the easiest task, similar to the observations made in prior works (Qi et al., 2023; Yang et al., 2024).

| Method | $\pm x$-axis | | | $\pm y$-axis | | | $\pm z$-axis | | | General Rotation Axes | | | GO. Succ. |
|---|---|---|---|---|---|---|---|---|---|---|---|---|---|
| | RotR ↑ | TTF ↑ | RotP ↓ | RotR ↑ | TTF ↑ | RotP ↓ | RotR ↑ | TTF ↑ | RotP ↓ | RotR ↑ | TTF ↑ | RotP ↓ | |
| **AnyRotate\* (re-implementation)** | $91.90_{\pm11.60}$ | $0.67_{\pm0.17}$ | $0.72_{\pm0.05}$ | $163.78_{\pm20.44}$ | $0.73_{\pm0.18}$ | $0.81_{\pm0.19}$ | $173.87_{\pm11.70}$ | $0.82_{\pm0.15}$ | $0.52_{\pm0.14}$ | $162.55_{\pm19.18}$ | $0.86_{\pm0.18}$ | $0.79_{\pm0.11}$ | $64.33_{\pm4.70}$ |
| **Ours (Generalist in Sim)** | $\mathbf{144.22}_{\pm13.91}$ | $\mathbf{0.77}_{\pm0.19}$ | $\mathbf{0.54}_{\pm0.03}$ | $\mathbf{224.28}_{\pm23.69}$ | $\mathbf{0.88}_{\pm0.17}$ | $\mathbf{0.58}_{\pm0.09}$ | $\mathbf{314.28}_{\pm27.91}$ | $\mathbf{0.92}_{\pm0.14}$ | $\mathbf{0.37}_{\pm0.05}$ | $\mathbf{242.33}_{\pm23.30}$ | $\mathbf{0.94}_{\pm0.05}$ | $\mathbf{0.46}_{\pm0.06}$ | $\mathbf{88.27}_{\pm3.21}$ |

Table 1: **Generalization Test in Simulation.** Comparisons of the rotation performance on the *unseen test object set* along each axis with hand wrist orientation randomized over rotation metrics.

| Method | "Cube" | | | | "Container" | | | | "Tin Cylinder" | | | | "Gum Box" | | | |
|---|---|---|---|---|---|---|---|---|---|---|---|---|---|---|---|---|
| | Rotation Axis | | Hand Orientation | | Rotation Axis | | Hand Orientation | | Rotation Axis | | Hand Orientation | | Rotation Axis | | Hand Orientation | |
| | Rot (rad) | TTF (s) | Rot (rad) | TTF (s) | Rot (rad) | TTF (s) | Rot (rad) | TTF (s) | Rot | TTF (s) | Rot (rad) | TTF (s) | Rot | TTF (s) | Rot (rad) | TTF (s) |
| **AnyRotate** | $6.53_{\pm1.32}$ | $24.00_{\pm4.30}$ | $5.52_{\pm3.02}$ | $23.00_{\pm10.9}$ | $2.63_{\pm0.75}$ | $25.00_{\pm7.1}$ | $3.70_{\pm1.19}$ | $27.80_{\pm3.1}$ | $5.78_{\pm2.64}$ | $29.7_{\pm0.5}$ | $5.09_{\pm1.51}$ | $28.3_{\pm3.3}$ | $4.08_{\pm3.20}$ | $18.3_{\pm13.1}$ | $5.21_{\pm2.82}$ | $24.2_{\pm11.0}$ |
| **Ours (Direct Transfer)** | $14.92_{\pm1.36}$ | $38.67_{\pm4.16}$ | $8.73_{\pm0.60}$ | $21.89_{\pm2.67}$ | $8.49_{\pm0.36}$ | $40.22_{\pm3.14}$ | $8.81_{\pm0.54}$ | $26.67_{\pm2.02}$ | $9.16_{\pm2.76}$ | $23.67_{\pm8.52}$ | $8.03_{\pm0.30}$ | $29.22_{\pm2.46}$ | $10.65_{\pm1.91}$ | $38.56_{\pm3.50}$ | $5.76_{\pm0.45}$ | $32.50_{\pm2.18}$ |
| **Ours (DexNDM)** | $\mathbf{39.10}_{\pm4.75}$ | $\mathbf{198.39}_{\pm21.65}$ | $\mathbf{10.12}_{\pm1.09}$ | $\mathbf{38.33}_{\pm2.52}$ | $\mathbf{10.79}_{\pm0.54}$ | $\mathbf{45.00}_{\pm2.52}$ | $\mathbf{11.00}_{\pm4.44}$ | $\mathbf{31.50}_{\pm14.85}$ | $\mathbf{15.68}_{\pm3.30}$ | $\mathbf{37.83}_{\pm6.71}$ | $\mathbf{9.42}_{\pm0.52}$ | $\mathbf{35.33}_{\pm3.18}$ | $\mathbf{13.96}_{\pm0.60}$ | $\mathbf{47.22}_{\pm1.07}$ | $\mathbf{7.59}_{\pm0.83}$ | $\mathbf{32.50}_{\pm2.29}$ |

Table 2: **Comparisons to AnyRotate.** Comparison of rotation degrees (Rot (radian)) and time-to-fall (TTF (s)) under two test settings introduced in AnyRotate (Table 12, 13) on replicable objects.

| Method | Cow | Bear | Truck | GRAB Elephant | Bunny | Duck | Teapot | Dragon | Train | Hundepaar | Elephant | Airplane | Mouse |
|---|---|---|---|---|---|---|---|---|---|---|---|---|---|
| **Visual Dexterity** | 7 | **10** | 6 | 3 | 2 | 5 | 8* | 2* | 2* | 3* | 4* | 3* | 4* |
| **DexNDM** | **8** | **10** | **6** | **7** | **5** | **6** | **48** | **4** | **3** | **4** | **4** | **3** | **4** |

Table 3: **Comparisons to Visual Dexterity** of Survival Angles ($\lfloor \text{radian}/0.5\pi \rfloor$), roughly measuring (from videos) how many 90 degrees the object can be rotated before falling. The subscript * denotes the performance achieved by rotating the object with a supporting table.

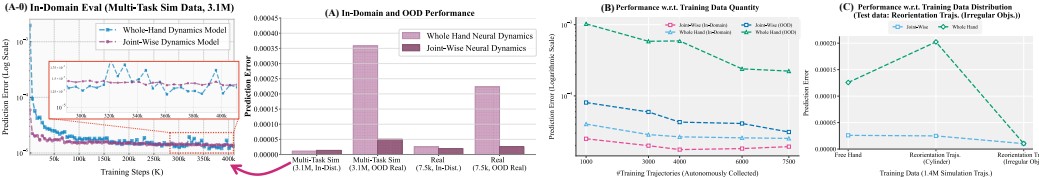

Figure 6: **Comparisons to Whole-Hand Neural Dynamics w.r.t. Model Expressivity, Sample Efficiency and Transferrability.** (A,A-0) In-domain and out-of-distribution performance in high (3.1M) and low (7.5k) data regimes. (B) Sample efficiency. (C) Transferrability from different training distributions.

**Real World Results.** Our sim-to-real method consistently improves real-world performance, and the policy exhibits unprecedented dexterity, rotating high-aspect-ratio geometries, small objects, and complex shapes under challenging hand wrist orientations in the air (Tables 4 (multi-axis with palm-down), 5 (multi-wrist-pose, $z$-rot); Fig. 1; Fig. 22, object gallery (Fig. 21) (in Appendix); videos). Contrary to AnyRotate, which finds "Thumb Up/Down" most difficult, we observe "Base Up/Down" are harder, likely due to different actuator performance between Allegro and LEAP.

*Comparisons to AnyRotate.* We evaluate on four replicable items from AnyRotate's suit—"Tin Cylinder", Cube, "Gum Box", and "Container" (Sec. C)—which are their most difficult cases (according to Table 12-13), and compare with their reported real-world results. Table 2 shows our

| Object Set | Method | ±x-axis | | ±y-axis | | ±z-axis | | Cubic Diagonal Axes | |
| --- | --- | --- | --- | --- | --- | --- | --- | --- | --- |
| | | Rot (rad) | TTF (s) | Rot (rad) | TTF (s) | Rot (rad) | TTF (s) | Rot (rad) | TTF (s) |
| Regular | **Direct Transfer** | $9.84_{\pm0.36}$ | $26.80_{\pm0.20}$ | $10.37_{\pm0.55}$ | $30.73_{\pm1.67}$ | $11.69_{\pm0.30}$ | $21.67_{\pm2.74}$ | $9.03_{\pm0.47}$ | $22.71_{\pm2.04}$ |
| | **Whole Hand NDM** | $5.92_{\pm0.14}$ | $15.04_{\pm1.43}$ | $2.41_{\pm0.22}$ | $8.59_{\pm0.45}$ | $7.38_{\pm0.49}$ | $16.33_{\pm1.79}$ | $3.30_{\pm0.44}$ | $8.87_{\pm0.62}$ |
| | **DexNDM** | $\mathbf{11.36}_{\pm0.40}$ | $\mathbf{32.40}_{\pm1.78}$ | $\mathbf{14.24}_{\pm1.19}$ | $\mathbf{44.60}_{\pm5.44}$ | $\mathbf{23.82}_{\pm3.86}$ | $\mathbf{37.50}_{\pm5.02}$ | $\mathbf{16.93}_{\pm1.84}$ | $\mathbf{30.44}_{\pm3.08}$ |
| Small | **Direct Transfer** | $4.71_{\pm0.00}$ | $25.17_{\pm9.41}$ | $6.11_{\pm0.30}$ | $26.22_{\pm1.90}$ | $6.94_{\pm0.85}$ | $20.17_{\pm0.72}$ | $5.40_{\pm0.32}$ | $23.21_{\pm3.80}$ |
| | **Whole Hand NDM** | $0.35_{\pm0.06}$ | $0.44_{\pm0.08}$ | $0.87_{\pm0.10}$ | $1.33_{\pm0.13}$ | $0.00_{\pm0.00}$ | $0.00_{\pm0.00}$ | $0.26_{\pm0.14}$ | $0.67_{\pm0.21}$ |
| | **DexNDM** | $\mathbf{5.24}_{\pm1.35}$ | $\mathbf{28.00}_{\pm9.13}$ | $\mathbf{6.81}_{\pm0.91}$ | $\mathbf{29.78}_{\pm5.09}$ | $\mathbf{9.29}_{\pm1.63}$ | $\mathbf{26.75}_{\pm5.24}$ | $\mathbf{6.03}_{\pm0.51}$ | $\mathbf{27.34}_{\pm4.97}$ |
| Irregular | **Direct Transfer** | $4.41_{\pm0.34}$ | $19.95_{\pm2.26}$ | $6.13_{\pm0.47}$ | $24.62_{\pm2.54}$ | $5.26_{\pm0.31}$ | $21.19_{\pm2.22}$ | $6.53_{\pm0.37}$ | $26.29_{\pm1.25}$ |
| | **Whole Hand NDM** | $1.34_{\pm0.21}$ | $5.51_{\pm0.36}$ | $2.91_{\pm0.50}$ | $10.32_{\pm0.72}$ | $0.72_{\pm0.06}$ | $4.03_{\pm2.92}$ | $2.33_{\pm0.68}$ | $11.68_{\pm2.05}$ |
| | **DexNDM** | $\mathbf{6.35}_{\pm0.69}$ | $\mathbf{24.21}_{\pm2.87}$ | $\mathbf{11.32}_{\pm2.08}$ | $\mathbf{39.04}_{\pm7.28}$ | $\mathbf{8.61}_{\pm0.76}$ | $\mathbf{29.33}_{\pm1.38}$ | $\mathbf{9.19}_{\pm1.01}$ | $\mathbf{33.14}_{\pm1.86}$ |

Table 4: **Multi-Axis Rotation in Real.** Comparison of rotation degrees (Rot (radian)) and time-to-fall (TTF (s)) along each axis under the palm down wrist orientation. The metric was first averaged over all objects within each trial. We then report avg. ± std of these results across three independent trials.

| Method | Palm Up | | Palm Down | | Base Up | | Base Down | | Thumb Up | | Thumb Down | |
| --- | --- | --- | --- | --- | --- | --- | --- | --- | --- | --- | --- | --- |
| | Rot (rad) | TTF (s) | Rot (rad) | TTF (s) | Rot (rad) | TTF (s) | Rot (rad) | TTF (s) | Rot (rad) | TTF (s) | Rot (rad) | TTF (s) |
| **Direct Transfer** | $10.03_{\pm0.59}$ | $25.63_{\pm2.88}$ | $7.64_{\pm0.32}$ | $20.98_{\pm2.00}$ | $5.40_{\pm0.23}$ | $21.48_{\pm1.04}$ | $4.92_{\pm0.18}$ | $18.37_{\pm0.93}$ | $6.46_{\pm0.20}$ | $25.02_{\pm3.84}$ | $5.90_{\pm0.48}$ | $20.77_{\pm1.10}$ |
| **Whole Hand NDM** | $7.37_{\pm0.25}$ | $20.42_{\pm1.83}$ | $3.46_{\pm0.83}$ | $14.21_{\pm3.72}$ | $4.17_{\pm0.40}$ | $18.22_{\pm4.97}$ | $2.33_{\pm0.41}$ | $7.06_{\pm1.25}$ | $4.79_{\pm0.88}$ | $20.15_{\pm4.46}$ | $1.91_{\pm0.04}$ | $6.33_{\pm0.75}$ |
| **DexNDM** | $\mathbf{14.61}_{\pm1.15}$ | $\mathbf{32.82}_{\pm3.06}$ | $\mathbf{13.20}_{\pm1.71}$ | $\mathbf{29.33}_{\pm3.94}$ | $\mathbf{9.42}_{\pm1.39}$ | $\mathbf{36.00}_{\pm4.67}$ | $\mathbf{7.59}_{\pm1.63}$ | $\mathbf{44.67}_{\pm6.51}$ | $\mathbf{11.93}_{\pm1.29}$ | $\mathbf{28.37}_{\pm2.84}$ | $\mathbf{8.60}_{\pm0.72}$ | $\mathbf{26.93}_{\pm3.06}$ |

Table 5: **Multi-Wrist Orientation Rotation in Real.** Comparison of rotation degrees (Rot (radian)) and time-to-fall (TTF (s)) under six representative hand orientations across direction $z$.

method substantially outperforms AnyRotate and is more versatile: whereas AnyRotate targets moderately sized, simple shapes (min 5cm, max aspect ratio 1.67) with conservative motions, our policy handles smaller objects (3cm) and high aspect ratios (up to 5.3) with sophisticated finger gaiting.

*Comparisons to Visual Dexterity*. A direct comparison with Visual Dexterity (VD) is infeasible due to differing task definitions (axis-oriented continuous rotation vs. goal-oriented reorientation). To enable comparison, we introduce the survival rotation angle metric: the angle an object is rotated before being dropped. We estimate VD's best performance by analyzing their videos. Despite this metric favoring VD (their setup sometimes includes a supporting table), we achieve comparable or superior results on their showcased and replicable objects (Table 3). Besides, we can uniquely manipulate small objects and high aspect ratios as well as handle diverse wrist orientations (Fig. 22).

*Comparisons to Whole-Hand Nueral Dynamics.* We compare against the whole-hand dynamics model to answer: **(Q1)** Does predicting each joint's transition from its own history (without global information) reduce expressivity? **(Q2)** Is our model more sample-efficient? **(Q3)** Does it generalize better? **(A1)** Trained on 3.1M simulated trajectories and evaluated in-domain, our model is nearly as expressive as the whole-hand model (Fig. 6(A, column 1)(A-0)). **(A2)** With limited data—using 7.5k autonomously collected trajectories in the real world (Fig.6(A, column 3)) and across varying real-world dataset sizes (Fig.6(B))—our model achieves better in-domain performance, indicating higher sample efficiency. The advantage is more obvious under insufficient data settings. **(A3)** On an OOD real-world test set (task-relevant transitions under "Thumb Up" wrist), our model generalizes much better in both high- and low-data regimes; see Fig.6(A, column 2,4) and Fig.6(B). Fig. 6(C) systematically studies the cross-domain transferability in various settings. **Summary**: For data-driven neural dynamics, joint-wise model significantly outperform whole-hand models in insufficient-data or train–test distribution-shift settings; with ample data and in-domain evaluation, performance is similar, with only a slight loss in expressivity for joint-wise models.

*Comparisons to ASAP and UAN*. We implement UAN and ASAP, but their resulting policies fail entirely in real-world tests—unable to rotate even a simple cylinder (Fig. 36; videos). We attribute this to an OOD issue: compensators trained solely on free-hand data do not generalize to the interaction dynamics introduced by manipulated objects. Please note that their methods can only use either free-hand data or task-relevant data with object states—difficult and noisy to obtain, and unusable even for compensator training—and cannot leverage our autonomously collected data with randomized object loads; see Sec. D. Our strategy is more tolerant of real-data imperfections (Figs. 8, 9, 36).

**"Sim-to-Sim" Comparisons.** We conduct a cross-simulator transfer evaluation (Isaac Gym to Genesis and MuJoCo). We collect object-loaded rotation data in the target simulator for training. Table 4.2

| Simulator | Genesis | | | MoJoCo | | |
| --- | --- | --- | --- | --- | --- | --- |
| Method | RotR ↑ | TTF ↑ | RotP ↓ | RotR ↑ | TTF ↑ | RotP ↓ |
| Direct Transfer | $72.74_{\pm18.13}$ | $16.83_{\pm4.50}$ | $0.70_{\pm0.17}$ | $82.03_{\pm25.38}$ | $15.33_{\pm1.11}$ | $0.65_{\pm0.07}$ |
| UAN | $87.23_{\pm16.54}$ | $17.81_{\pm1.56}$ | $1.03_{\pm0.05}$ | $99.14_{\pm17.02}$ | $18.67_{\pm1.18}$ | $0.75_{\pm0.14}$ |
| ASAP | $75.72_{\pm11.29}$ | $19.11_{\pm0.74}$ | $1.48_{\pm0.31}$ | $26.25_{\pm6.37}$ | $15.60_{\pm2.30}$ | $1.89_{\pm0.12}$ |
| DexNDM | $\mathbf{111.29}_{\pm33.30}$ | $\mathbf{19.26}_{\pm1.61}$ | $\mathbf{0.66}_{\pm0.18}$ | $\mathbf{124.69}_{\pm14.06}$ | $\mathbf{18.90}_{\pm1.57}$ | $\mathbf{0.57}_{\pm0.09}$ |

Table 6: **"Sim-to-Sim" Transfer.**

shows our method consistently surpasses prior work, owing to designs on dynamics modeling, higher data efficiency, and practical choices (*e.g.*, pre-train in source sim). We find UAN outperforms ASAP, likely because its history-based design better captures object effects. Details in Sec. C.

**Applications.** A meaningful application of in-hand rotation/reorientation policies is their integration into teleoperation systems to enhance manipulation capabilities (Yin et al., 2025). In our work, we demonstrate a similar application using our in-hand rotation policy within a teleoperation setup (with a Meta Quest 3). The robotic hand is controlled by our rotation policy, conditioned on the

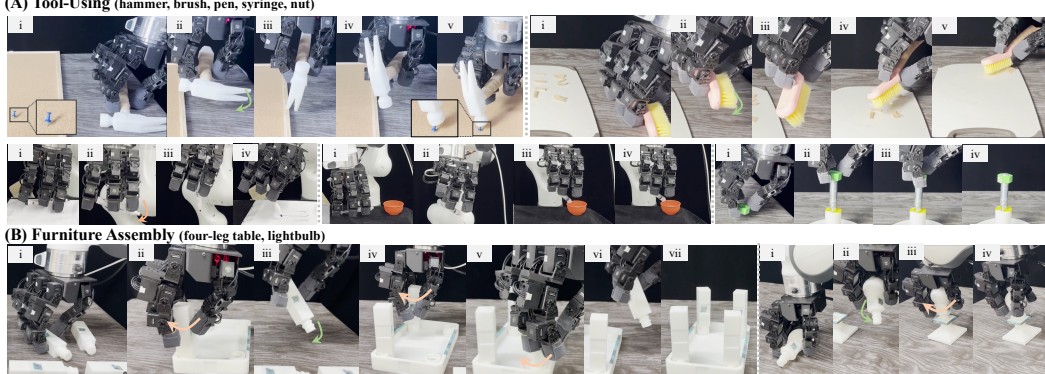

Figure 7: **Application.** Our rotation policy enables a teleoperation system to perform complex, long-horizon manipulation tasks. See videos and more results on our project website.

rotation-axis commands sent by the human operator, while the robot arm is teleoperated through the operator's arm motion. Details are provided in Sec. C. We demonstrate its strong ability in performing long-horizon and complex dexterous manipulation tasks (Fig. 7, videos).

# 5 ABLATION STUDIES

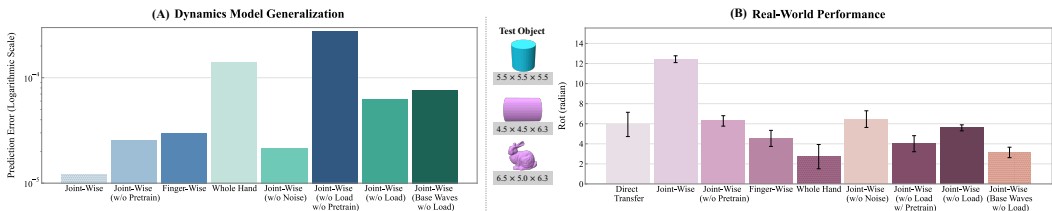

Figure 8: **Ablation Study of the Dynamics Model.** (A) Generalization error of different model ablations (lower is better). (B) Corresponding real-world task performance.

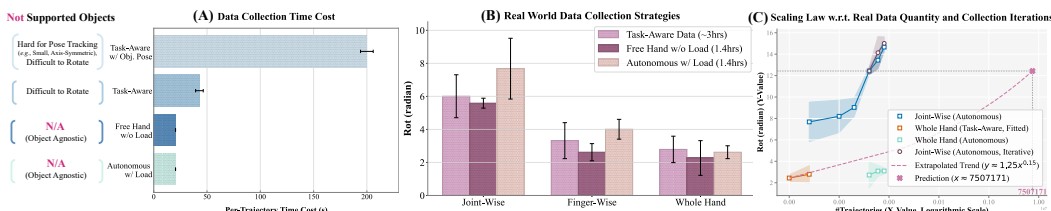

Figure 9: **Analysis of Data Collection Strategies.** (A) Time efficiency of different collection methods. (B) Resulting model performance on datasets of equal size. (C) Performance scaling with dataset size and data collection iterations, including a power-law fit for extrapolation.

We conduct ablations to validate key design choices of our method. Real-world experiments are performed with the hand fixed palm-down, evaluating z-axis rotation; data are collected under the same wrist pose. Dynamics model are evaluated in an OOD test setting. See Sec. C for details.

**Designs on the Joint-Wise Neural Dynamics Model.** We ablate five design choices: (i) joint-wise vs. finger-wise (per finger prediction from its own history) and whole-hand modeling; (ii) simulation pretraining; (iii) injecting noise into replayed actions during real-world data collection; (iv) collecting with object loads rather than free-hand w/o load; and (v) replaying policy rollouts instead of base waves (Fey et al., 2025). As summarized in Fig. 8, these choices consistently improve learned dynamics generalization and real-world performance.

**Real-World Data Collection Strategies.** We compare our autonomous data collection against three baselines—task-aware with vision-based object states, task-aware without object states, and free-hand motions—evaluating limitations, efficiency, and model performance (Figure 9). Task-aware pipelines are slow and intervention-heavy: estimating object poses is prohibitively slow (∼200s on average), requires continuous human supervision, yields noisy poses and complex setup, and fails on small, occluded, or axis-symmetric objects; without vision they still need intervention, remain slow (42.86 s), and produce low-diversity, low-coverage data (data restricted to policy's ability). In contrast, our method is fully automated and, by continuously varying hand loads, collects diverse data spanning a wide range of external influences. Figure 9(B) shows the resulting performance

gains: broader coverage improves prediction, and the joint-wise model is most robust to training-distribution shifts, whereas other variants tend to overfit to the source data.

**Scaling with Real-World Data Quantity and Collection Iterations.** As shown in Fig. 9, our performance improves with more real-world data. However, iterative data collection—intended to align real-world and simulated transition distributions for better policy updates—yields only modest gains. We hypothesize this is because the dynamics model already generalizes well, and adding noise to replay actions provides broad coverage, reducing sensitivity to this distribution shift. In contrast, the whole-hand model benefits little from additional data, especially under autonomous collection, likely due to its higher dimensionality and a distributional mismatch between autonomous data and rotation task transitions. A simple extrapolation suggests matching our 4,000-trajectory result would require 7.5M task-aware trajectories (417k hours; 52k 8-hour workdays), which is impractical. While approximate, this highlights the superiority of our approach.

## 6 CONCLUSIONS AND LIMITATIONS

We propose a neural sim-to-real framework centered on a joint-wise neural dynamics model and autonomous data collection. This enables unprecedented dexterity in rotating challenging objects. **The main limitation** is that the model's performance ceiling is constrained by its reliance on partial, proprioception-only observations, as well as task- and hardware-specific evaluation. Jointly modeling hand–object transitions using richer sensory signals, incorporating vision and tactile feedback, and extending the framework to a broader set of tasks and hardware platforms represent valuable directions for future work.

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

APPENDIX

We include a **video** and a **website** to introduce our work. The website and the video contain robot videos. We highly recommend exploring these resources for an intuitive understanding of the challenges, the effectiveness of our method, and its superiority over prior approaches.

# A  ADDITIONAL EXPLANATIONS OF THE METHOD

## A.1  POLICY DESIGN

**Observations.** The observation of the oracle policy contains: 3-length joint position history (48-dim), 3-length joint positional target history (48-dim), joint velocity (16-dim), fingertip state and velocity (52-dim), object state and velocity (13-dim), object guiding goal pose (4-dim), joint and rigid body forces (40-dim), contact force and binary contact (92-dim), wrist orientation (quaterion, 4-dim), and rotation axis (3-dim).

**Rewards.** The reward function consists of three parts $r = \alpha_{\text{rot}} r_{\text{rot}} + \alpha_{\text{goal}} r_{\text{goal}} + \alpha_{\text{penalty}} r_{\text{penalty}}$, with $r_{\text{rot}}$ and $r_{\text{penalty}}$ following RotateIt (Qi et al., 2023). The rotation term $r_{\text{rot}} = \text{clip}(\omega_t \cdot \mathbf{k}, -c, c)$ encourages rotation about the unit target axis $\mathbf{k} \in \mathbb{R}^3$, $\|\mathbf{k}\|_2 = 1$, where $\omega_t$ is the object angular velocity and $c = 0.5$ caps excessive speed. The penalty $r_{\text{penalty}}$ discourages off-axis angular velocity, deviation from a canonical hand pose, object linear velocity, and joint work/torque: $r_{\text{penalty}} = -\alpha_{\text{rotp}} \|\omega_t \times \mathbf{k}\|_1 - \alpha_{\text{lin}} \|\mathbf{v}_t\|_2^2 - \alpha_{\text{pose}} \|\mathbf{q}_t - \mathbf{q}_{\text{init}}\|_2^2 - \alpha_{\text{work}} \tau^T \dot{\mathbf{q}} - \alpha_{\text{torque}} \|\tau\|_2^2$, where $\mathbf{v}_t$, $\mathbf{q}_{\text{init}}$, and $\tau$ denote the object pose, initial hand joint position, and joint commanded torques at the current timestep $t$, $\alpha_{\text{lin}} = 0.3, \alpha_{\text{pose}} = 0.3, \alpha_{\text{torque}} = 0.1, \alpha_{\text{work}} = 2.0$. We schedule the coefficient $\alpha_{\text{rotp}}$ linearly: set it to zero at the beginning of the training; use the number of resets to count the training process; at the 10 resets, we keep $\alpha_{\text{rotp}}$ to zero; from 10 to 100, linearly increase it to 0.1; after 100, keep it at 0.1. $\alpha_{\text{penalty}} = 1.0$

We find that solely relying on these rewards cannot solve challenging problems like rotating a long object. Therefore, we add an intermediate goal: at episode start set $\mathbf{p}^{\text{goal}}$ 90° ahead along the

desired rotation and update it whenever $\text{ang\_diff}(\mathbf{p}_t, \mathbf{p}^{\text{goal}}) < 15°$; the guidance term is $r_{\text{goal}} = \text{clip}\left(\frac{g_{\text{goal}}}{\text{ang\_diff}(\mathbf{p}_t, \mathbf{p}^{\text{goal}})+\epsilon}, 0, c_{\text{goal}}\right) + g_{\text{bonus}}\mathbf{1}_{\text{ang\_diff}(\mathbf{p}_t, \mathbf{p}^{\text{goal}})<c_{\text{threshold}}}$, where $\text{ang\_diff}(\cdot, \cdot)$ is the quaternion angular distance, $\epsilon > 0$ ensures numerical stability, and $c_{\text{threshold}}$ is the proximity threshold. We set $r_{\text{goal}} = 1.0$.

**Control Strategy.** We use torque control with 20Hz, where each control step is realized by running the torque control for 6 times. Each time the joint torque is calculated as $\tau_t = \mathbf{K}_p(\mathbf{q}_t^{\text{tar}} - \mathbf{q}) - \mathbf{K}_d\dot{\mathbf{q}}_t$, where the $\mathbf{q}$ and $\dot{\mathbf{q}}$ represent the current joint position and joint velocity, $\mathbf{K}_p$ and $\mathbf{K}_d$ are preset constant positional gain and damping parameters.

**Generalist Policy Architecture.** We use a residual MLP with five residual blocks. The input layer is a single linear network with a hidden dimension of 1024. After that, we stack five residual blocks each with the hidden dimension of 1024. Each residual block processes input $\mathbf{x}$ via $\mathbf{y} = \text{ReLU}(\text{NN}_1(\mathbf{x}) + \text{NN}_3(\text{ReLU}(\text{NN}_2(\mathbf{x}))))$. The output layer is a single linear network that maps the latent to the output dimension.

**Further Discussions on Design Choices.** The BC-style training allows us to achieve a real-world deployable multi-geometry policy in a simple way by combining datasets resulting from different multiple oracle policies, each trained for a specific object category, to train a unified policy. We use BC to achieve both real-world deployment ability and generality across diverse objects. An alternative is achieving the generality in the teacher level, *e.g.,* training RL for an any-wrist orientation any-axis on all object categories. However, this can hardly work. This may require us to add an automatic or multi-stage curriculum to make sure the final policy can perform at least as good as each individual policy. This is a valuable research direction. In this work, we choose to leave the oracle policy training a neat pipeline, adopt to train a collection of teacher policies, and achieve the unified real-world deployable policy at once in the student policy training stage.

## A.2 Proof of Main Theorems

**Theorem A.1 (Data Processing Inequality for KL (strict form))** *Let $\mathcal{P}$ and $\mathcal{Q}$ be two probability distributions on $\mathbb{R}^n \times \mathbb{R}$ with respective probability density functions (PDFs) $P(x)$ and $Q(x)$. Let $g: \mathbb{R}^n \times \mathbb{R} \to \mathbb{R}^m \times \mathbb{R}$ be a measurable function, where $m \leq n$. This function transforms a random variable $X \sim \mathcal{P}$ (or $X \sim \mathcal{Q}$) into a new random variable $Y = g(X)$. Let $g(\mathcal{P})$ and $g(\mathcal{Q})$ denote the resulting pushforward distributions on $\mathbb{R}^m \times \mathbb{R}$.*

*The Kullback-Leibler (KL) divergence between the distributions is reduced or remains the same after the transformation, a property known as the Data Processing Inequality:*

$$\text{KL}(\mathcal{P}\|\mathcal{Q}) \geq \text{KL}(g(\mathcal{P})\|g(\mathcal{Q})). \tag{1}$$

*The inequality is strict, $\text{KL}(\mathcal{P}\|\mathcal{Q}) > \text{KL}(g(\mathcal{P})\|g(\mathcal{Q}))$, if $g$ is non-injective in a way that merges points where $\mathcal{P}$ and $\mathcal{Q}$ have a different relative structure. More concretely, it indicates that there $\exists y_0 \in \mathbb{R}^m \times \mathbb{R}, P(Y = y_0) > 0, P(X|Y = y_0) \neq Q(X|Y = y_0)$.*

**Proof A.1** *We start with prove that $\text{KL}(\mathcal{P}\|\mathcal{Q}) \geq \text{KL}(g(\mathcal{P})\|g(\mathcal{Q}))$ always holds for any function $g$. Let $X$ be a random variable drawn from one of two distributions, $\mathcal{P}$ or $\mathcal{Q}$. Denote their PDFs as $P_X(x)$ and $Q_X(x)$.*

*Let $Y$ be a new random variable created by applying a function to $X$: $Y = g(X)$. The distributions of $Y$ are the pushforward distributions $f(\mathcal{P})$ and $f(\mathcal{Q})$, with PDFs $P_Y(y)$ and $Q_Y(y)$. Consider the joint distribution of $(X, Y)$, since $Y$ is a deterministic function of $X$, the joint probability is simple:*

$$P_{X,Y}(x,y) = P_X(x), \text{if } y = g(x) \tag{2}$$
$$P_{X,Y}(x,y) = 0, \text{if } y \neq g(x) \tag{3}$$

*Using "chain rule" of KL divergence, we can expand the joint distributions in two ways:*

$$(A)\ \text{KL}(P_{X,Y}\|Q_{X,Y}) = \text{KL}(P_X\|Q_X) + \text{KL}(P_{Y|X}\|Q_{Y|X}) \tag{4}$$
$$(B)\ \text{KL}(P_{X,Y}\|Q_{X,Y}) = \text{KL}(P_Y\|Q_Y) + \text{KL}(P_{X|Y}\|Q_{X|Y}) \tag{5}$$

*Since $Y$ is completely determined by $X$ ($Y = f(X)$), we have*

$$P(y|x) = 1, \text{ if } y = f(x), \tag{6}$$
$$P(y|x) = 0, \text{ if } y \neq f(x) \tag{7}$$

*And the same property for $Q(y|x)$:*

$$Q(y|x) = 1, \text{ if } y = f(x), \tag{8}$$

$$Q(y|x) = 0, \text{ if } y \neq f(x) \tag{9}$$

*Therefore $P_{Y|X} = Q_{Y|X}$, and the KL divergence between them is zero:*

$$\mathrm{KL}(P_{Y|X}\|Q_{Y|X}) = \mathbb{E}_{x \sim P_X} \int_y P(y|x) \log\left(\frac{P(y|x)}{Q(y|x)}\right) dy = \mathbb{E}_{x \sim P_X}[0] = 0. \tag{10}$$

*Thus, the expansion 4 simplifies to*

$$\mathrm{KL}(P_{X,Y}\|Q_{X,Y}) = \mathrm{KL}(P_X\|Q_X). \tag{11}$$

*We have:*

$$\mathrm{KL}(P_X\|Q_X) = \mathrm{KL}(P_Y\|Q_Y) + \mathrm{KL}(P_{X|Y}\|Q_{X|Y}). \tag{12}$$

*Since KL divergence is always non-negative, which implies $\mathrm{KL}(P_{X|Y}\|Q_{X|Y}) \geq 0$, we have*

$$\mathrm{KL}(P_X\|Q_X) \geq \mathrm{KL}(P_Y\|Q_Y). \tag{13}$$

*The inequality is strict if and only if the second term of the RHS in Eq. 12 is strictly positive, i.e., $\mathrm{KL}(P_{X|Y}\|Q_{X|Y}) > 0$. This term is the expected KL divergence between the conditional distributions $P(x|y)$ and $Q(x|y)$, averaged over the distribution $P_Y(y)$. It will be strictly positive if and only if $\exists y_0 \in \mathbb{R}^m \times \mathbb{R}, P_Y(y_0) > 0, P(X|Y = y_0) \neq Q(X|Y = y_0)$.*

*This is direct. We provide the proof below.*

**Sufficiency.** *Since $\mathrm{KL}(P_{X|Y}\|Q_{X|Y}) = \mathbb{E}_{y\sim P_Y}\left[\mathrm{KL}(P_{X|Y=y}\|Q_{X|Y=y})\right]$, if the condition is satisfied, we have $\mathrm{KL}(P_{X|Y}\|Q_{X|Y}) \geq P_Y(y_0)\mathrm{KL}(P_{X|Y=y_0}\|Q_{X|Y=y_0}) > 0$. Thus, it is a sufficient condition.*

**Necessity.** *We can prove it by disproof. Suppose that we can find a case with $\mathrm{KL}(P_{X|Y}\|Q_{X|Y}) > 0$ but for every $y_0$ with non-zero $P_Y(y_0)$, we have $\mathrm{KL}(P_{X|Y=y_0}\|Q_{X|Y=y_0}) = 0$, then we have $\mathrm{KL}(P_{X|Y}\|Q_{X|Y}) = \mathbb{E}_{y\sim P_Y}\left[\mathrm{KL}(P_{X|Y=y}\|Q_{X|Y=y})\right] = 0$, which contradicts the assumptions. Thus, it is a necessary condition.*

In our setting, as $g$ strictly reduces the dimensionality and is a continuous function (because it extracts the history of a joint from the whole hand history), $g$ is a non-injective function, which we will show later in Theorem A.3. Since $\mathcal{P}$ and $\mathcal{Q}$ lie in different data domains (a visualization is shown in Figs. 16 17), and since as we've demonstrated $g(\mathcal{P})$ and $g(\mathcal{Q})$ share similarities (a visualization is shown in Fig. 15), the condition $\exists y_0 \in \mathbb{R}^m \times \mathbb{R}, P(Y = y_0) > 0, P(X|Y = y_0) \neq Q(X|Y = y_0)$ is then typically satisfied.

**Theorem A.2 (Generalization Gap Contraction)** *Given data point $(X, Y) \in \mathbb{R}^n \times \mathbb{R}$, a measurable function $g : (X, Y) \in \mathbb{R}^n \to (g_X(X), Y) \in \mathbb{R}^m, m < n$, and two different distributions $\mathcal{P}, \mathcal{Q}$ in the manifold $\mathbb{R}^n$ whose pushforward distribution by $g$ satisfy $KL(g(\mathcal{P})\|g(\mathcal{Q})) < KL(\mathcal{P}\|\mathcal{Q})$. Under the covariant shift condition, i.e., $\mathcal{P}(Y|X) = \mathcal{Q}(Y|X)$, for any function $f_1 : X \in \mathbb{R}^n \to Y \in \mathbb{R}$ and $f_2 : g_X(X) \in \mathbb{R}^m \to Y \in \mathbb{R}$, we have*

$$\sup|R_\mathcal{P}(f_2 \circ g_X) - R_\mathcal{Q}(f_2 \circ g_X)| < \sup|R_\mathcal{P}(f_1) - R_\mathcal{Q}(f_1)|, \tag{14}$$

*where $R_\mathcal{P}(h) = \mathbb{E}_{(X,Y)\sim\mathcal{P}}[L(h(X), Y)]$ is the risk for the predictor $h$, $L$ measures prediction error and is bounded by B.*

**Proof A.2** *Using the law of total expectation and the covariate shift assumption:*

$$R_\mathcal{P}(h) = \mathbb{E}_{X\sim P_X}\left[\mathbb{E}_{Y\sim P(Y|X)}[L(h(X), Y)]\right]$$

$$R_\mathcal{Q}(h) = \mathbb{E}_{X\sim Q_X}\left[\mathbb{E}_{Y\sim Q(Y|X)}[L(h(X), Y)]\right] = \mathbb{E}_{X\sim Q_X}\left[\mathbb{E}_{Y\sim P(Y|X)}[L(h(X), Y)]\right]$$

*Define the "inner risk" function for a fixed $x$:*

$$r_h(x) := \mathbb{E}_{Y\sim P(Y|X=x)}[L(h(x), Y)]$$

*The risk difference could be converted to an expectation over the marginals $P_X$ and $Q_X$:*

$$R_\mathcal{P}(h) - R_\mathcal{Q}(h) = \mathbb{E}_{X\sim P_X}[r_h(X)] - \mathbb{E}_{X\sim Q_X}[r_h(X)] = \int r_h(x)(p_X(x) - q_X(x))dx$$

*An IPM between two distributions $P_X$ and $Q_X$ over a function class $\mathcal{F}$ is defined as:*

$$d_{\mathcal{F}}(P_X, Q_X) = \sup_{\phi \in \mathcal{F}} |\mathbb{E}_{X \sim P_X}[\phi(X)] - \mathbb{E}_{X \sim Q_X}[\phi(X)]|$$

*Define two classes of "inner risk" functions:*

$$\mathcal{F}_1 = \{r_{f_1} \mid f_1 : \mathbb{R}^n \to \mathbb{R} \text{ is in the function space for } f_1\}$$
$$\mathcal{F}_2 = \{r_{f_2 \circ g_X} \mid f_2 : \mathbb{R}^m \to \mathbb{R} \text{ is in the function space for } f_2\}$$

*The inequality we want to prove becomes:*

$$d_{\mathcal{F}_2}(P_X, Q_X) < d_{\mathcal{F}_1}(P_X, Q_X)$$

*Consider any function $\phi \in \mathcal{F}_2$. By definition, $\phi = r_{f_2 \circ g_X}$ for some function $f_2$. Define a new function $f_1(x) = (f_2 \circ g_X)(x)$. Assuming the $\mathcal{F}_1$ is rich enough to contain this composition, we have $r_{f_1} = r_{f_2 \circ g_X} = \phi$. This means $\phi \in \mathcal{F}_1$. Therefore, $\mathcal{F}_2 \subseteq \mathcal{F}_1$.*

*We immediately have the non-strict inequality, since we are taking the supremum over a smaller set:*

$$\sup_{\phi \in \mathcal{F}_2} |\mathbb{E}_{P_X}[\phi] - \mathbb{E}_{Q_X}[\phi]| \leq \sup_{\phi \in \mathcal{F}_1} |\mathbb{E}_{P_X}[\phi] - \mathbb{E}_{Q_X}[\phi]|$$

*Consider the given KL condition $\mathrm{KL}(g(P_X)\|g(Q_X)) \leq \mathrm{KL}(P_X\|Q_X)$ and the covariant shift condition, we have: $\mathrm{KL}(g_X(P_X)\|g_X(Q_X)) < \mathrm{KL}(P_X\|Q_X)$. This implies that $g_X(X)$ is **not** a sufficient statistic for distinguishing $P_X$ from $Q_X$. This means the likelihood ratio $w(x) = p_X(x)/q_X(x)$ cannot be written as a function of $g_X(x)$. This further implies there exist $x_a, x_b$ such that $g_X(x_a) = g_X(x_b)$ but $w(x_a) \neq w(x_b)$.*

*Now, consider the function classes:*

- *Any function $\phi \in \mathcal{F}_2$ must be constant on the level sets of $g_X$. If $g_X(x_a) = g_X(x_b)$, then $\phi(x_a) = \phi(x_b)$. These functions are blind to the information that $g_X$ discards.*

- *The function $\phi^* \in \mathcal{F}_1$ that maximizes the IPM difference, $d_{\mathcal{F}_1}(P_X, Q_X)$, must be maximally sensitive to the difference between $P_X$ and $Q_X$. Since this difference (captured by the likelihood ratio $w(x)$) depends on information discarded by $g_X$, the optimal discriminating function $\phi^*$ cannot be a function of $g_X(x)$ alone.*

*This means that the function $\phi^*$ that achieves the supremum for the larger set $\mathcal{F}_1$ is not contained in the smaller set $\mathcal{F}_2$ (i.e., $\phi^* \notin \mathcal{F}_2$).*

*Because the supremum for $\mathcal{F}_1$ is achieved by a function that is not available in the strictly smaller set $\mathcal{F}_2$, the inequality is strict.*

$$\sup_{\phi \in \mathcal{F}_2} |\mathbb{E}_{P_X}[\phi] - \mathbb{E}_{Q_X}[\phi]| < \sup_{\phi \in \mathcal{F}_1} |\mathbb{E}_{P_X}[\phi] - \mathbb{E}_{Q_X}[\phi]|$$

*This completes the proof.*

Define the optimal predictors trained on the source distribution $\mathcal{Q}$ as:

$$f_1^{\mathcal{Q}} = \arg\min_{f_1} R_{\mathcal{Q}}(f_1) \tag{15}$$

$$f_2^{\mathcal{Q}} = \arg\min_{f_2} R_{\mathcal{Q}}(f_2 \circ g_X) \tag{16}$$

We move on to show that under specific conditions, the predictor trained on the simpler representation generalizes better to the target distribution $\mathcal{P}$.

**Proposition** *Let $f_1^{\mathcal{Q}}$ and $f_2^{\mathcal{Q}}$ be the optimal predictors on the source distribution $\mathcal{Q}$ in the full and reduced-dimensional spaces, respectively. Let the following assumptions hold:*

**Assumption (Small Approximation Error)** *The function class $\{f_2 \circ g_X \mid f_2 : \mathbb{R}^m \to \mathbb{R}\}$ is sufficiently expressive to model the relationship on the source distribution $\mathcal{Q}$. The increase in source risk due to the reduced representation is bounded by a small constant $\epsilon_A$:*

$$R_{\mathcal{Q}}(f_2^{\mathcal{Q}} \circ g_X) - R_{\mathcal{Q}}(f_1^{\mathcal{Q}}) = \epsilon_A. \tag{17}$$

**Assumption (Generalization Gap Reduction)** *Building on Theorem A.2, we further assume a relatively large distribution shift from $\mathcal{P}$ to $\mathcal{Q}$, such that $f_2^{\mathcal{Q}}$ exhibits a strong generalization advantage, and the difference in generalization gap achieved by the $f_1^{\mathcal{Q}}$ and $f_2^{\mathcal{Q}}$ satisfies:*

$$\left(R_{\mathcal{P}}(f_2^{\mathcal{Q}} \circ g_X) - R_{\mathcal{Q}}(f_2^{\mathcal{Q}} \circ g_X)\right) - \left(R_{\mathcal{P}}(f_1^{\mathcal{Q}}) - R_{\mathcal{Q}}(f_1^{\mathcal{Q}})\right) = -\epsilon_B, \tag{18}$$

*where $\epsilon_B$ is a positive constant.*

*If $\epsilon_B > \epsilon_A$, then the risk of the predictor trained in the reduced-dimensional space is strictly lower on the target distribution:*

$$R_{\mathcal{P}}(f_2^{\mathcal{Q}} \circ g_X) < R_{\mathcal{P}}(f_1^{\mathcal{Q}}). \tag{19}$$

**Proof A.3** *Decompose the target risk:*

$$R_{\mathcal{P}}(h) = R_{\mathcal{Q}}(h) + \left(R_{\mathcal{P}}(h) - R_{\mathcal{Q}}(h)\right). \tag{20}$$

*We further have:*

$$R_{\mathcal{P}}(f_2^{\mathcal{Q}} \circ g_X) - R_{\mathcal{P}}(f_1^{\mathcal{Q}}) = \left[R_{\mathcal{Q}}(f_2^{\mathcal{Q}} \circ g_X) + \left(R_{\mathcal{P}}(f_2^{\mathcal{Q}} \circ g_X) - R_{\mathcal{Q}}(f_2^{\mathcal{Q}} \circ g_X)\right)\right]$$
$$- \left[R_{\mathcal{Q}}(f_1^{\mathcal{Q}}) + \left(R_{\mathcal{P}}(f_1^{\mathcal{Q}}) - R_{\mathcal{Q}}(f_1^{\mathcal{Q}})\right)\right]. \tag{21}$$

*Rearranging the terms, we have:*

$$R_{\mathcal{P}}(f_2^{\mathcal{Q}} \circ g_X) - R_{\mathcal{P}}(f_1^{\mathcal{Q}}) = \underbrace{\left[R_{\mathcal{Q}}(f_2^{\mathcal{Q}} \circ g_X) - R_{\mathcal{Q}}(f_1^{\mathcal{Q}})\right]}_{\text{Term A: Approximation Error}}$$
$$+ \underbrace{\left[\left(R_{\mathcal{P}}(f_2^{\mathcal{Q}} \circ g_X) - R_{\mathcal{Q}}(f_2^{\mathcal{Q}} \circ g_X)\right) - \left(R_{\mathcal{P}}(f_1^{\mathcal{Q}}) - R_{\mathcal{Q}}(f_1^{\mathcal{Q}})\right)\right]}_{\text{Term B: Difference in Generalization Gaps}}. \tag{22}$$

*From Assumption 1, Term A is equal to $\epsilon_A$:*

$$R_{\mathcal{Q}}(f_2^{\mathcal{Q}} \circ g_X) - R_{\mathcal{Q}}(f_1^{\mathcal{Q}}) = \epsilon_A. \tag{23}$$

*From Assumption 2, Term B is equal to $-\epsilon_B$:*

$$\left(R_{\mathcal{P}}(f_2^{\mathcal{Q}} \circ g_X) - R_{\mathcal{Q}}(f_2^{\mathcal{Q}} \circ g_X)\right) - \left(R_{\mathcal{P}}(f_1^{\mathcal{Q}}) - R_{\mathcal{Q}}(f_1^{\mathcal{Q}})\right) = -\epsilon_B. \tag{24}$$

*We have:*

$$R_{\mathcal{P}}(f_2^{\mathcal{Q}} \circ g_X) - R_{\mathcal{P}}(f_1^{\mathcal{Q}}) = \epsilon_A - \epsilon_B. \tag{25}$$

*Given the condition $\epsilon_B > \epsilon_A$, we have:*

$$R_{\mathcal{P}}(f_2^{\mathcal{Q}} \circ g_X) < R_{\mathcal{P}}(f_1^{\mathcal{Q}}). \tag{26}$$

*This completes the proof.*

**When are these assumptions valid?** Assumption 1 characterizes the in-domain performance gap between the joint-wise neural dynamics model and the whole-hand model. As shown in Sec. 4.2 and Fig. 6, it holds even when data are sufficient. In low-data regimes, the joint-wise model not only avoids increasing source-domain risk but actually reduces it, thanks to better sample efficiency.

Assumption 2 characterizes the generalization behavior of these two models. Under train–test distribution shift, it is satisfied in all our experiments (Sec. 4.2; Fig. 6); the joint-wise model exhibits **much better** transferability than the whole-hand dynamics model.

In our dexterous manipulation setting, data scarcity and train–test shift are pervasive, because obtaining perfectly distributionally aligned data is often infeasible or difficult to scale (Sec. 3.3), with empirical evidence in Secs. 5 and B.4. Even with autonomous data collection, the volume of real-world data is far smaller than in simulation, keeping us in the low-data regime. Consequently, joint-wise modeling is the preferable choice for our task and a key to our success. By contrast, using a whole-hand dynamics model degrades sim-to-real transfer (Tables 4 and 5). We attribute the success of the whole-body dynamics model employed in bin Shi et al. (2024) to its in-distribution setting and to dynamics that are less complex than in our scenario.

**Theorem A.3** $\forall C^1$ *function* $f : \mathbb{R}^n \to \mathbb{R}^m, m < n$ *that projects* $n$-*dim data point in* $\mathbb{R}^n$ *to that in a lower dimensional space* $\mathbb{R}^m$, *then* $f$ *is a non-injective function.*

**Proof A.4** *For any point* $\mathbf{x} \in \mathbb{R}^n$, *its derivative is the Jacobian matrix* $Df_{\mathbf{x}}$, *which represents a linear map from the tangent space at* $\mathbf{x}$ *(i.e.,* $\mathbb{R}^n$) *to the tangent space at* $f(\mathbf{x})$ *(i.e.,* $\mathbb{R}^m$). $Df_{\mathbf{x}}$ *is an* $m \times n$ *matrix. The rank of this matrix is at most* $\min(m, n) = m$. *Applying the Rank-Nullity Theorem to this linear map* $Df_{\mathbf{x}} : \mathbb{R}^n \to \mathbb{R}^m$, *we find that its null space has dimension* $\geq n - m > 0$. *According the Inverse Function Theorem (Munkres, 2018; Guillemin & Pollack, 2010), which states that a function is locally injective around a point* $\mathbf{x}$ *only if its derivative* $Df_{\mathbf{x}}$ *is injective. As we've shown,* $Df_{\mathbf{x}}$ *is never injective when* $n > m$. *Since* $f$ *is not locally injective at any point, it cannot possibly be globally injective.*

### A.3 RATIONALITY OF JOINT-WISE DYNAMICS MODELING (PART I)

We model the hand with the standard manipulator equation (Murray et al., 2017; Spong et al., 2020), treating the object effect as an external force:

$$\mathbf{M}(\mathbf{q})\ddot{\mathbf{q}} + \mathbf{C}(\mathbf{q}, \dot{\mathbf{q}})\dot{\mathbf{q}} + \mathbf{G}(\mathbf{q}) = \tau + \tau_{\text{ext}}, \tag{27}$$

where $\mathbf{M}(\mathbf{q})$, $\mathbf{C}(\mathbf{q}, \dot{\mathbf{q}})$, and $\mathbf{G}(\mathbf{q})$ are the inertia, Coriolis, and gravity matrices, respectively. $\tau$ is the applied joint torque, and $\tau_{\text{ext}}$ represents the external force from the object. Given low-speed operation, we neglect the Coriolis term (Craig, 2009; Spong et al., 2005), $\mathbf{C}(\mathbf{q}_t, \dot{\mathbf{q}}_t)\dot{\mathbf{q}}_t \approx 0$.

Assuming we are modeling the $i$-th joint, we use $(\mathbf{q}^m, \dot{\mathbf{q}}^m)$ to represent the state of "modeled joints", *e.g.,* $\mathbf{q}^m = [\mathbf{q}^i]^T \in \mathbb{R}^1$, while treating the joints as "slave" joints and denote their state as $(\mathbf{q}^s, \dot{\mathbf{q}}^s)$, *i.e.,* $\mathbf{q}^s = [\mathbf{q}^j, \forall 1 \leq j \leq 16, j \neq i]^T \in \mathbb{R}^{15}$. Rearranging other full dynamic equations (Eq. 27), we write it as

$$\begin{bmatrix} \mathbf{M}_t^{mm} & \mathbf{M}_t^{ms} \\ \mathbf{M}_t^{sm} & \mathbf{M}_t^{ss} \end{bmatrix} \begin{bmatrix} \ddot{\mathbf{q}}_t^m \\ \ddot{\mathbf{q}}_t^s \end{bmatrix} + \begin{bmatrix} \mathbf{G}_t^m \\ \mathbf{G}_t^s \end{bmatrix} = \begin{bmatrix} \tau_t^{m,\text{total}} \\ \tau_t^{s,\text{total}} \end{bmatrix}. \tag{28}$$

Derive the equation of the modeled joints:

$$(\mathbf{M}^{mm} - \mathbf{M}^{ms}(\mathbf{M}^{ss})^{-1}\mathbf{M}^{sm})\ddot{\mathbf{q}}^m + \mathbf{M}^{ms}(\mathbf{M}^{ss})^{-1}(\tau^{s,\text{total}} - \mathbf{G}^s) + \mathbf{G}^m = \tau^m = [\tau^i + \tau^{i,\text{ext}}]^T. \tag{29}$$

Introducing an "effective" torque as $\tau^{\text{eff}} = [\tau^{i,\text{ext}}]^T \in \mathbb{R}^1$, and write the equation as follows:

$$(\mathbf{M}^{mm} - \mathbf{M}^{ms}(\mathbf{M}^{ss})^{-1}\mathbf{M}^{sm})\ddot{\mathbf{q}}^m + \mathbf{M}^{ms}(\mathbf{M}^{ss})^{-1}(\tau^{s,\text{total}} - \mathbf{G}^s) + \mathbf{G}^m - \tau^{\text{eff}} = [\tau_i]^T. \tag{30}$$

Let $\mathbf{H}_t^{\text{eff}}$ denote the effective inertia matrix, $\mathbf{H}_t^{\text{eff}} \triangleq \mathbf{M}^{mm} - \mathbf{M}^{ms}(\mathbf{M}^{ss})^{-1}\mathbf{M}^{sm}$, and let $\mathbf{G}_t^{\text{eff}}$ denote the effective external term, $\mathbf{G}_t^{\text{eff}} \triangleq \mathbf{M}^{ms}(\mathbf{M}^{ss})^{-1}(\tau^{s,\text{total}} - \mathbf{G}^s) + \mathbf{G}^m - \tau^{\text{eff}}$. Given $\mathbf{H}_t^{\text{eff}}$, $\mathbf{G}_t^{\text{eff}}$, and the modeled joint torque $\tau_t^i$, the acceleration $\ddot{\mathbf{q}}_t^i$ is uniquely determined. $\mathbf{H}_t^{\text{eff}}$ and $\mathbf{G}_t^{\text{eff}}$ are related to joint state and torques of other joints.

It indicates that in the highly coupled interaction system, the dynamics of each single joint is related to other joints' states, torque, and the external influence of the objects. Employing a neural-based approach to solve the dynamics evolution with the aim to account for all of those high-DoF influences would inevitably require a large amount of data with correct distribution, cannot resolve the challenges in the data aspect.

Focusing on each single joint dynamics system, joint-wise neural dynamics predicts each single joint transition from its own state-action history. Predicting from history generalizes the idea of the RMA approach in rotation (Qi et al., 2022) to implicitly account for time-varying influences at a high level. We will show that, in a short time window (*e.g.,* 10 frames, corresponding to 0.5s) and under certain assumptions, this approach is reasonable.

Specifically, we assume that in any short time window during the action trajectory execution, the state trajectory of each slave joint, *i.e.,* $\mathbf{q}^s$, the active torque applied to each slave joint, *i.e.,* $\tau^s$, and the effective external torque applied to each joint, $\tau^{\text{ext}}$, can be approximated by an infinitely differentiable continuous function to within an acceptable error threshold. Intuitively, this assumption holds true for joint states and active torques (related to input positional targets) in a continuously evolved dynamical system where the actions are the policy network's output. If we further assume a soft contact model (Tedrake & the Drake Development Team, 2019; Pang & Tedrake, 2021), the

assumption of the effective external torques, which is caused by contact forces with the object, is thus reasonable.

We give statistical evidence for these two assumptions. Specifically, we demonstrate that they could be fitted to an acceptable error using polynomial functions, a special group of infinitely differentiable continuous functions.

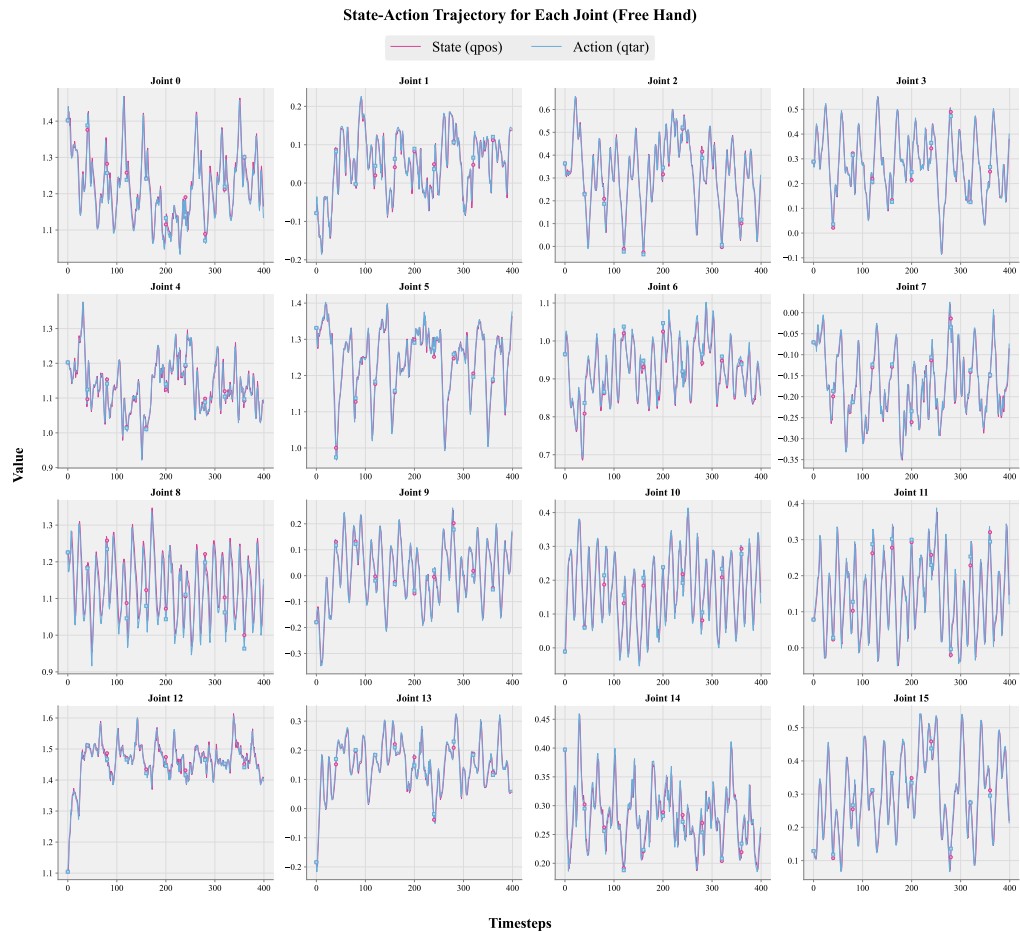

Figure 10: **Per-Joint State-Action Sequences (Free Hand, w/o Load).**

**Patterns of Per-Joint State Trajectory.** Figure 10, 11, and 12 show the real-world state-action trajectories collected using a free robot hand without object load, via our autonomous data collection system with load, and the task-aware data collection with human interventions. Both action and state trajectories of the hand under such three types of external influences are visually smooth.

We further analyze their polynomial fitting results. Figure 41 shows the 3-ordered polynomial fitting results of per-joint state sequence over a 10-length time window. Figure 43 shows the per-joint fitting error averaged over all tested 10-length sequences. We can observe good fitting results where the original curve can be roughly approximated by the fitted curve. If we increase the polynomial order to 5, we could observe excellent fitting results (Figure 42 44). These statistical results show the rationality of the continuous function assumption on joint state sequences.

**Patterns of Per-Joint Active Torque Trajectory.** Since we cannot sense the torque directly, for each joint $i$, we analyze the difference between the positional target and the joint state at each timestep $t$, *i.e.,* $\mathbf{q}_t^{i,\text{tar}} - \mathbf{q}_t^i$, to reflect the corresponding statistics of actuation torques. Figure 45 and 46 illustrate the fitting results using 3-ordered polynomial functions and 5-ordered polynomial functions, respectively. Figure 47 and 48 further show the per-joint average fitting error. The action

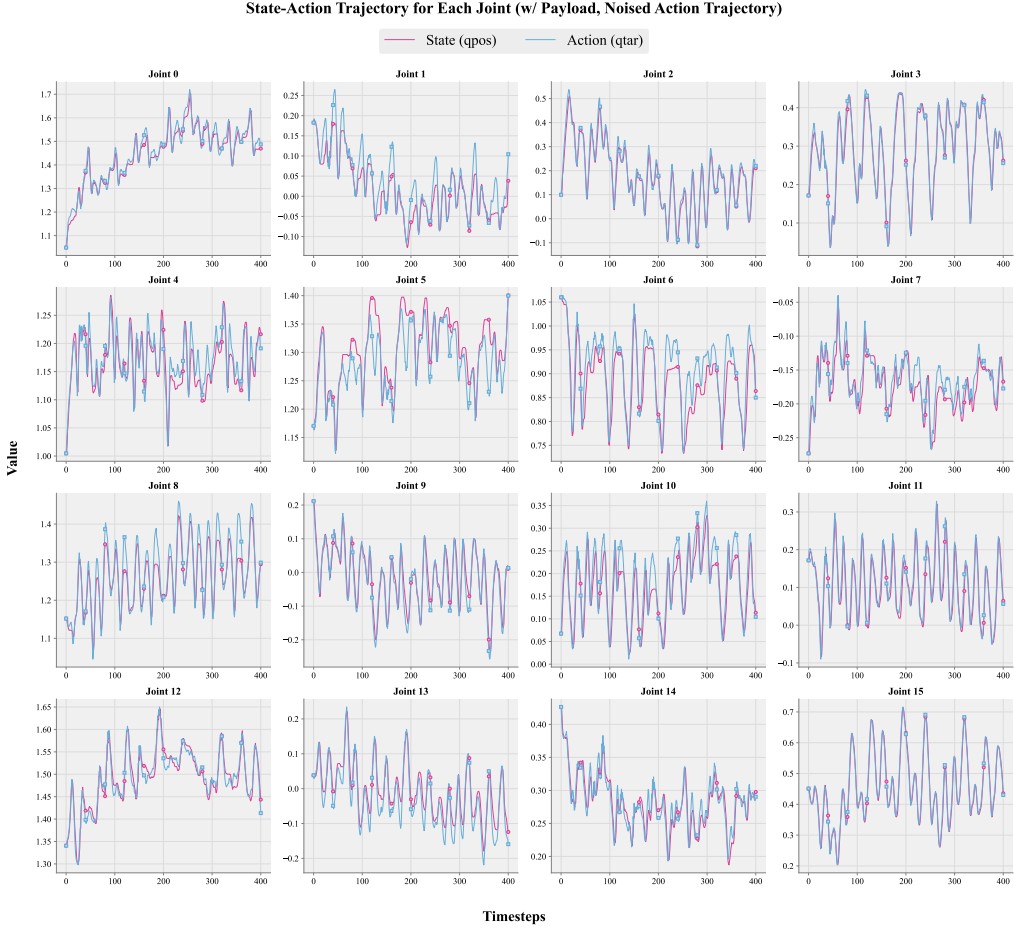

Figure 11: **Per-Joint State-Action Sequences (Autonomous Data Collection, w/ Load).**

force's evolution is more complex than joint states. But we could still see satisfactory fitting results. As the polynomial order increases, the fitting results become better.

**Patterns of Per-Joint External Torques Trajectory.** Since we cannot measure per-joint effective external torques from the real world directly, which is related to the contact force between the object and the hand, we introduce "virtual object force" (also denoted as "virtual force" or "virtual torque") as a proxy of the actual external torque. Specifically, we first train per-joint inverse dynamics models that predicts the applied action from the state-action history and the next actual state, *i.e.,* $f^{\text{invdyn},i}$ : $\{(\mathbf{s}_{k+1}^i, \mathbf{a}_k^i)\}_{k=t-W+1}^{t} \in \mathbb{R}^{2W} \to \hat{\mathbf{a}}_{t+1} \in \mathbf{R}^{2W}$, from the **free hand** replay trajectories. Thus, it predicts what action should be applied so that the next joint state can reach the desired value, without the influence of the object (without the external torques). Then, for a collected task-aware trajectory, we first use the inverse dynamics model to predict the desired action $\hat{\mathbf{a}}_{t+1}$. We then calculate the "virtual force" using its difference from the actual action, *i.e.,* $\mathbf{a}_{t+1} - \hat{\mathbf{a}}_{t+1}$. Since this discrepancy reflects what amount of additional action is required to resist the object so that the joint can reach the desired state. We then analyze the statistics of this quantity.

As shown in Figure 49, 50, 51, 52, we can still get satisfactory fitting results, although the evolution of this quantity is more complex than both that of the active torque and the joint state.

Based on this, we can assume the evolution of $\mathbf{H}^{\text{eff}}$ and $\mathbf{G}^{\text{eff}}$ are good continuous functions over the considered time window. We can then approximate their evolution by a low-order function, *e.g.,* using its Taylor expansions, to an acceptable error. Assuming $k_1$ order for $\mathbf{H}^{\text{eff}}$ while $k_2$ for $\mathbf{G}^{\text{eff}}$, the underlying number of unknown variables becomes $k_1 + k_2$. Solving for all unknown variables is enough to solve the next step transition. The state-action history of each joint could be viewed as

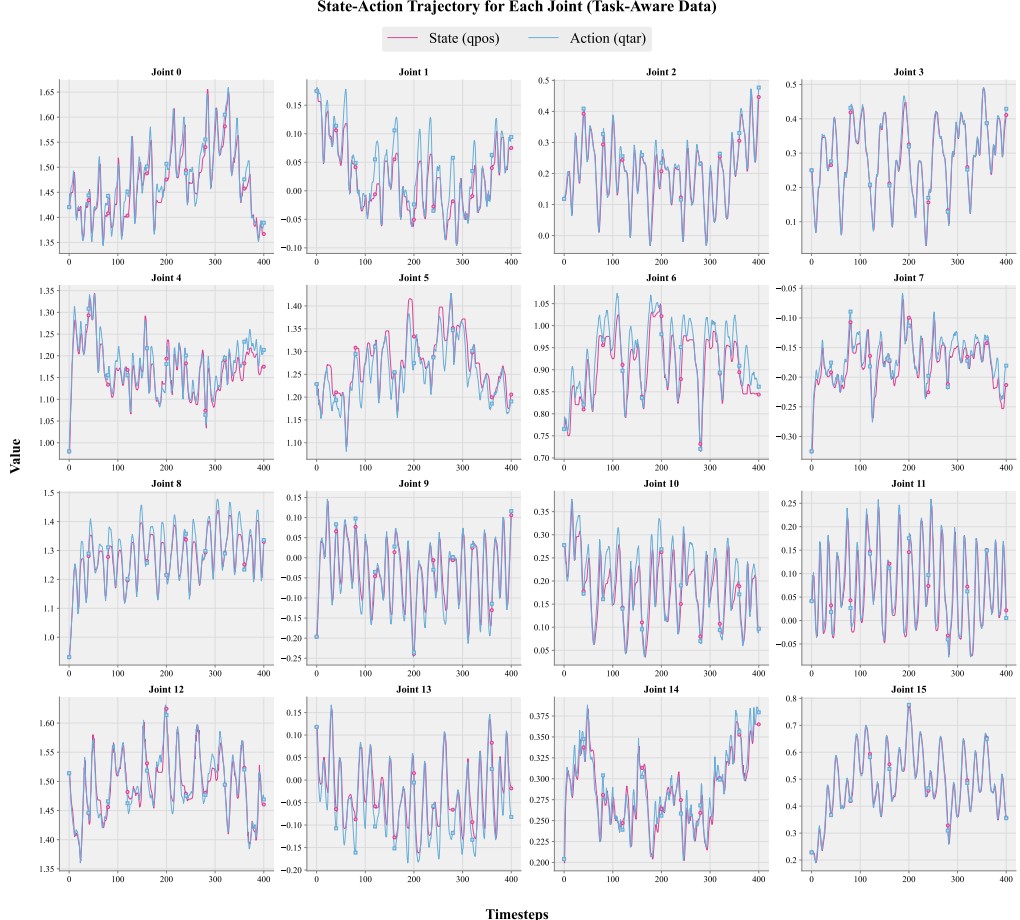

Figure 12: **Per-Joint State-Action Sequences (Task-Aware Data).**

the input and output of the function 30 with $k_1 + k_2$ unknown parameters, which contain enough information to solve for them if the history is long enough. It then indicates the reasonability of using a neural network to predict the next transition from the state-action history, considering the sufficient information contained in the input and the universal approximation ability of neural networks.

## A.4 RATIONALITY OF JOINT-WISE DYNAMICS MODELING (PART II)

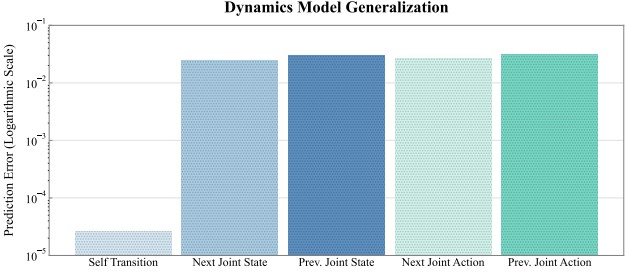

Figure 13: **Predicting via Single Joint State-Action History (Generalization Error).**

In the previous section, we demonstrated that the state-action history of a single joint is sufficient to predict its own next transition. This indicates that the information contained in the single joint

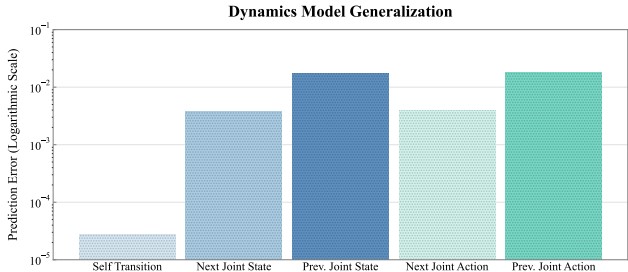

Figure 14: **Predicting via Single Joint State-Action History (In-Distribution Validation Error).**
m

state action history is at least sufficient to account for the evolution of low-dimensional effective variables over a short time window, *i.e.,* $\mathbf{H}_t^{\text{eff}}$ and $\mathbf{G}_t^{\text{eff}}$. However, this is not enough to demonstrate that a model that learns to predict from the history *would not* implicitly learn to predict the original high-dimensional complex forces like inter-joint coupling to predict the transition. Demonstrating this point is important since if the single joint state-action history contains sufficient information to predict a higher-ordered system's states, learning from the single joint history is thus not an effective dimensionality reduction and would hamper the generalization ability as the model would still overfit to the system's high-variance influences.

We demonstrate via experiments aiming to say that the state action history of a specific joint does not contain sufficient information to predict other joints' information.

We train the joint-wise dynamics model to predict the following information 1) its next joint's current state, 2) the previous joint's current state, 3) the next joint's action (positional target), and 4) the previous joint's action (positional target). We then compare their prediction and generalization error with that achieved by the joint-wise dynamics model (predicting itself's next state) for analysis.

We train all models from scratch using real-world transition data without pretraining using simulation data. Real-world transition data is the same as that we use in the ablation study. As shown in Figure 14 and 13, utilizing a single joint state-action history to predict statistics of other joints cannot even achieve reasonable performance in the original distribution. The generalization error is three order larger than that achieved by using a single joint state-action history to predict its own next transition. As for the in-distribution validation error (which is achieved on the in-distribution validation set and is close to the training error), predicting neighboring joints' states achieves a slightly better performance than predicting their actions. However, this is still far from a reasonable prediction, with the error two-ordered larger than that achieved in predicting the joint's own transition.

These experiments demonstrate that even predicting the easiest information that results in the complex coupling (*i.e.,* neighboring joints' state and action) via a single joint's state-action history is not feasible. This further indicates that a single joint's state-action history does not contain enough information to account for the complex influence factors in the original high-dimensional space. Since such information is sufficient to predict the joint's own transition, a reasonable assumption is that the network tends to leverage such net effects implicitly from the history for predicting the dynamics evolution.

**What does the joint-wise neural dynamics model implicitly capture?** Analyses and experiments in Secs.A.3 and A.4 clarify what is and is not predictable from a single joint's state–action history. Our comprehensive experiments (Sec.4.2) show that joint-wise neural dynamics are expressive, sample-efficient, and generalize well. The analysis in Sec.A.3 indicates that a single joint's history contains sufficient information to approximate its next transition, whereas Sec.A.4 shows it cannot recover each underlying coupling effect. Thus, the per-joint history captures low-dimensional net effects while avoiding overfitting to system-wide variations. This factorized, per-joint modeling transfers across changes in whole-hand interaction because the distribution of net effects is comparatively more stable than that of full-system interactions.

**Limitations of joint-wise neural dynamics mode.** As shown in Fig. 6, the joint-wise dynamics model performs slightly worse than the whole hand dynamics model in the in-domain test setting

under the multi-task high data regime. The optimization speed is also a limitation, as iterating over all joints takes time, resulting in a longer training time.

## A.5 COMPARISONS OF DATA DISTRIBUTIONS BETWEEN COLLECTED TRAJECTORIES AND ROTATION TRAJECTORIES

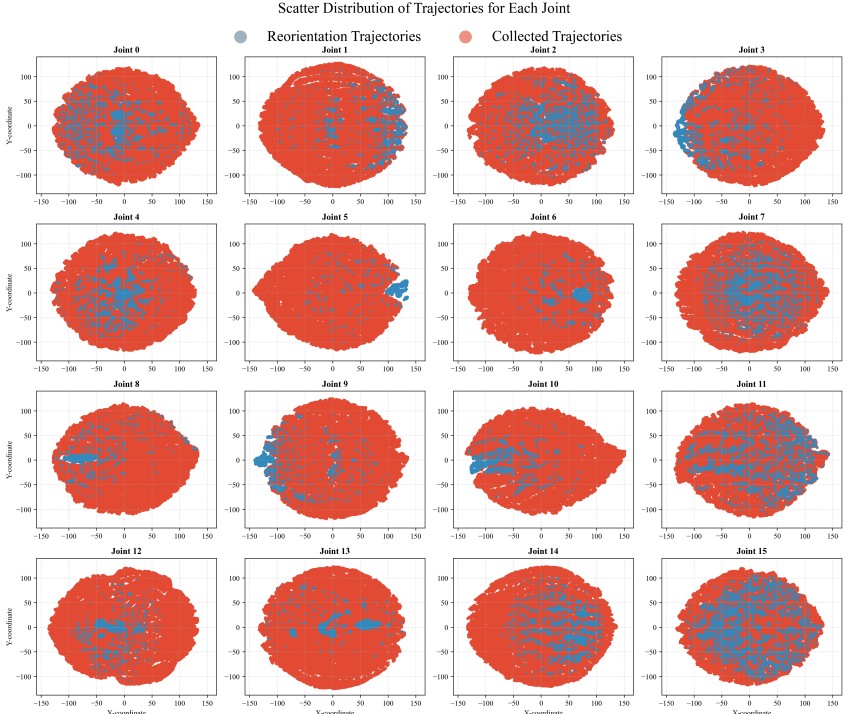

Figure 15: **Per-Joint Distribution**

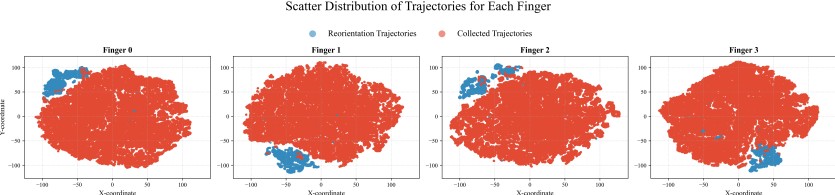

Figure 16: **Per-Finger Distribution**

Figure 15, 16, and 17 summarize the per-joint, per-finger, and whole hand data distribution. It compares trajectories collected by our autonomous data collection strategy and task-relevant rotation trajectories. The task relevant trajectories are 20 cube-rotation trajectories (∼8,000 data points in total) collected using under the "Thumb Up" wrist orientation. Per-Joint state-action trajectories can well cover the distribution of task-aware rotation trajectories. However, per-finger and whole hand distributions exhibit a huge discrepancy.

We additionally compare the data distribution between trajectories collected via rolling out the policy on a small object and that collected autonomously in "Chaos Box" to further demonstrate the broad coverage of per-joint transition data collected in "Chaos Box". We use T-SNE to visualize the per-joint I/O history space achieved by the autonomously collected data and the real-world rollouts on the small corn object. Results show that the joint I/O history space formed by the autonomously collected data can still offer a good coverage for that of policy rollouts on the small object (Fig. 18). In comparison, their whole-hand I/O history distributions differ a lot from each other (Fig. 19).

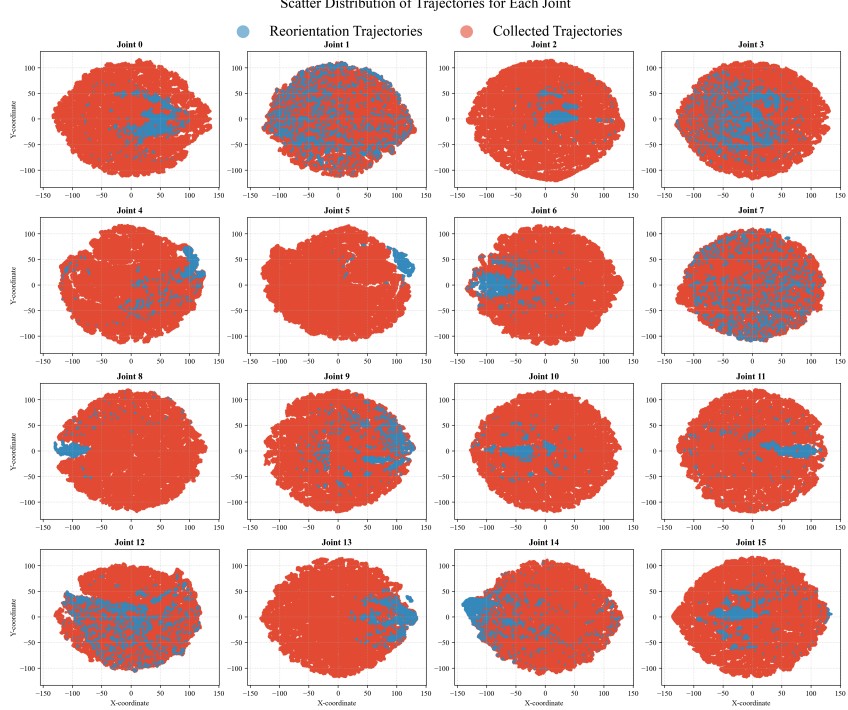

Figure 17: **Whole Hand Distribution**

Figure 18: **Per-Joint Distribution**. Real-world data collected by rolling out the policy on a small object, *i.e.,* corn, for -z axis rotation.

# B   ADDITIONAL EXPERIMENTS AND ANALYSIS

## B.1   TRAINING PERFORMANCE

AnyRotate (Yang et al., 2024) improves over prior works regarding the generality to diverse writing orientations and various rotation axes. However, they only considered regular objects. Achieving such general rotation ability for complex objects poses additional challenges, even in the policy training aspect. In our experiments, we find that prior RL designs for rotation policies (Qi et al., 2022; 2023; Yang et al., 2024), where only proprioceptions and object and system parameters-related privileged information, such as masses, are considered in the observation, may let the training get stuck in a local optimum. Thus, we include more privileged information into the observation, followed by observation space distillation for sim-to-real (Sec. 3.1). We compare with our re-implemented

## Scatter Distribution of Trajectories for Whole Hand

● Reorientation Trajectories ● Collected Trajectories

**Whole Hand**

Figure 19: **Whole Hand Distribution**. Real-world data collected by rolling out the policy on a small object, *i.e.,* corn, for -z axis rotation.

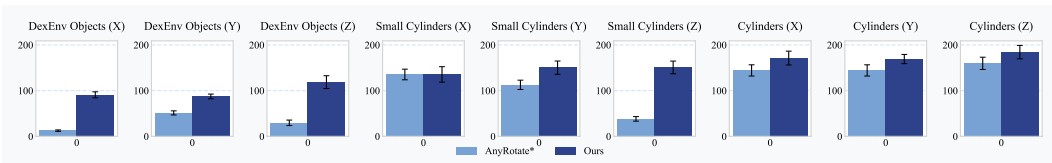

Figure 20: **Training Performance.** Comparison of the final training performance (total reward) achieved by our method and the re-implemented AnyRotate on different training sets. "DexEnv Objects" denote an irregular training object category.

AayRotate to demonstrate this design's superiority. Our method shows noticeably better training performance over AnyRotate (Fig. 20), especially on challenging object sets, *i.e.,* "DexEnv Objects" with irregular and complex geometries and "Small Cylinders" featured by small sizes, where stable finger gaiting cannot emerge in AnyRotate. We also re-implement RotatIt (Qi et al., 2023) in the Hora (Qi et al., 2022) codebase, but find that it can hardly achieve satisfactory results in the most basic cylinder object set. We also adapt Hora to the down-facing hand scenario but find it cannot work.

### B.2    ADDITIONAL REAL WORLD RESULTS

Fig. 22 and 23 provide more real-world qualitative results. See more results and videos in our website.

**Recovery Behaviour**. Fig. 24 illustrates two examples of the recovery behavior exhibited by our final policy.

**Effectiveness in In-Hand Translation.** We conduct preliminary validations on the in-hand translation task to confirm our method's effectiveness beyond the specific in-hand rotation task. For instance, in the downward-facing configuration, the NDM enables the leap hand to translate a 3cm x 3cm x 10cm cuboid from being initially grasped by the thumb, middle, and pinky fingers be grasped by the thumb, middle, and index finger (Fig. 25). Without NDM, for such a thin object, directly transferring the corresponding in-hand-translation policy fails to do this.

This study aims to demonstrate the cross-task effectiveness of our sim-to-real methodology. Conducting a more comprehensive evaluation and developing a multi-task benchmark are high-priority directions for our future work.

### B.3    CASE STUDY ON THE EFFECTIVENESS OF OUR SIM-TO-REAL METHOD

As shown in Table 4 and 5, our design on learning neural dynamics and residual policy for sim-to-real can achieve notably superior results than the policy without sim-to-real design. Below, we

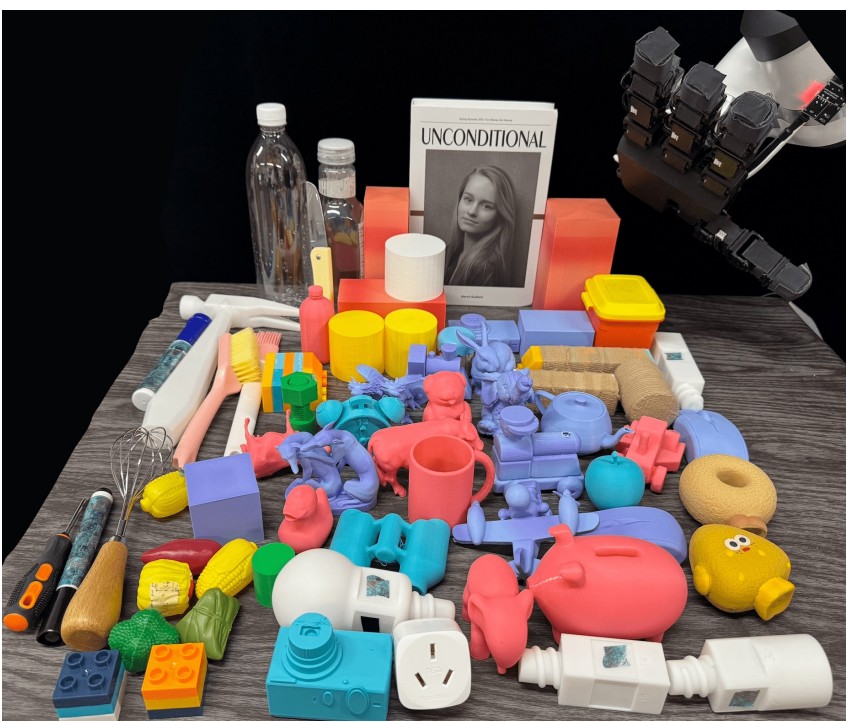

Figure 21: **Evaluated Objects in the Real World.**

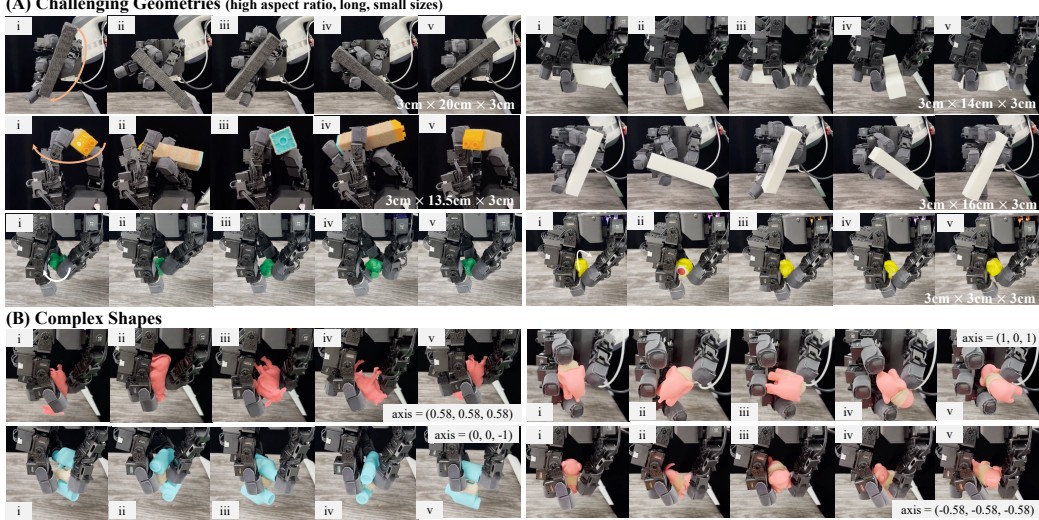

Figure 22: **Real World Results.** Rotating challenging objects in the air. See more and videos in our website.

introduce several empirical observations and case studies on our sim-to-real method. Notably, the residual policy can effectively improve the performance on challenging shapes, helping us solve previously unsolvable rotation tasks, and also enhancing the stability of the rotation (Table 7).

**Rotating Challenging Objects.** One of the important features of the residual policy is enabling us to rotate challenging objects with high aspect ratios or difficult object-to-hand ratios. For instance, without the sim-to-real strategy, the policy can only rotate the long "Lego" leg (width=3cm, lenght=13.5cm) for at most 180 degrees. However, introducing the residual policy can help us rotate it for (almost) a complete circle (demonstrated in Figure 22 and videos in our website). For longer (14cm, 16cm) objects, the base policy can only rotate the object for at most 90 degrees, while incorporating our sim-to-real method can let it rotate the object for almost a full circle. Similar ob-

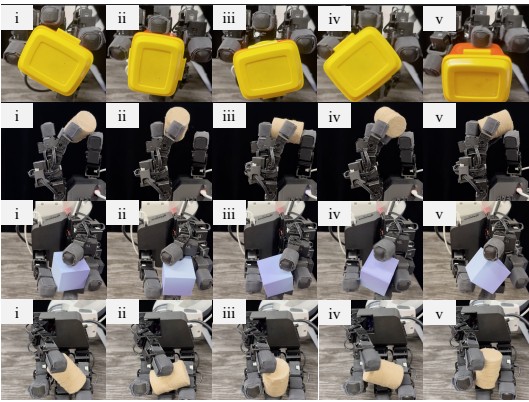

Figure 23: **Diverse Wrist Orientations.**

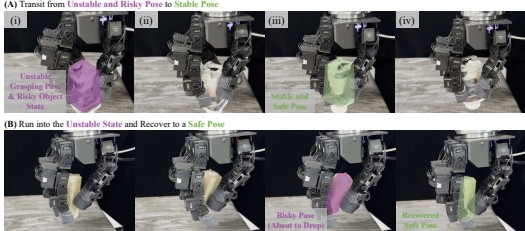

Figure 24: **Recovery Behaviour.**

servations for the "book" object, which is 16cm long. For complex shapes (*e.g.,* Cow, Truck, Bear, Dragon), the base policy can rotate for 1/4, 1/2, or at most 3/4 circles. While our sim-to-real strategy can improve it to a full circle rotation. Similar observations for small objects (*e.g.,* "Zongzi", Broccoli). For in-hand rotation, there is typically no standard definition of "success". If one is required, a reasonable criterion could be completing at least one full rotation of the object. Failing to achieve a full 360 degree rotation indicates that the policy cannot successfully navigate the fingers around the object's surface and would likely fail on challenging features (*e.g.*, the four legs of the cow). For such difficult shapes, the role of our sim-to-real strategy can be seen as elevating performance from failure to success.

**Improving the Stability.** Apart from rotating, equipping us with the ability to rotate challenging objects, the residual policy can effectively make the rotation more stable and thus help us achieve long-term rotation. A representative example is rotating the 3cm×3cm×10cm cuboid in this vertical pose. When dealing with such thin objects, the policy would use three fingers – the thumb, middle, and pinky fingers – to rotate the object. Compared to using four fingers, this rotation gait is unstable. If we do not include the residual policy, we can rotate the object for at most 5 circles. However, including the residual policy can let us rotate the object continuously for more than 5 minutes, which corresponds to about 30 circles. Similar observations for rotating the "cube" object along the y-axis.

### B.4 FURTHER DISCUSSIONS, ANALYSIS, AND ABLATION STUDIES

**Residual Policy v.s. Direct Finetuning.** A natural alternative for adapting the base policy is direct fine-tuning. We evaluated this by fine-tuning the base policy on the learned dynamics model. We empirically find that the direct fine-tuning fails to produce good behaviour and is even unable to execute basic rotations. We attribute this to two main reasons: 1) Fine-tuning requires careful hyperparameter selection, such as using a small initial learning rate to prevent overly large updates to the base policy. 2) The learned neural dynamics do not cover the full state-action distribution (*i.e.,* is not "globally accurate"), making them unsuitable for direct policy tuning. For instance, when the

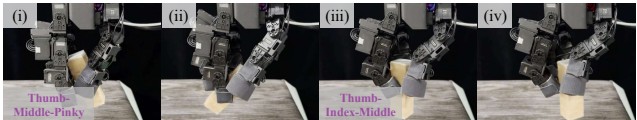

Figure 25: **Example of In-hand Translation .**

| Method | Bunny (z) | Elephant (z) | Cow (z) | Truck (z) | Bear (z) | Cuboid (V, -z) | Cuboid (H, z) | Corn (-z) | Broccoli (-z) | Cube (y) | 14-cm Cuboid (z) | 16-cm Cuboid (z) | Dargon (z) | "Zongzi" (z) |
|---|---|---|---|---|---|---|---|---|---|---|---|---|---|---|
| **Direct Transfer** | 7.33 | 6.28 | 3.67 | 4.36 | 4.19 | 31.42 | 3.67 | 10.47 | 5.76 | 19.37 | 1.57 | 1.57 | 3.14 | 3.14 |
| **DexNDM** | **8.38** | **7.07** | **6.28** | **6.81** | **6.28** | **99.48** | **6.28** | **16.76** | **10.47** | **130.90** | **6.28** | **6.28** | **6.28** | **7.85** |

Table 7: **Effectiveness of the Sim-to-Real Method on Challenging Shapes.** Comparison on Rot (in radian) achieved by the base policy w/ and w/o **DexNDM** on challenging shapes (*i.e.,* high aspect ratios, small sizes, and complex geometry). Performance tested on a down-facing hand. Symbols in parentheses indicate the rotation axis. Values are the average over three independent trials.

fine-tuned policy outputs actions outside the distribution supported by the learned dynamics, *i.e.*, $f^r(\mathbf{s}, \pi^{\text{finetune}}(\mathbf{s}, \mathbf{a}))$, the predicted state transitions are unreliable.

One potential solution is to regularize the base policy update, *e.g.,* penalizing deviations from the original base policy's outputs. This is similar in spirit to our practical residual-policy design, which uses a relatively small action scale. Compared to direct fine-tuning, this approach is easier to implement and yields stable training without complex hyperparameter tuning.

In terms of performance, adding a residual policy on top of the base policy and directly fine-tuning the base policy could produce roughly comparable results. Practically, the residual-policy approach is more flexible, easier to implement, and more stable, which is why we adopted it.

**Evaluated Objects in the Real World.** Our policy demonstrates effectiveness in rotating a wide variety of objects in the real world. Photo of real-world object gallery: Figure 21.

| Joint Index | 0 | 1 | 2 | 3 | 4 | 5 | 6 | 7 | 8 | 9 | 10 | 11 | 12 | 13 | 14 | 15 |
|---|---|---|---|---|---|---|---|---|---|---|---|---|---|---|---|---|
| Delta Action Magnitude | 0.0075 | 0.0104 | 0.0074 | 0.0043 | 0.0116 | 0.0093 | 0.0089 | 0.0061 | 0.0113 | 0.0066 | 0.0054 | 0.0059 | 0.0085 | 0.0113 | 0.0052 | 0.0047 |

Table 8: **Per-Joint Delta Action Magnitude.** Running average of per-joint delta action scale when rotating a cylinder (radius = 5.5cm, length = 5.5cm) along the z axis in the real world. Joints are arranged according to the joint order in Isaac Gym.

**Per-Joint Delta Action Value.** Table 8 summarizes the per-joint delta-action magnitudes observed when rotating a cylinder (radius 5.5 cm, length 5.5 cm) about the z-axis in real-world experiments. These values quantify the amount of compensation applied to each joint.

**Inherent Limitations of Task-Relevant Data Collection.** Collecting **task-relevant transitions with estimating object poses** suffer from the following inherent limitations: 1) Inability to be applied to small objects due to heavy occlusions; 2) Inability to estimate an accurate full pose for axis-symmetric objects like cylinders. 3) Noisy poses caused by fast movements, tracking inaccuracy, and heavy occlusions; 4) Huge time cost for the first time setup, *i.e.,* several days, and large time cost for launching the pipeline before each data collection, *i.e.,* about one minute. Besides, only successful trajectories can be kept, as the hand would then experience no load, and the object falling off would lead to a fast movement and an estimation failure. We can only roll out the policy and use clean actions without the flexibility to add noise, which may lead to task failure. As such, the diversity of the data would be restricted to objects that can be estimated and is biased towards easy geometries. Moreover, the object shape and scales used should match those used in the training. The dynamics model learning, even though we can collect a large amount of data, is relatively ill-posed if learning only from object states without the shape information, as for different objects, the same states and actions may lead to different transitions. Including the object shape in the dynamics modeling would inevitably further increase the modeling dimensionality and require an even larger amount of data to learn.

Collecting task-relevant data, even without estimating object poses, is also inherently limited to low efficiency, limited coverage, and restricted diversity since 1) data would be biased to easy objects that can be rotated well, 2) cannot add noise as it leads to the rotation failure, and 3) requires human

interventions to reset the object to the hand. According to our experiments, the average time cost is 42.86s.

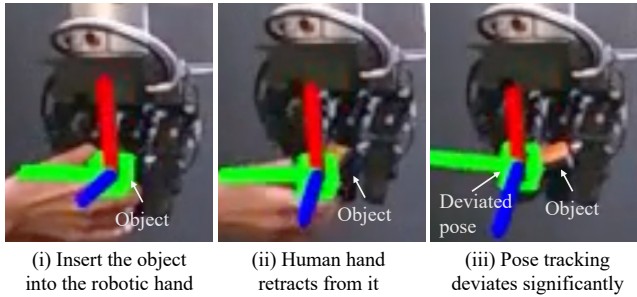

(i) Insert the object into the robotic hand    (ii) Human hand retracts from it    (iii) Pose tracking deviates significantly

Figure 26: **Pose Tracking During Manipulation for A Small Object.**

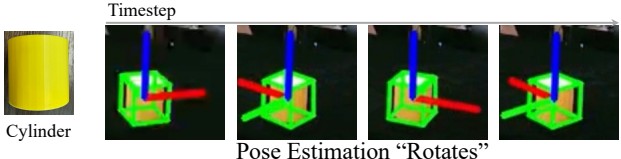

Figure 27: **Pose Tracking for Axis-Symmetric Objects.**

**Case Study on Estimating Object Poses via Foundation Pose.** Collecting real-world transitions by leveraging a vision-based estimator to track object poses is difficult, requires frequent and tedious human interventions, and is prone to yielding noisy results. For each object, we need its CAD model with exactly the same scale. Initialization steps involve capturing images via the camera and utilizing XMem (Cheng & Schwing, 2022) to get the object mask. At the beginning of each trail, we need to put the object near to the pose where we get the mask. After that, we need to move the object from the table to the robotic hand and launch the policy.

The difficulty of the data collection varies across the object geometry. For normal-sized objects, limitations primarily lie in noisy estimations, time-consuming, and human labor extensive. On average, we need 200s to collect a usable transition trajectory.

However, for small objects, it struggles to yield successful or even usable data. If we put the object initially on a table, then as we move the object up to the robotic hand, the pose tracking would fail, even if we move it very slowly. To resolve this, we hold the object by hand at a pose near to the robotic hand for initialization. After that, we need to insert it to the robotic hand for rotation. As the human hand retracts from the object, the estimated pose deviates from the object (Fig. 26).

Besides, for axis-symmetric objects, Foundation Pose cannot give stable estimations, where the pose continuously "rotates" while the object is kept still (Fig. 27). It prevents us from getting high-quality and clean pose estimations.

**Superiority of Our Autonomous Data Collection.** Compared to task-relevant data, our autonomous data collection is object-agnostic. The hand would be continuously affected by time-varying object influences during the task execution. Joint effects of all loads to each joint simulate various external influences coming from coupling effects and the object. One can also use any other objects in he data collection to expand the diversity. Besides, we can add noise to the replay actions to expand the diversity and coverage. Moreover, it is efficient and requires no human intervention.

**Inherent Limitations of Playing Base Waves to Collect Data.** To get real-world transitions, a different approach from open-loop replaying policy action rollouts and rolling out the policy is playing parameterized waves such as sine waves, square waves, and Gaussian noise (Fey et al., 2025). This strategy suffers from the following drawbacks compared to using policy data: **1)** For dexterous hands, sending signals to a single joint while keeping others still would cause self-collision, which may harm the hardware. **2)** The model, either the dynamics model in our work or the compensator

**Scaling Law w.r.t. Real World Data Quantity**

Legend:
- □ Per-Joint (Autonomous)
- □ Task-Aware w/ Obj. Pose (Fitted)
- --- Extrapolated Trend ($y \approx 0.30x^{0.21}$)
- ✖ Prediction ($x \approx 52483440$)

Y-axis: Rot (radian) (Y-Value)
X-axis: #Trajectories (X-Value, Logarithmic Scale)

52483440

Figure 28: **Performance scaling with dataset size.** We fit the curve of "Task-Awre w/ Obj. Pose" via power-law and extrapolate it to estimate the number of data required to achieve the desired result.

in UAN and ASAP, learned based on transition data obtained via playing such signals, would potentially suffer from a distribution shift when applied in the following policy finetuning or compensator training scenarios, especially when the model input contains a history. **3)** Designing the frequency and magnitude of such waves is labor-intensive and time-consuming. Thus, we adopt to use of policy rollout to obtain real-world transitions.

**Task-Relevant Data w/ Obj. Pose.** We use a 5 cm × 5 cm × 5 cm cube to collect real-world transition trajectories with object-state annotations. During data collection, we roll out the policy while rotating the object about the z-axis, and estimate its pose with FoundationPose. Because the cube is symmetric, we resolve the pose-frame ambiguity at the start of tracking by flipping the model to align with our frame convention. Each data-collection episode lasts about 200 s on average. We evaluated datasets containing 17 and 54 trajectories. Under the same real-world evaluation protocol as in our ablations, the average rotation is 0.55 and 0.70, respectively. Fitting a learning curve to these points, we estimate how many trajectories would be required to match the performance of our method with 4,000 autonomous trajectories. As shown in Figure 28, the estimate is 52,483,440 trajectories—clearly impractical. Although this extrapolation is based on a small number of data points, it highlights the data efficiency and generalization of our approach.

We attempted to train the sim-to-real baselines (ASAP and UAN) using these task-relevant, object-state–annotated data, but even the first stage—compensator training—failed to converge, and rewards showed no meaningful improvement, likely due to poor data quality.

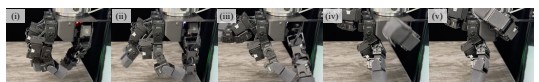

Figure 29: **Out-of-Distribution Behaviour.**

**BC v.s. DAgger.** The BC vs. DAgger issue arose in our early experiments when we tried to distill specialist single-object-category rotation oracle policies, trained with a downward-facing hand, into real-world deployable proprioception-only policies. We ultimately chose BC because, in these downward-facing rotation experiments, DAgger failed, whereas BC could reliably work well on hardware. We explain below:

- We first trained basic cylinder-rotation specialist policies with the robotic hand kept in the downward-facing direction for x-, y-, z-axis rotation, and attempted to distill them using both DAgger and BC, followed by real-world testing.

- Initially, we tried DAgger, following prior works. For cylinder-rotation policies with an up-facing hand, the distillation process could be successfully optimized (achieving total

rewards around 80–100), and the distilled policy could be successfully transferred to the real robot, rotating the cylinders with the hand facing up.

- However, when the oracle policy is trained with the hand kept in the downward-facing orientation, DAgger fails to optimize the distillation process (*i.e.,* rewards around 10–30; rotations along the z-axis may reach 50). The transferred policy immediately fails in the real world. For visualizations of a typical out-of-distribution behavior, see the section (Out-of-Distribution Behaviour) on the website and Fig. 29 for a visualization.

- We then switched to BC and observed that it works well in this setting. Here, "works well on hardware" means that the transferred cylinder-rotation BC policy with the hand facing down avoids the out-of-distribution failures seen with DAgger and can perform basic rotations reliably.

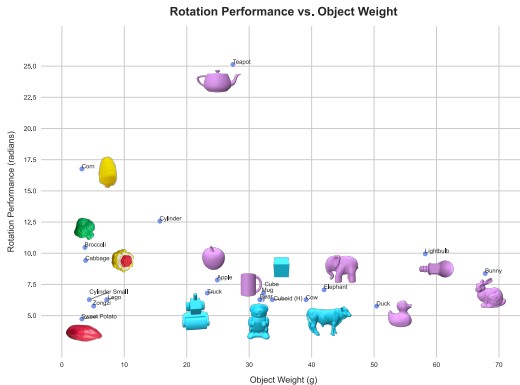

Figure 30: **Performance w.r.t. Object Physical Weight.**

|  | Cube | Cyliner | Apple | Cuboid (H) | Lightbulb | Bear | Truck | Cow | Bunny | Elephant | Mouse | "Zongzi" | Cabbage | Cylinder Small | Corn | Broccoli | Lego | Sweet Potato | Teapot | Mug | Duck |
|---|---|---|---|---|---|---|---|---|---|---|---|---|---|---|---|---|---|---|---|---|---|
| Weight (g) | 32.3 | 15.7 | 24.9 | 33.7 | 58.2 | 32.1 | 23.3 | 39.1 | 67.8 | 42.0 | 17.93 | 5.1 | 3.8 | 4.4 | 3.2 | 3.7 | 7.2 | 3.2 | 27.4 | 31.7 | 50.4 |

Table 9: **Object Weight** measured in the real world.

**Performance w.r.t. Object Weight.** The plot of performance w.r.t. object weight is shown in Fig. 30. We measure the object weight in the real world (Table 9).

Because the objects vary significantly in shape, size, and geometric complexity, the plot does not exhibit a clear trend showing how performance changes as weight increases.

| Weight (g) | 15.7 | 33.2 | 39.2 | 49.4 | 62.0 |
|---|---|---|---|---|---|
| Rot (radian) | 12.57 | 12.57 | 12.57 | 9.42 | 7.85 |

Table 10: **Rot (radian) v.s. Weight** for cylinder (5.5cm $\times$ 5.5cm $\times$ 5.5cm) with z-axis rotation.

To conduct a more controlled study and isolate the effect of object weight, we designed an experiment using cylinders of identical geometry but different masses as test objects, and evaluated the performance on z-axis rotation. Results are summarized in Table 10 and Fig. 31. During training, the upper limit of the object weight randomization range is 50.0g. When the test object weight is below 40 g, we do not observe a clear difference in rotation performance. This may seem unintuitive, as lighter objects might appear easier to rotate. However, in practice, for relatively light objects, the robotic hand often tends to continuously translate the object upward during the rotation, causing it to become "stuck" between several fingers, which can actually hinder rotation. For heavier objects—for example, the 62 g case—a common cause of failure is that the robot hand cannot maintain a stable grasp during the rotation, leading to the object being dropped.

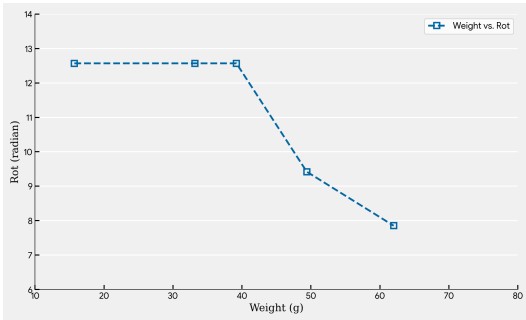

Figure 31: **Performance w.r.t. Object Physical Weight.** Rotate the cylinder object (5.5cm × 5.5cm × 5.5cm) along the z-axis with the hand facing down.

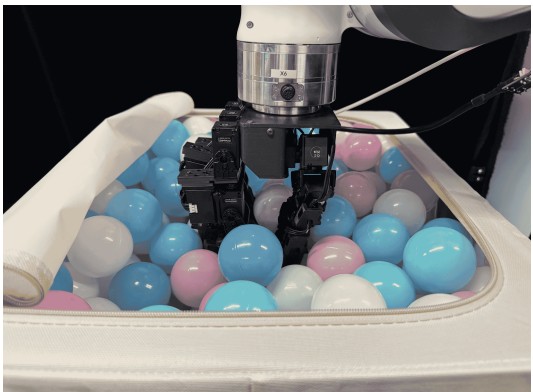

Figure 32: **"Half Load" Chaos Box.** Instead of totally sinking into balls, the hand only touches the surface of the balls in the box using fingertips.

**Sensitivity to Noise and Load Distribution.** To explore the influence of the magnitude of the injected noise, we design two additional settings and replay ation sequences with noise magnitude 0.005 and 0.02, *i.e.,* $\sigma = 0.005, 0.02$. and compare the results with those acheived using default value, *i.e.,* $\sigma = 0.01$.

To explore the influence of the distribution of random loads, we add an experiment where the hand only touches the surface of balls in the box using fingertips (denoted as "half loads" (Fig. 32), while the original setting where the hand is sunk into the balls is represented as "full loads")

Other settings, including the replayed trajectories and the distribution of balls in the box, are kept the same as the settings in our ablation study.

|  | $\sigma = 0.01$, full loads | $\sigma = 0.005$, full loads | $\sigma = 0.02$, full loads | $\sigma = 0.01$, half loads |
|---|---|---|---|---|
| Generalization Ability (Prediction Error ) | $1.21 \times 10^{-5}$ | $1.89 \times 10^{-5}$ | $1.47 \times 10^{-5}$ | $3.81 \times 10^{-5}$ |
| Real-World Performance (Rot (radian)) | $12.43 \pm 0.34$ | $7.51 \pm 1.03$ | $10.58 \pm 0.92$ | $6.28 \pm 0.67$ |

Table 11: **Sensitivity to Noise and Load Distribution.** Generalization performance and the final real world effectiveness w.r.t. magnitude of the noise injected to the replayed actions and the load distribution.

Same as we evaluate in the ablation study, we evaluate the generalization ability of the resulting neural dynamics model and the final real world performance achieved by the residual policy with the base policy. Results are summarized in Table 11.

Regarding the noise magnitude, increasing the noise has only a minor impact on both the generalization ability of the neural dynamics model and the final policy performance. It does slightly affect the model's behavior, likely because expanding the randomization range broadens the data distribution while reducing the amount of data available at some noise levels, which can make learning

more difficult. *Reducing the noise magnitude*, however, has a more noticeable negative effect on the learning of the neural dynamics model, its generalization ability, and the final policy performance. When the noise magnitude is too small, the data distribution becomes narrower, making the model less robust and reducing the degree of extrapolation required for training the residual policy. This in turn degrades overall performance.

Regarding the load distribution, narrowing the load distribution—for example, when only the fingertips are able to contact the small sphere—reduces the effective disturbances experienced by other joints. This weakens the robustness of the neural dynamics model and the extrapolation ability needed for training the residual policy, ultimately harming generalization and sim-to-real performance.

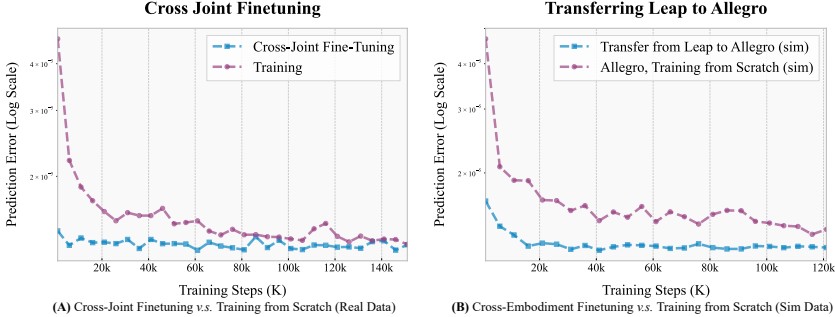

Figure 33: **Cross-Joint and Cross-Embodiment Transfer.** (A) Comparisons of the training loss in the cross-joint transfer setting (fine-tuning v.s. training from scratch); (B) Comparisons of the training loss in the cross-embodiment transfer setting (fine-tuning v.s. training from scratch).

**Cross-Joint and Cross-Embodiment Transfer.** We test the cross-joint and cross-embodiment transfer ability of our joint-wise neural dynamics model. We conduct the following experiments:

- **Cross-joint transfer on the same embodiment** (Leap Hand, real-world collected data), where we transfer the dynamics model of joint $i + 1 \pmod{N_J}$ to joint $i$.
- **Cross-embodiment transfer** (Leap Hand $\rightarrow$ Allegro Hand, simulator rollout data).

In both scenarios, we observe that:

- the neural dynamics model initialized from a different joint or from a different embodiment provides a good starting point; and
- achieving the same prediction loss requires significantly less fine-tuning time compared to training from scratch.

More specifically, with such initialization, for cross-joint transfer:

- The initial training loss is over **five times lower** than that obtained when training from scratch;
- Fine-tuning a pre-trained network requires approximately **51× less time** to reach the final prediction loss of the training from scratch.

For cross-embodiment transfer in the simulator:

- The initial training loss is over **four times lower** than that obtained when training from scratch;
- Fine-tuning a pre-trained network requires approximately **11× less time** to reach the final prediction loss of the training from scratch.

Fig. 33 plots the training loss curves in such two settings.

B.5    RUNTIME PERFORMANCE ANALYSIS

While the joint-wise neural dynamics model enjoys generalization ability and sample efficiency, it would require a longer training time. Compared to whole-hand neural dynamics model, both the neural dynamics training process and the residual policy learning would take more time. We include the detailed training process of the joint-wise neural dynamics model and the residual policy in the following text.

- Joint-wise dynamics model is trained in a per-joint manner. All 16 joint-wise models are trained in parallel. To be more specific:
  - Given a batch of input history and output state of the whole hand, *i.e.,* $\{\mathbf{h}, \mathbf{s}\}$, we split for each joint, *i.e.,* $\{(\mathbf{h}_i, \mathbf{s}_i)\}_{i=1}^{16}$.
  - For each joint $i$, the history is passed to the model to get the corresponding prediction $\mathbf{s}_i'$.
  - After that the full prediction is obtained by concatenating them together, *i.e.,* $\mathbf{s}' = \{\mathbf{s}_i'\}_{i=1}^{16}$. And the loss is computed between $\mathbf{s}'$ and $\mathbf{s}$.
  - Compared to the whole hand dynamics model, the extra time cost come from forwarding through 16 joint networks.
  - **Comparisons of the time consumption.** In practice, training the neural dynamics in a joint-wise manner roughly takes 4 6 times longer than training the whole hand dynamics model.

- The residual policy is a full-hand model, *i.e.,* accepting the I/O history and the base action of the hand and outputting the residual action for all joints in the hand. Since we need to query the learned neural dynamics model, compared to whole hand dynamics model, the extra time cost of the joint-wise dynamics model come from forwarding through 16 joint networks.
  - **Comparisons of the time consumption.** In practice, training the residual policy using a joint-wise neural dynamics model is roughly 6.2 times slower than training it using the whole hand dynamics model.

| # Trajectories | 4,000 | 6,000 | 8,000 | 24,000 |
|---|---|---|---|---|
| Joint-Wise | 3.44 hrs | 5.30 hrs | 6.83 hrs | 18.72 hrs |
| Whole Hand | 0.61 hrs | 0.88 hrs | 1.47 hrs | 4.67 hrs |

Table 12: **Training Time Comparison** between joint-wise neural dynamics model and the whole hand dynamics model using training datasets with different sizes.

| # Trajectories | 937,275 | 1,792,431 | 2,073,973 |
|---|---|---|---|
| Joint-Wise | 6.2 hrs | 13.68 hrs | 15.25 hrs |
| Whole Hand | 3.6 hrs | 5.5 hrs | 5.8 hrs |

Table 13: **Training Time Comparison** between residual policies using the joint-wise neural dynamics and that using whole hand dynamics.

We train both the dynamics model and the residual policy in parallel on eight 23GB A10 GPUs in a Ubuntu 20.04 server. For the neural dynamics model, we vary the dataset size (#Trajectories, where each trajectory contains 400 step transitions) and train each model for 100 epochs. The training time for the joint-wise dynamics model and the whole hand dynamics model is summarized in Table 12.

Training the neural dynamics in a joint-wise manner roughly takes 4 6 times longer than training the whole hand dynamics model. However, the time cost is still acceptable. For our final model where 24,000 trajectories are leveraged to train the neural dynamics model, it takes less than 19 hours to complete.

For the residual policy, we train on the simulation rollout dataset for one epoch. We vary the dataset size (#Trajectories, where each trajectory contains 400 step transitions) and train the residual policy using the joint-wise neural dynamics model and the whole hand dynamics model, respectively. The training time for them is summarized in Table 13.

# C ADDITIONAL EXPERIMENTAL DETAILS

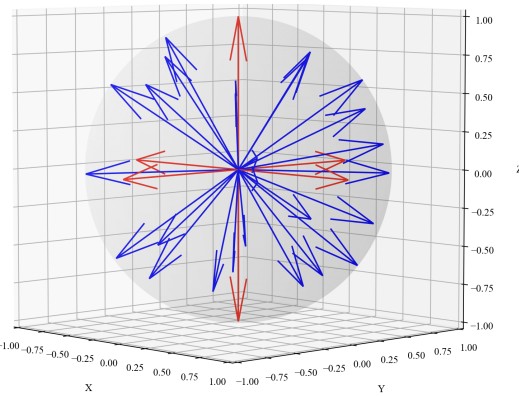

Figure 34: **General Rotation Axes.**

| Object Set | Normal-Sized Cylinders | Normal-Sized Cuboids | Long Cuboids | Small Cylinders | DexEnv Objects | ContactDB Objects (Test Set) |
|---|---|---|---|---|---|---|
| #Shapes | 9 | 9 | 4 | 9 | 120 | 26 |
| Object Minimum Extent | 0.04 | 0.064 | [0.06, 0.08] | 0.025 | [0.056, 0.115] | [0.017, 0.153] |
| Object Aspect Ratios | [1.6, 2.4] | [1.25, 1.5] | [2.5, 6.67] | [1.92, 2.56] | [1.05, 2.00] | [1.0, 11.67] |
| Object Scale | [0.70, 0.86] | [0.70, 0.86] | 0.5 | [0.5, 0.6] | [0.6, 0.7] | [0.5, 0.6] |
| Mass | [0.01, 0.05] kg | [0.01, 0.05] kg | [0.01, 0.05] kg | [0.01, 0.05] kg | [0.01, 0.05] kg | [0.01, 0.05] kg |
| Coefficient of Friction | [0.3, 3.0] | [0.3, 3.0] | [0.3, 3.0] | [0.3, 3.0] | [0.3, 3.0] | [0.3, 3.0] |
| External Disturbance | (2, 0.25) | (2, 0.25) | (2, 0.25) | (2, 0.25) | (2, 0.25) | (2, 0.25) |

Table 14: **Information and Physical Parameter Randomization Ranges** of Training Object Sets and the Test Object Set.

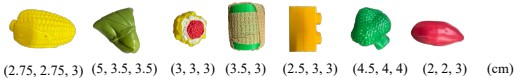

(2.75, 2.75, 3)    (5, 3.5, 3.5)    (3, 3, 3)    (3.5, 3)    (2.5, 3, 3)    (4.5, 4, 4)    (2, 2, 3)    (cm)

Figure 35: **Dimensions of Small Objects Used in Real World Experiments.**

**Datasets.** Our training objects comprise the following subsets: 1) Normal-sized cylinders from Hora (Qi et al., 2022); 2) Normal-sized cuboids from Hora (Qi et al., 2022); 3) Long cuboids; 4) Small-sized cylinders; and 5) Normal-sized complex shapes from Visual Dexterity (Chen et al., 2022) (denoted as "DexEnv Objects"). Details with scale randomization ranges are summarized in Table 14. To test the generalization performance in unseen shapes, we filter objects with an aspect ratio no larger than 2:1 from the ContactDB dataset (Taheri et al., 2020) (obtained from GRAB dataset) as our test set, resulting in 26 objects in total. The filter rule follows RotateIt (Qi et al., 2023). As we aim to test the generalization performance on shape variation in this evaluation, we do not consider high aspect ratio ones or scale them to small sizes. In the real world, we test the performance on three subsets (Fig. 5, purple objects and small objects are unseen):

- Regular objects: cube (5 cm × 5 cm × 5 cm), cylinder (radius 5.5 cm, length 5.5 cm), apple (GRAB/ContactDB apple, scaled to 0.5×), cuboid (3 cm × 10 cm × 3 cm), and light bulb ("lamp_bulb" from FurnitureBench).

- Small objects: Purchased online; vendor links are withheld to preserve anonymity during review and will be provided upon acceptance. Fig. 35 shows dimensions of those objects used in the real-world experiment.

- Normal-sized irregular objects: bear, truck, and cow from Visual Dexterity (each scaled to 0.7×); and bunny, elephant, duck, mug, teapot, and mouse from GRAB/ContactDB (each scaled to 0.5×).

**Policy Optimization.** We use PPO for policy optimization. Training environments are 30,000 for cylinders and cuboids, while 50,000 for long cuboids, small cylinders, and "DexEnv Objects". We randomly sample a wrist pose and a target rotation axis at each environment reset.

**General Rotation Axes.** To construct the general rotation axis set, we generate 32 axes evenly distributed in SO(3). Removing six principal axes, $\pm x$, $\pm y$, and $\pm z$, we get the general rotation axis set. Figure 34 provides a visualization of all 32 evenly distributed rotation axes.

**Generalist Training via Behaviour Cloning.** To obtain the dataset to train the generalist policy, we roll out each oracle policy in the simulation to construct the dataset. Only transition trajectories that would not terminate in the full 400 steps would be saved in the dataset. We set the maximum number of tested environments to 1,500,000. In each step, the hand joint states, positional targets, object states, rotation axis, and the hand wrist orientation would be saved. Numbers of trajectories collected by each object category are summarized in Table 15. The number of successful rollouts could reflect the difficulty of different training object sets. Among all five object sets, regular cylinders and cuboids construct the easiest rotation tasks. Small cylinders introduces additional challenges due to its small scales. Complexity in the geometry further increases the difficulty. Rotating long objects with large aspect ratios is the most difficult task, which yields the smallest transition dataset.

| Object Set | Cylinders | Cuboids | Long Cuboids | Small Cylinders | DexEnv Objects |
|---|---|---|---|---|---|
| **# Transitions** | 1,333,282 | 1,282,973 | 235,413 | 743,543 | 681,199 |

Table 15: **The Number of Collected Transition Trajectories in Simulation.**

**Metrics (detailed version).** We evaluate using RotateIt metrics (Qi et al., 2023) in simulation and the real world, plus a goal-oriented success metric: *Time-to-Fall (TTF)*—duration until the object drops; in simulation, episodes are capped at 400 steps (20s) and TTF is normalized by 20s, while in the real world we report raw time; *Rotation Reward (RotR)*—episode sum of $\boldsymbol{\omega} \cdot \mathbf{k}$ (simulation only); *Rotation Penalty (RotP)*—per-step average $\boldsymbol{\omega} \times \mathbf{k}$ (simulation only); and *Radians Rotated (Rot)*—total radians rotated in the real world, measured from videos. We also report *Goal-Oriented Success (GO Succ.)* following Visual Dexterity (simulation only): we sample a random goal pose, set the target axis to the relative rotation axis, and count success if the final orientation is within $0.1\pi$ of the goal.

**Automatic System Identification.** In addition to training neural dynamics models and the delta action model to bridge the sim-to-real gap, we would align the dynamics between the simulator and the real world by performing an automatic system identification process at the beginning. The process involves the following steps: 1) Training probing rotation skills in the simulator using the default PD gains and link configurations in the URDF. 2) Rollout probing skills in the simulator for multiple state-action trajectories (denoted as "probing trajectories"). Replay probing trajectories on the real robot. 3) Collect the resulting state and action trajectories. 4) Launch multiple parallel environments in the simulator, each with different system parameters; 5) Replay probing action trajectories to get resulting state trajectories. 6) Select parameters of the environment whose resulting state trajectories are the most similar to those in the real world as the identified system parameters. We identify PD gains and the mass of each link. Identified values are summarized in Table 16 and 17.

| Joint Index | 0 | 1 | 2 | 3 | 4 | 5 | 6 | 7 | 8 | 9 | 10 | 11 | 12 | 13 | 14 | 15 |
|---|---|---|---|---|---|---|---|---|---|---|---|---|---|---|---|---|
| P Gain | 3.52 | 1.78 | 2.84 | 2.30 | 1.94 | 2.18 | 2.55 | 2.01 | 2.26 | 2.30 | 3.76 | 4.64 | 1.86 | 3.44 | 4.82 | 1.53 |
| D Gain | 0.194 | 0.106 | 0.091 | 0.195 | 0.199 | 0.192 | 0.149 | 0.050 | 0.088 | 0.135 | 0.027 | 0.081 | 0.123 | 0.042 | 0.082 | 0.068 |

Table 16: **Identified PD Gains.** Per-Joint PD Gains identified by the automatic system identification process. Joints are arranged according to the joint order in Isaac Gym.

**Domain Randomization.** We apply domain randomization during training. We also randomize the physical parameters during the test in the simulator. The randomization ranges of each object set are summarized in Table 14. Following previous works (Qi et al., 2022; 2023), we apply a random

| Link Index | 0 | 1 | 2 | 3 | 4 | 5 | 6 | 7 | 8 | 9 | 10 |
|---|---|---|---|---|---|---|---|---|---|---|---|
| Mass (kg) | $1.00 \times 10^{-7}$ | $2.57 \times 10^{-1}$ | $2.41 \times 10^{-2}$ | $1.90 \times 10^{-2}$ | $2.79 \times 10^{-2}$ | $1.05 \times 10^{-2}$ | $1.00 \times 10^{-7}$ | $4.68 \times 10^{-2}$ | $3.00 \times 10^{-3}$ | $3.65 \times 10^{-2}$ | $5.38 \times 10^{-2}$ |
| Link Index | 11 | 12 | 13 | 14 | 15 | 16 | 17 | 18 | 19 | 20 | 21 |
| Mass (kg) | $1.00 \times 10^{-7}$ | $3.12 \times 10^{-2}$ | $2.63 \times 10^{-2}$ | $2.11 \times 10^{-2}$ | $1.63 \times 10^{-2}$ | $1.00 \times 10^{-7}$ | $5.03 \times 10^{-2}$ | $3.43 \times 10^{-2}$ | $4.76 \times 10^{-2}$ | $2.23 \times 10^{-2}$ | $1.00 \times 10^{-7}$ |

Table 17: **Identified Link Mass.** Per-Link mass identified by the automatic system identification process. Links are arranged according to IsaacGym's link order.

disturbance force to the object. The force scale is 2m, where $m$ is the object mass. We also resample the force at each timestep with the probability 0.25. We add a noise sampled from the distribution $\mathcal{U}(0, 0.005)$ to the joint positions to increase the robustness.

**Baselines (detailed version).** We compare our method against both previous in-hand rotation/reorientation works and prior neural-based sim-to-real works. We compare with two strong in-hand rotation/reorientation works, Visual Dexterity (Chen et al., 2022) and AnyRotate (Yang et al., 2024). The experimental setup of AnyRotate is the most similar to ours. It demonstrates multi-axis object rotation under various wrist orientations. However, its code is not publicly available, and the method requires tactile information. We re-implemented their environment setup and training pipeline in IsaacGym based on the paper's description. We've tried our best to set up a fair comparison with it in the real world. Unfortunately, faithfully replicating their tactile sensor model and sim-to-real methodology from the paper alone is difficult. We find that discarding the tactile information in its second stage training can hardly yield a policy with even basic rotation capabilities in the real world. Thus, a direct real-world comparison was not possible. Instead, we demonstrate our method's superior performance by evaluating it on the same challenging object shapes used in their experiments. For Visual Dexterity, the open-sourced code is designed for the D'Claw hand, which is much large than and quite morphologically different from anthropomorphic hands like the Allegro or LEAP. Despite our extensive efforts to adapt their code to the LEAP hand, the policy failed to achieve reasonable performance in simulation on a basic cylinder shape, even after 1.5 days of training. Thus, a direct comparison was infeasible. We therefore compare our method's performance with the quantitative results reported in their paper and the qualitative results shown in their website.

We also compare with prior sim-to-real methods designed for robotic arms and legged robots, namely UAN (Unsupervised Actuator Net) and ASAP. The core of both UAN and ASAP is similar, which lies in collecting real-world transition data for actuators, training neural compensators to bridge the dynamics gap between the simulator and the real world, followed by tuning/training the task policy based on the learned neural compensator. The main differences lie in two aspects, including data collection and model design. ASAP rollouts tracking policies and locomotion policies in the real world for collecting real-world transitions, while UAN avoids using policy data by playing sine waves, square waves, and Gaussian noises to prevent overfitting. UAN uses a shared network for every actuator while ASAP trains a full-body compensator (four ankle joints for sim-to-real). As discussed before (Sec. 3.3), neither including the object into the system modeling nor replicating object influence in the simulator is possible. Thus, we collect 24,000 real-world free-hand replay trajectories to train their corresponding compensators. To compare UAN, we employ their real-world collection strategy and train a shared compensator for each joint in the hand. To compare ASAP, we replay the policy rollouts and train a compensator for each finger in the sim-to-real comparison, mirroring their four ankle joints sim-to-real setting. In sim-to-sim, we train a compensator for the whole hand and the object.

**Comparisons to AnyRotate (detailed version).** We compare our real-world performance against reported values in AnyRotate. As they did not provide links to obtain their real-world test objects, we test our model on four of its tested objects that are easy to replicate, including "Tin Cylinder", Cube, "Gum Box" and "Container" (see details below). While the remaining plastic vegetable models and the "Rubber Toy" are not reproducible according to the object size information provided in their Table 10. According to its experiments, objects with sharp edges are more difficult to rotate compared to plastic vegetable models (their performance on "Tin Cylinder", "Gum Box", and "Container" is the worst regarding the number of rotations and survival time among all of its tested objects as shown in its Table 12 and 13). We test the performance on three test rotation axes from AnyRotate in the rotation axis test setting. We also employ the same rotation axis setting and the hand orientation setting to AnyRotate in the hand orientation test setting. We conduct three

independent experiments and present the average and deviations across the three trials in the Table 2. As shown, we can outperform AnyRotate by a large margin.

Besides, as demonstrated, our policy can rotate a wide range of objects with diverse aspect ratios and various object-to-hand ratios. Rotating some of them, such as the long Lego leg and animal shapes, requires quite sophisticated finger gaiting. However, AnyRotate only demonstrates the ability of rotating normal sized objects with relatively flat surfaces using conservative behaviours. As stated in their paper, they would encounter difficulties when rotating objects with sharp edges. Besides, the smallest objects that they have demonstrated the effectiveness are the "Rubber Toy" (8cm × 5.3cm × 4.8cm ), "Tin Cylinder" (4.5 × 4.5cm × 6.3cm), and "Cube" (5.1cm × 5.1 cm × 5.1cm). However we can deal with much smaller objects like vegetable models with sizes 3cm × 3cm ×2.5cm, 3cm × 2.75cm × 2.75cm, and 3cm × 2cm × 2.1cm. Moreover, the most challenging aspect ratios of their objects is 1.67 (Rubber Toy), while we can handle objects with challenging aspect ratios such as Lego leg (4.5), Book (5.3), and long cuboid (3.33). Such comparisons further demonstrate the superiority of our method in solving difficult in-hand rotation problems.

**Details w.r.t. Our Replicated Objects from AnyRotate.** We replicated their four test objects as follows:

- **Cube**: We 3D-printed a cube to the specified dimensions of 5.1cm × 5.1cm × 5.1cm.
- **Container**: We buy a commercially available product that precisely matches the container used in their experiment. We removed the labels from the container to maintain regional anonymity.
- **Tin Cylinder**: We 3D-printed a cylinder with the specified 4.5cm radius and 6.3cm length.
- **Gum Box**: We identified a discrepancy in the documented dimensions (9cm × 8cm × 7.6cm), which were identical to those of the "Container". However, figures in the original paper indicate the "Gum Box" is substantially smaller. Therefore, we estimated its dimensions from the figures to be approximately 5cm × 4cm × 8cm and 3D-printed an object of this size to serve as a proxy.

**Comparisons to Visual Dexterity (detailed version)**. Compared to prior works, visual dexterity shows improved results in rotating more complex objects with uneven surfaces and better generalization ability to unseen geometries. Conducting a direct and completely fair comparison between our method and Visual Dexterity, however, is infeasible due to the different task settings (*i.e.,* ours axis-oriented continuous rotation v.s. Visual Dexterity's goal pose-driven reorientation). Therefore, we introduce a new metric, survival rotation angles, that could be computed from qualitative results in both settings to facilitate a comparison. Specifically, it evaluates the angles the object could be rotated before it falls from the hand. This metric is friendly for Visual Dexterity since, in some settings, it has a supporting table. The object can touch the table during the rotation process. We obtain Visual Dexterity's results by carefully examining all of its demos present in all videos from its website. Its best performance and the comparisons to our results are summarized in Table 3. Though the metric is more friendly to Visual Dexterity, we can still achieve on par performance or bypass its results for all irregular objects included in its demos (see videos in our website). Specifically, we make the following observations: 1) For objects on which Visual Dexterity has demonstrated strong results, including cow, bear, and truck, where they have shown the ability to rotate the object to achieve several goals continuously without falling, we can at least achieve on-par performance with it. 2) For objects that it struggles with, including elephant, bunny, duck, teapot, and dragon, we can outperform it and achieve a much better performance regarding the survival angles. 3) We have shown superiorities in rotating objects with challenging aspect ratios (up to 5.33) and difficult object-to-hand ratios (*i.e.,* long objects like the Lego leg and small plastic vegetable models, Fig. 1). However, Visual Dexterity does not demonstrate such ability.

**Comparisons to ASAP and UAN (detailed version).** We evaluated our method against two prominent sim-to-real transfer approaches in both sim-to-sim and sim-to-real settings. Considering the difficulty in collecting real-world data with object states and the fact that their original data collection strategy does not account for the object influence, we collect 24,000 freehand trajectories in the real world by replaying policy action rollouts using the same hand wrist configurations as in our data collection strategy for data with load. After that, we train a dynamics compensator in the corresponding free-hand simulation setup. This compensator is subsequently used to fine-tune the original policy. We reward the compensator training using the hand-only training penalty:

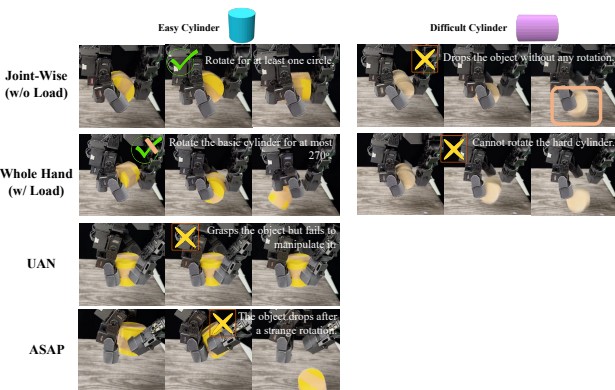

Figure 36: **Case Study on Failure Cases of Baselines (UAN and ASAP) and Ablated Versions (Joint-Wise (w/o Load) and Whole Hand (w/ Load)).**

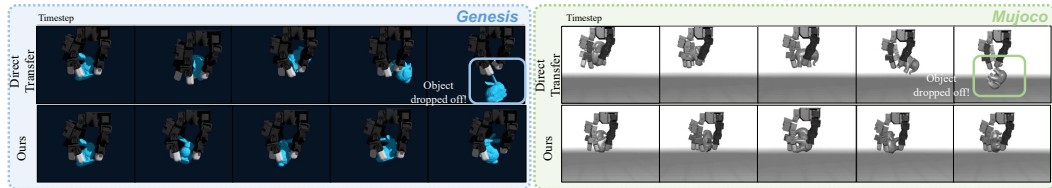

Figure 37: **Qualitative "Sim-to-Sim" Evaluation. Left:** Results in Genesis. **Right:** Results in MuJoco.

$r^{\text{compensator}} = -\|\mathbf{q}_t^{\text{ref}} - \mathbf{q}\|_2$, where $\mathbf{q}_h^{\text{ref}}$ and $\mathbf{q}_t$ are the reference joint state and the current joint state respectively. While we originally intended to conduct a comprehensive comparison in all settings covered in Table 4 and 5, we found that the policies produced by these baseline methods failed to function in the real world. They were unable to rotate the easiest cylinder object. The typical failure modes involved the robot either grasping the object firmly without movement or failing after a strange perturbation (Fig. 36 (A)). (Videos demonstrating these failures are available on our website.) Notably, the policy fine-tuning process did achieve satisfactory results. We therefore hypothesize that an OOD issue causes this: the compensator, trained only on the dynamics of a free hand, fails when the policy must handle the novel dynamics introduced by an object during the rotation. This finding underscores the critical importance of modeling object dynamics in the design of sim-to-real strategies for manipulation, which also aligns with discoveries in ablation studies (Sec. 5).

We also attempted to train the baseline sim-to-real methods (ASAP and UAN) using our collected task-relevant, object-state–annotated dataset (54 trajectories). However, the first stage—compensator training—failed to converge; the reward showed little to no improvement. We attribute this to the dataset's limited size and object state noise.

Sim-to-sim comparisons are summarized in Table 4.2.

Our compensation strategy also shows better resistance to the quality of real-world transitions. As shown in Figure 36, our ablated version "Joint-Wise (w/o Load)" trains the dynamics model via free hand replay data, whose data amount is even smaller than that used to train UAN and ASAP, can rotate the basic cylinder object for at least one circle, though its final performance cannot even surpass the base policy. However, the above two strategies totally fail in this task. Since they would use the compensator to fine-tune the base policy, their final policy's performance is quite sensitive to the quality of the learned compensator. Thus, only if the learned compensator is of very high quality and can generalize quite well can its fine-tuning achieve satisfactory results. Otherwise, the final policy may totally fail since they are learned with "wrong" dynamics. However, we compensate the base policy by using it with the learned residual policy together. With a good base, the final performance would not at least totally fail.

**"Sim-to-Sim".** We collect the data in Genesis by running the evaluation for the unified policy using 30.000 environments. We use cylinders to collect the data. We run the evaluation on each cylinder

instance with the maximum number of evaluation trails set to 1,500,000. We use all rollout data to train the joint-wise neural dynamics model (pre-trained using transitions in Isaac Gym). The training is conducted on eight A10 GPUs for 2 epochs with a batch size of 64, which takes approximately two days. We collect the data in MuJoCo using one environment. For each training cylinder instance, we collect 4000 trajectories, resulting in 36,000 trajectories in total. We use all data to train a joint-wise dynamics model (pre-trained using transitions in Isaac Gym).

After that, we train the residual policy for two epochs, which takes about 13 hours. We then deploy the residual policy with the original base policy to the target simulator. The policy is tested on the ContactDB test object set. We roll out the policy using 10 different initial grasps. Reported values are the mean and standard deviation values of per-object average results over 10 trials.

Figure 37 shows a qualitative comparison of the policy's performance w/ and w/o our method to bridge the dynamics gap.

**"Sim-to-Sim" Comparison Settings.** We use the same data collection strategies to collect transitions in each simulator. The difference is that only successful rollouts are kept, resulting in 3280673 trajectories in Genesis, while 23650 trajectories in MuJoCo. These trajectories are leveraged to train their corresponding action compensators for ASAP and UAN. For ASAP, we use the whole hand formulation, different from the per-finger compensator that we leveraged in ASAP's sim-to-real setting. We reward the policy to track both the object state and the hand state: $r^{\text{compensator}} = -k_h \|\mathbf{q}_t^{\text{ref}} - \mathbf{q}\|_2 - k_o \text{ang\_diff}(\mathbf{o}_t^{\text{ref}}, \mathbf{o}_t)$, where $\mathbf{q}_t^{\text{ref}}$, $\mathbf{o}_t^{\text{ref}}$, and $\mathbf{o}_t$ are the hand reference joint state, object reference orientation and object current orientation respectively. $k_h$ and $k_o$ are coefficients to balance hand and object tracking. $k_h$ is set to 1.0. While we add a curriculum to $k_o$. It is set to a small value, *i.e.,* 0.001, at first. And we use the reset number of the first environment to count the reset step. During the first 10 reset steps, $k_o$ is kept at the initial value. While starting from that and until the 200-th reset step, $k_o$ is linearly increased to 2.0. After the compensator has been trained, we tune the policy based on it. The tuned policy is then deployed to the target simulator. We adopt the same evaluation strategy as for our method.

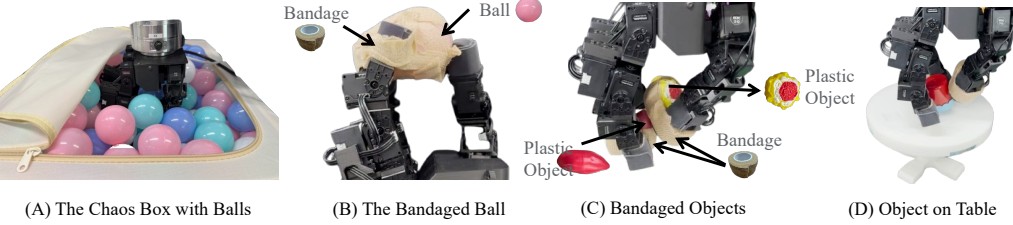

| (A) The Chaos Box with Balls | (B) The Bandaged Ball | (C) Bandaged Objects | (D) Object on Table |

Figure 38: **Autonomous Real Data Collection Setup with Load.** (A) A large box with many soft balls. (B) Bind the object to three fingertips to avoid the object falling off and to add external object influence to the hand. (C) Bind objects to two fingertip,s which adds external influence to the hand via collisions between these objects. (D) Adding a supporting table to avoid the object falling off.

**Grasping Pose Generation.** We generate grasping poses with the "Palm Down" orientation, which are used for the omni wrist orientation rotation training. For details, please refer to the cdoe in the supp ('DexNDM-Code/RL/README.md'). The canonical qpos of LEAP hand, from which we sample random noise to generate the grasping poses, is set to [1.244, 0.082, 0.265, 0.298, 1.163, 1.104, 0.953, -0.138, 1.096, 0.005, 0.080, 0.150, 1.337, 0.029, 0.285, 0.317].

**Real-World Hardware Setup.** We LEAP hand (Shaw et al., 2023) and Franka Arm for conducting real-world experiments (Fig. 39). We use positional control with a control rate of 20 Hz. The positional gain and damping coefficient are set to 800 and 200, respectively.

**Real-World Data Collection Setup.** To collect real-world transition data with varying loads while minimizing human intervention, we developed several strategies, as illustrated in Figure 38.

Among these, the "Chaos Box" with balls proved most effective. Its setup is straightforward: place the box on a table, open it, and position the robot's hand inside with a desired orientation. Crucially, this method operates autonomously, requiring no human intervention during data collection. This setup ensures continuous interaction with a load, as the robot's hand is always in contact with the balls. The constantly shifting positions of the lightweight balls provide a diverse and continuous

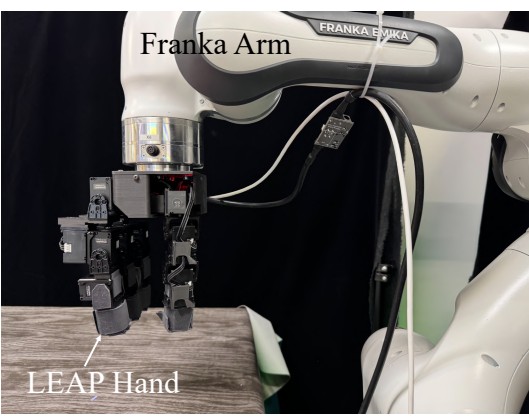

Figure 39: **Real World Experiment Hardware Setup.**

range of loads. Furthermore, the balls' deformable surfaces ensure that these interactions do not damage the robot's hardware. The autonomy of this system allows us to initiate data collection in the evening and let it run overnight unattended.

A key limitation of the Chaos Box is its inability to collect data in a palm-up orientation due to the robot arm's kinematic constraints. To address this, we developed a second setup where a ball is secured to three of the robot's fingers with a bandage (Fig. 38 (B)). Similar to the Chaos Box, this method runs autonomously once initiated. However, binding the ball takes time. A drawback is that the ball's fixed position results in a less diverse set of perturbation patterns.

Two other approaches were explored but ultimately not adopted (Fig. 38 (C,D)). One involved attaching an object to the finger (C), but this was unreliable as the object could fall and require manual reattachment. The other used a supporting table (D), but the object often moved outside the robot hand's workspace, necessitating human intervention to reposition it.

**Robotic Hand Sizes.** We define hand size as the fingertip span: for the D'Claw hand, the distance between diagonally opposite fingertips (19.10 cm); for the Allegro and Leap hands, the distance between the index and pinky fingertips (10.05 cm and 9.50 cm, respectively).

**Real-World Transition Data Collection.** We collect real-world transition data by replaying action trajectories rolled out in the simulation. Each episode contains 400 steps. Actions are executed in the hardware at 20Hz. Collecting one trajectory with a full episode takes approximately 20s. We collect transitions with all six tested hand wrist orientations, that is, palm up, palm down, thumb up, thumb down, base up, and base down. In each orientation, we collect 4,000 transition trajectories. In more detail, we randomly at uniform select 4,000 trajectories from rollouts of all oracle polices with the corresponding wrist orientation. We collect transitions using the "Chaos Box" system.

**Experimental Settings of Ablation Studies.** When comparing real-world performance of different models in ablation studies, we keep the hand in the palm down orientation and test the z-rot performance on three representative objects, including a regular cylinder, a cylinder with higher aspect ratios, and an irregular object. We roll out the policy for rotating the regular cylinders in this specific hand orientation and the rotation direction to construct the simulation dataset, which is composed of 937,275 trajectories, each of which has 400 transition steps.

*Real-World Data Collection.* We collect transition data via the Chaos Box setup (Fig. 38 (A)). We replay action trajectories rolled out in the simulation in the real world to collect the data. We collect 4,000 trajectories, resulting in 1,600,000 transitions in total. In addition, we collect 20 successful rotation trajectories (*i.e.,* object does not fall during the whole episode) with the thumb up orientation on a 5cm size cube by deploying policies in the real environment as the out-of-domain test data.

*Task-Relevant Data Collection.* We collected 1 hour of data per object using three objects: a 5 cm × 5 cm × 5 cm cube, the Stanford Bunny, and a cylinder (radius 5.5 cm, length 5.5 cm). In total, we obtained 111, 87, and 54 trajectories with the cube, cylinder, and Stanford Bunny, respectively.

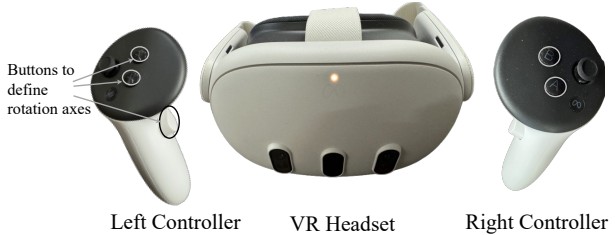

Figure 40: **Quest 3.** We teleoperate the arm using the right controller's pose, while the left controller's pose specifies the desired rotation axis. We also provide a button-controlled mode that restricts rotation to three fixed axes, selected via the X, Y, and LG buttons on the left controller.

*Collecting via Base Waves.* We collect 2,000 trajectories using sine waves, 1,000 trajectories using square waves, while 1,000 using Gaussian noise. When collecting the trajectory using the sine wave, we randomly select a joint to send signals while leaving the other joints fixed. Specifically, we fix other joints to the midpoint of their angle range. For LEAP hand, actuating the joint between mcp link to pip link when fixing other joints would lead to self-collision. So we would not select such joints when replaying trajectories. We use the sine wave with the form $f(t) = \sigma \sin(2\omega t)$. At the beginning of each data collection, we sample $\sigma$ and $\omega$ from a uniform distribution, *i.e.,* $\sigma \sim \mathcal{U}(0.5, 1.0), \omega \sim \mathcal{U}(0.2, 0.5)$. When using the square waves, we use $g(t) = A * \text{sign}(\sin(2 * \omega * t))$, where $A \sim \mathcal{U}(0.5, 1.0), \omega \sim \mathcal{U}(0.2, 0.5)$. We add Gaussian noise to the square wave to collect remaining 1,000 trajectories, *i.e.,* $\hat{g}(t) = A * \text{sign}(\sin(2 * \omega * t)) + \epsilon$, where $\epsilon \sim \mathcal{N}(0, 0.01)$.

*Dynamics Model Training.* The pretrained dynamics model is obtained by leveraging the same model architecture to fit the roll-out simulation trajectories. We then directly tune the model weights on the real-world data for fine-tuning. An evaluation dataset is split out from the 4000 training trajectories with a train: eval ratio of 9:1. The Model with the best evaluation loss is then leveraged to train the residual policy model. We report the final result on the OOD test dataset as the generalization performance. We train the residual policy on the simulation data for one epoch, which would typically cost for about 10 hours using eight A10 GPUs.

**Teleoperation System for Complex Dexterous Manipulation Data Collection.** We demonstrate an important application of our rotation policy: a teleoperation system for complext dexterous manipulation tasks with in-hand rotation. We implement it by pairing the policy with a Quest 3 headset (Fig. 40). Leveraging in-hand rotation, the system completes complex tasks requiring fine-grained finger coordination—scenarios where traditional teleoperation systems (Ding et al., 2024; Cheng et al., 2024) often struggle.

We adapt BunnyVisionPro (Ding et al., 2024) for Franka arm teleoperation. The arm is controlled with the Quest 3 right-hand controller, and we obtain controller states via oculus_reader. We use the left controller's orientation to define the rotation axis and down-weight the component around its short axis to reduce errors when inferring the axis from pose. In practice, this orientation-based specification is not very intuitive, so we introduce a button-controlled mode in which the rotation axis is selected by pressing the X, Y, or LG buttons on the left controller. Although this restricts the available axes to three, we find it sufficient for single tasks; for example, lightbulb assembly and disassembly can be completed using z, -z, and -y rotation modes.

All hand motions, including grasping, are controlled by the policy. We initialize the robotic hand in a default pose. To grasp an object, we approach it and activate the rotation policy. Conditioned on an initial open-hand observation, the policy outputs an action sequence that closes the fingers around the object to achieve a secure grasp.

## D DISCUSSIONS ON RELATED SIM-TO-REAL WORKS

Misaligned physical parameters, discrepancies in their physical models, and numerous unmodeled effects in the actuator and contact dynamics hinder successfully transferring the policy trained in simulation to the real world. Efforts to close this gap mainly fall into four types of approaches: 1) Domain Randomization (DR) expands the distribution of training environment to train robust poli-

cies that are expected to function well in different environments (Loquercio et al., 2019; Peng et al., 2017; Tan et al., 2018; Yu et al., 2019; Mozifian et al., 2019; Siekmann et al., 2020; Sadeghi & Levine, 2016). 2) System Identification (SysID) aligns The simulator dynamics to the real-world in a principled and interpretable way by estimating critical physical parameters from real data (An et al., 1985; Mayeda et al., 1988; Lee et al., 2023; Sobanbabu et al., 2025). 3) Adaptive Policy adapts the policy online according to the real-world dynamics that are implicitly identified from real-world feedback. 4) Neural-based Real World Modeling learns real dynamics to help with policy's transfer (He et al., 2025; Fey et al., 2025; Deisenroth & Rasmussen, 2011; Shi et al., 2018; Hwangbo et al., 2019). As a popular and standard strategy, DR requires heuristic designs (Sobanbabu et al., 2025) to find proper randomization ranges. While generalizable and interpretable, the upper bound of SysID is restricted by the coverage of parameters to be identified. For a successful adaption, the training environment should cover a wide distribution, which is typically achieved by DR. This limits their effectiveness when the real-world dynamics cannot be covered by randomizing the simulated environment. With the potential of aligning all kinds of discrepancies, guiding the policy's transfer via modeling real-world dynamics has the highest upper capabilities, making it the focus of our work. One approach is leveraging neural networks to perform system identification, learning residual dynamics or representations (Shi et al., 2018; O'Connell et al., 2022), followed by developing a model-based controller (Fig. 2 (A)). For systems involving higher degrees of freedom (DoFs) and more complex dynamics, learning a comprehensive dynamics model that supports controller optimization is difficult. An alternative strategy is bridging the gap between an existing simulator and the real world by learning a delta function (He et al., 2025; Fey et al., 2025), followed by policy finetuning to bridge the gap (Fig. 2 (B)).

However, directly extending those approaches to dexterous manipulation, with rich, rapidly varying contacts on moving objects, cannot work. The primary challenge lies in collecting high-quality real world transition data that can cover the vast task distribution, thereby reflecting dynamics during the task execution. This is achieved by replaying waveforms (*e.g.*, sine) or rolling out policies–none can work in our setting.

Wave-based collection is untenable: manipulated objects enlarge the transition space and impose time-varying loads, yielding dynamics unlike the no-object regime (see Appendix A.3). Because parameterized waves cannot reliably manipulate an object in air, they must be run without it, offering poor coverage of in-hand dynamics. On-policy rollouts across diverse objects are costly and unscalable—requiring frequent human resets (placing the object back in hand), biasing data toward easy objects, confining coverage to the policy rollout distribution, and suffering from low quality (imperfect policy).

Extending their methods to manipulation also necessitates modeling the interaction dynamics, which inevitably involves modeling the object. There are two approaches to model the object: 1) Explicitly including the object in the dynamics system. Achieving this requires collecting real-world transition trajectories with object state annotations. However, obtaining object states (*e.g.,* using vision-based pose trackers like FoundationPose (Wen et al., 2023))) is difficult and impossible for some cases. For instance, FoundationPose (Wen et al., 2023)) are unreliable for axis-symmetric, tiny, and occluded objects (see Sec. B.4). Besides, the object pose tracking results are noisy. It is also very time-consuming, requiring extra time to launch and frequent human interventions. Using the small, noisy dataset cannot even make the first stage, compensator training, successful. Another strategy is modeling the object as a time-varying disturbance. This requires us to a) collect transition data with the object loads; b) manage to simulate the object's influence to the hand in the simulator; and c) train the compensator to track the hand state only. However, it is almost impossible, as reproducing its influence would require near-perfect alignment of geometry, initialization, and contact evolution—unrealistic under mismatched dynamics.

**What data can we use to train ASAP and UAN in dexterous manipulation?** We discuss three options: (1) transitions with object-state annotations—possible in principle but impractical, as object states are hard and noisy to obtain and, in our tests, such small and noisy data fail to train their compensator; (2) our autonomously collected trajectories with randomized object loads—unsuitable because replicating the influence of such object loads to the hand in the simulator is infeasible; (3) free-hand data—the only practical choice, on which we train their compensator to close the dynamics gap in the free hand scenario. Hence, we use free hand transitions when comparing with their methods.

## E  LLMs Usage Disclosure

We use LLMs to aid and polish writing. Details are described as follows:

- Polish sentence or paragraph to correct grammatical errors, improve the writing, and make statements more concise. Typically, we will give the LLM a sentence with the instruction like "Help me polish this and make it more concise: [sentence to polish]".

- Refine theorems and proofs to improve rigor and professionalism. After writing each theorem and its proof, we iterate with an LLM to identify potential logical gaps and revise accordingly—for example, by adding missing assumptions or clarifying intermediate steps. We also ask the model to assess the proof's soundness, offer suggestions, based on which we will revise the proof. Finally, we use it to make the statements' wording more precise and professional.

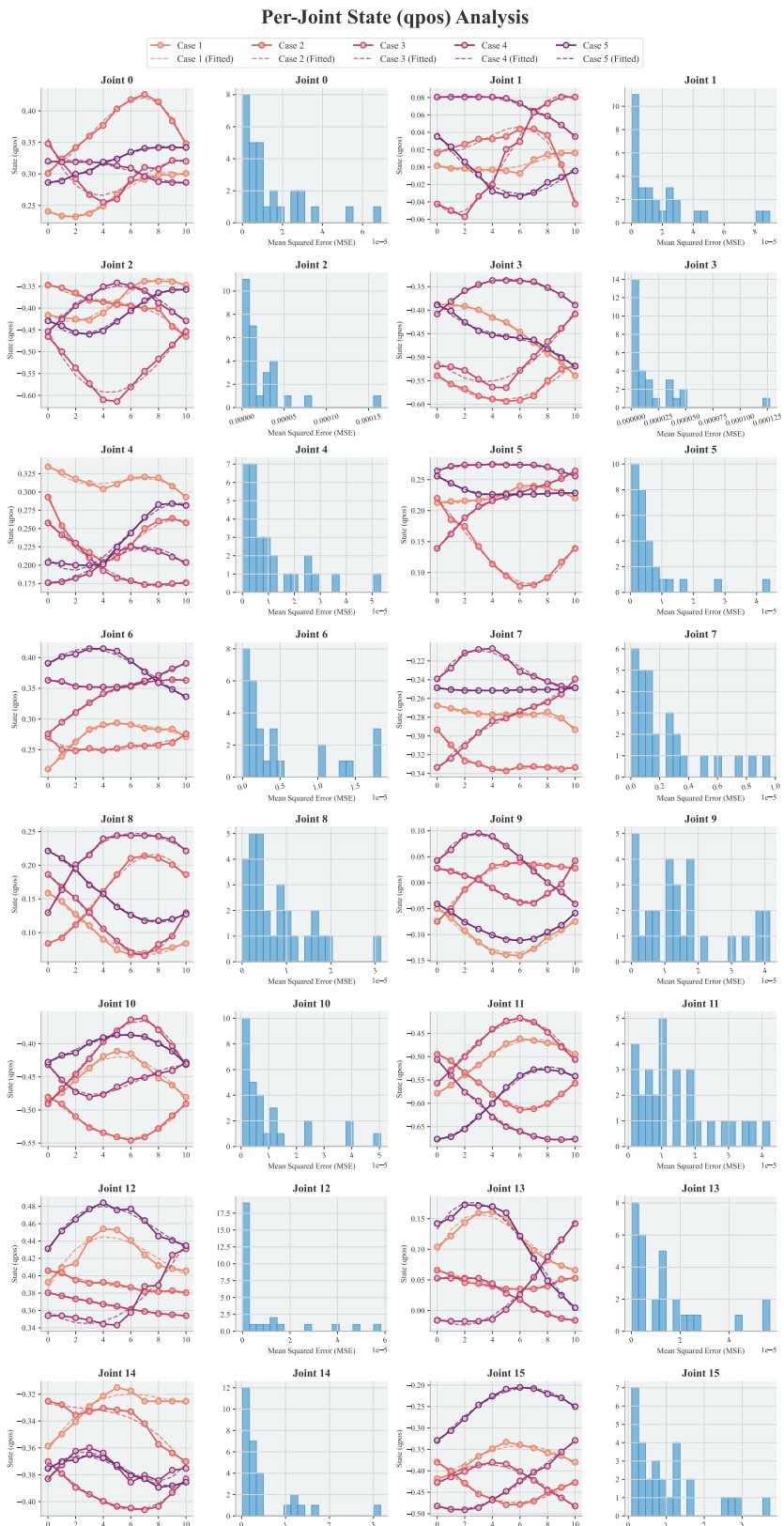

Figure 41: **Polynomial Fitting (order = 3) and Error Distribution of Per-Joint State Sequences (window length = 10).** In each group with two subfigures, the left one draws the original data sequence and the fitted sequence using a 3-ordered polynomial function while the right one shows the fitting error distribution.

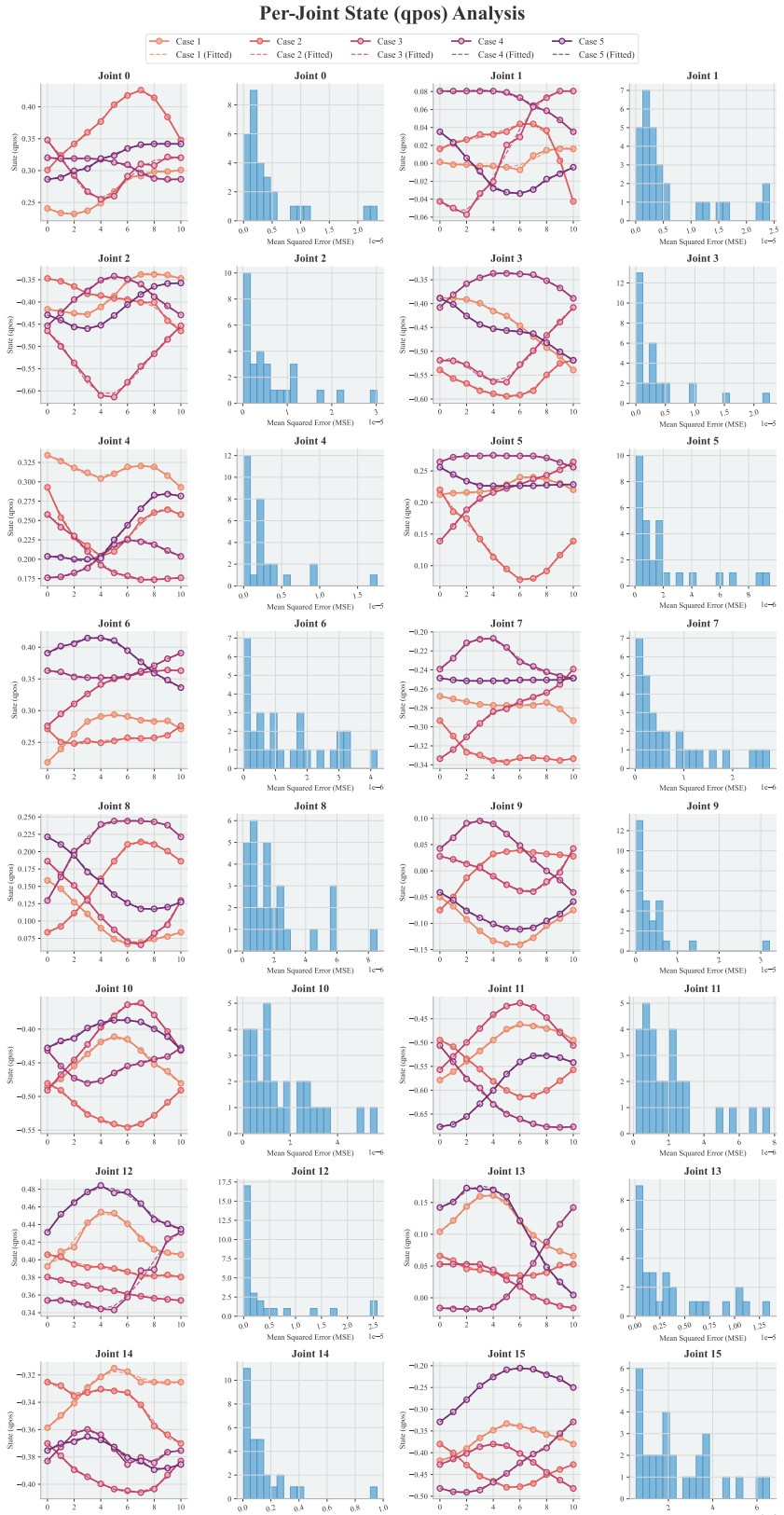

Figure 42: **Polynomial Fitting (order = 5) and Error Distribution of Per-Joint State Sequences (window length = 10).** In each group with two subfigures, the left one draws the original data sequence and the fitted sequence using a 5-ordered polynomial function while the right one shows the fitting error distribution.

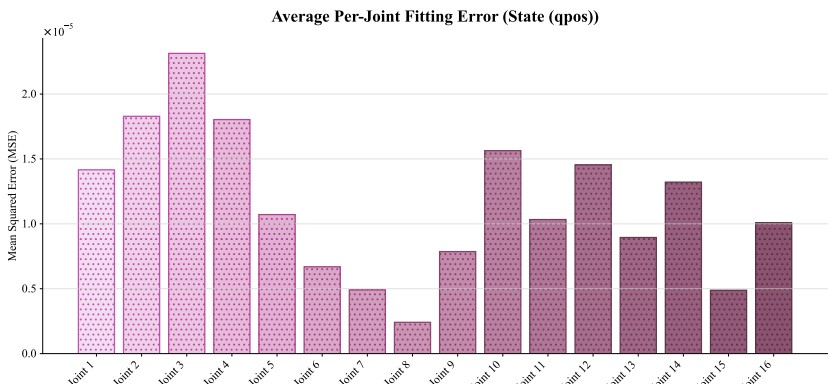

Figure 43: **Per-Joint Average Polynomial Fitting (order = 3) Error.**

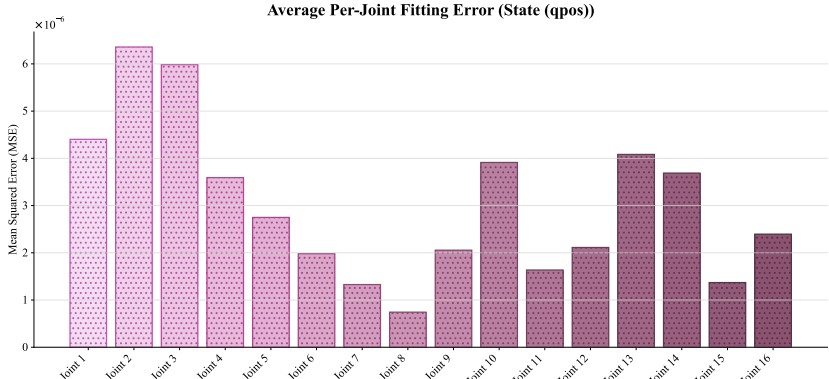

Figure 44: **Per-Joint Average Polynomial Fitting (order = 5) Error.**

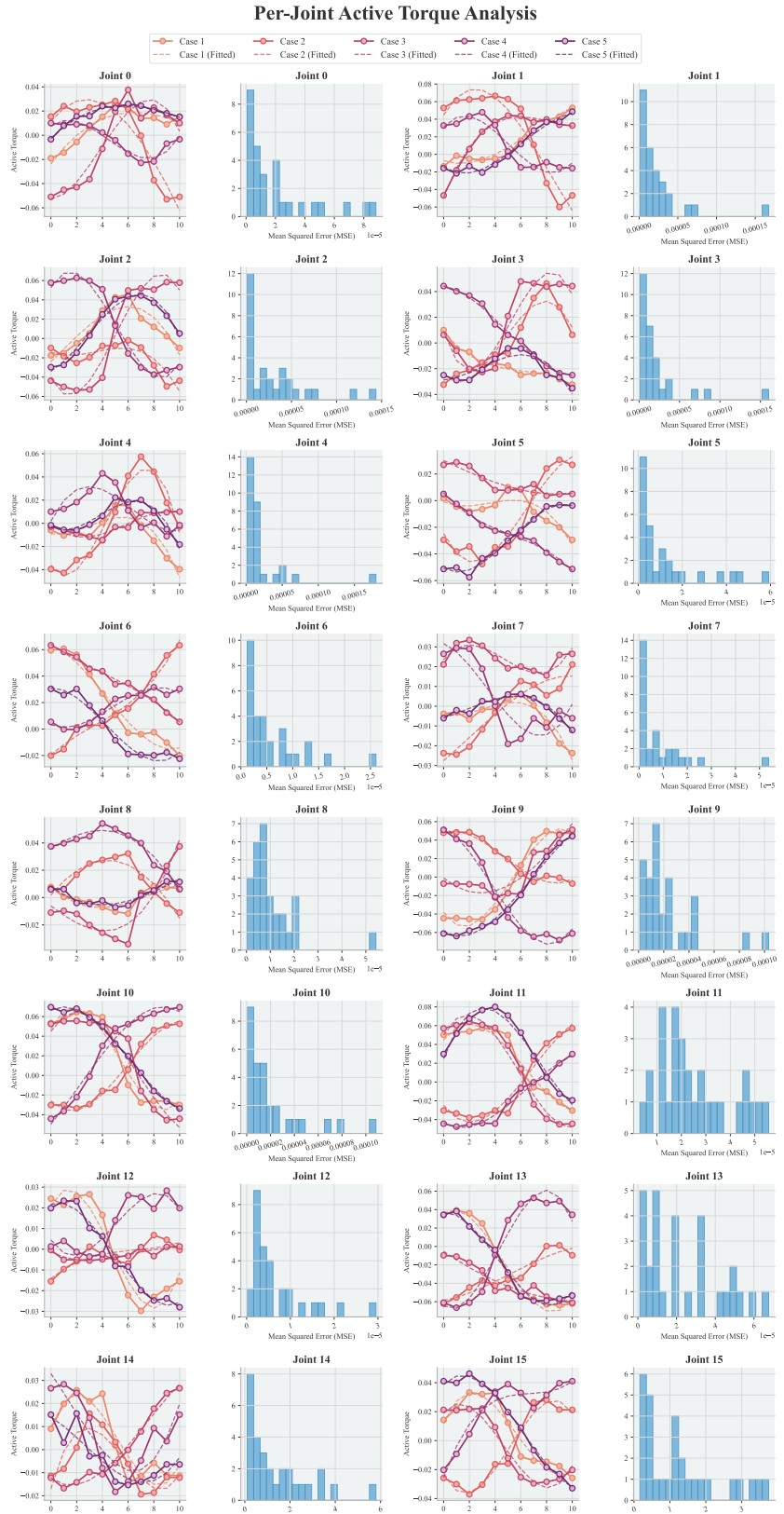

Figure 45: **Polynomial Fitting (order = 3) and Error Distribution of Per-Joint Active Force Sequences (window length = 10).** In each group with two subfigures, the left one draws the original data sequence and the fitted sequence using a 3-ordered polynomial function while the right one shows the fitting error distribution.

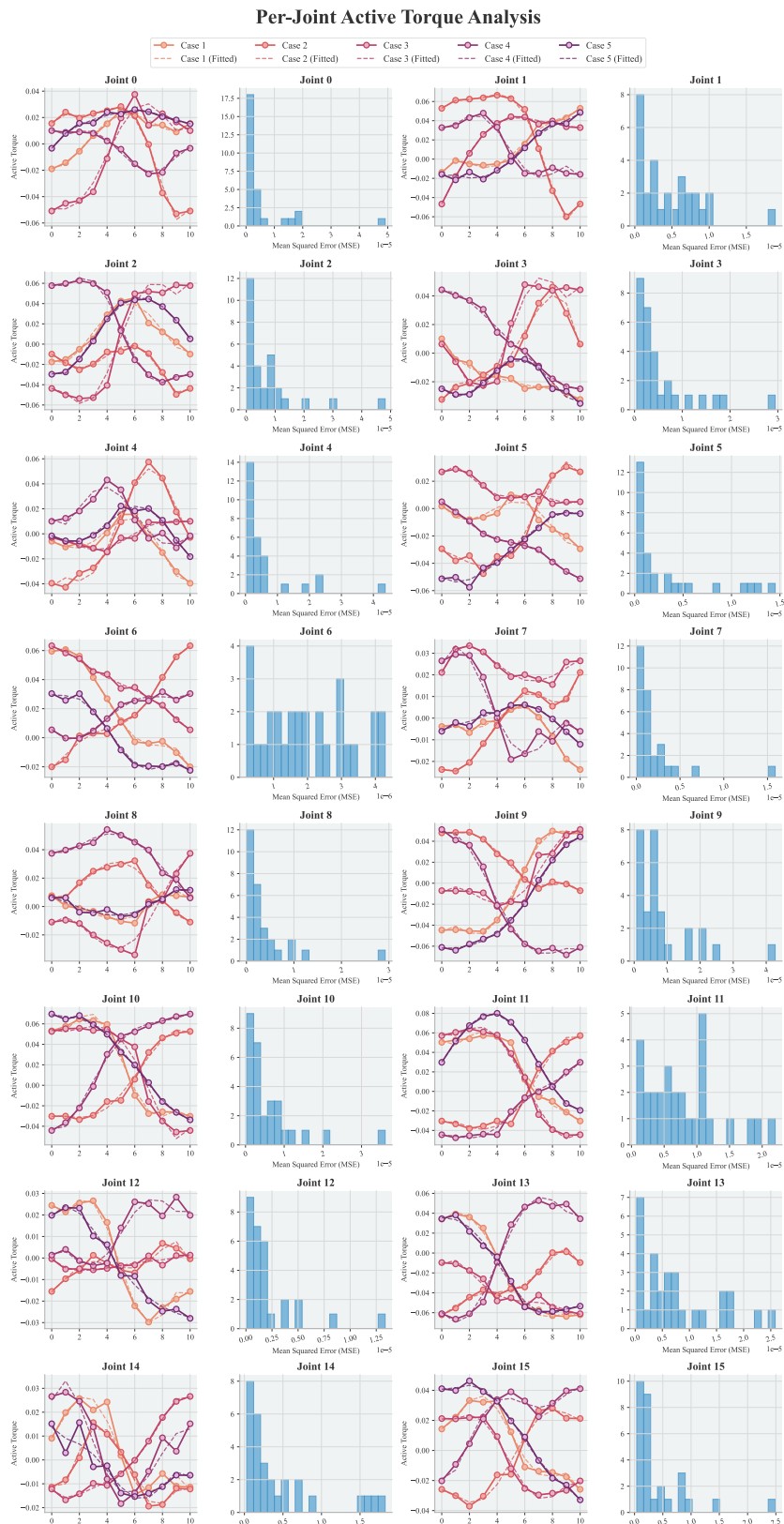

Figure 46: **Polynomial Fitting (order = 5) and Error Distribution of Per-Joint Active Force Sequences (window length = 10).** In each group with two subfigures, the left one draws the original data sequence and the fitted sequence using a 5-ordered polynomial function while the right one shows the fitting error distribution.

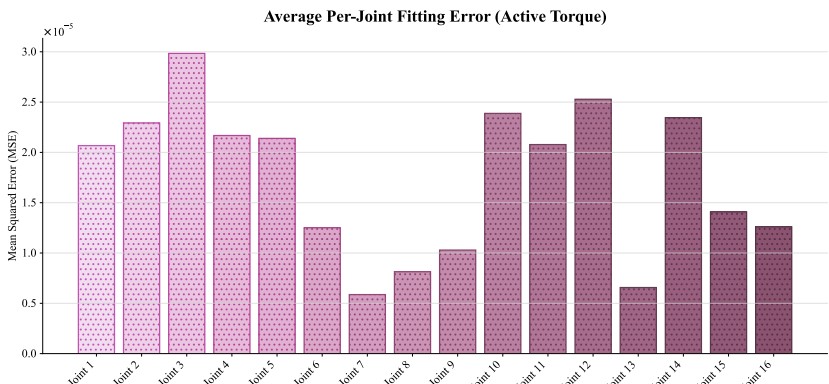

Figure 47: **Per-Joint Average Polynomial Fitting (order = 3) Error.**

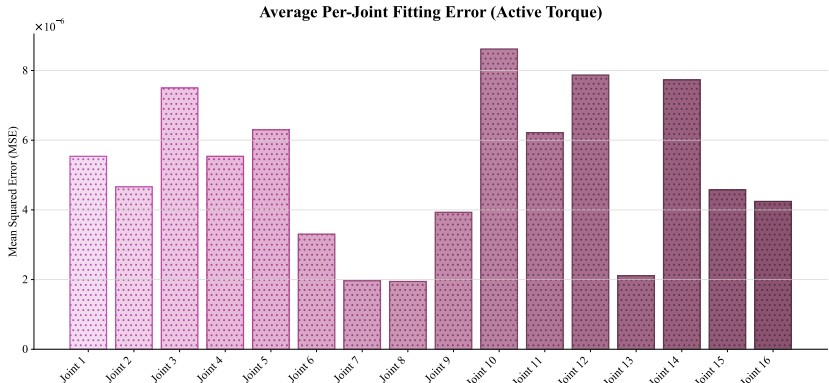

Figure 48: **Per-Joint Average Polynomial Fitting (order = 5) Error.**

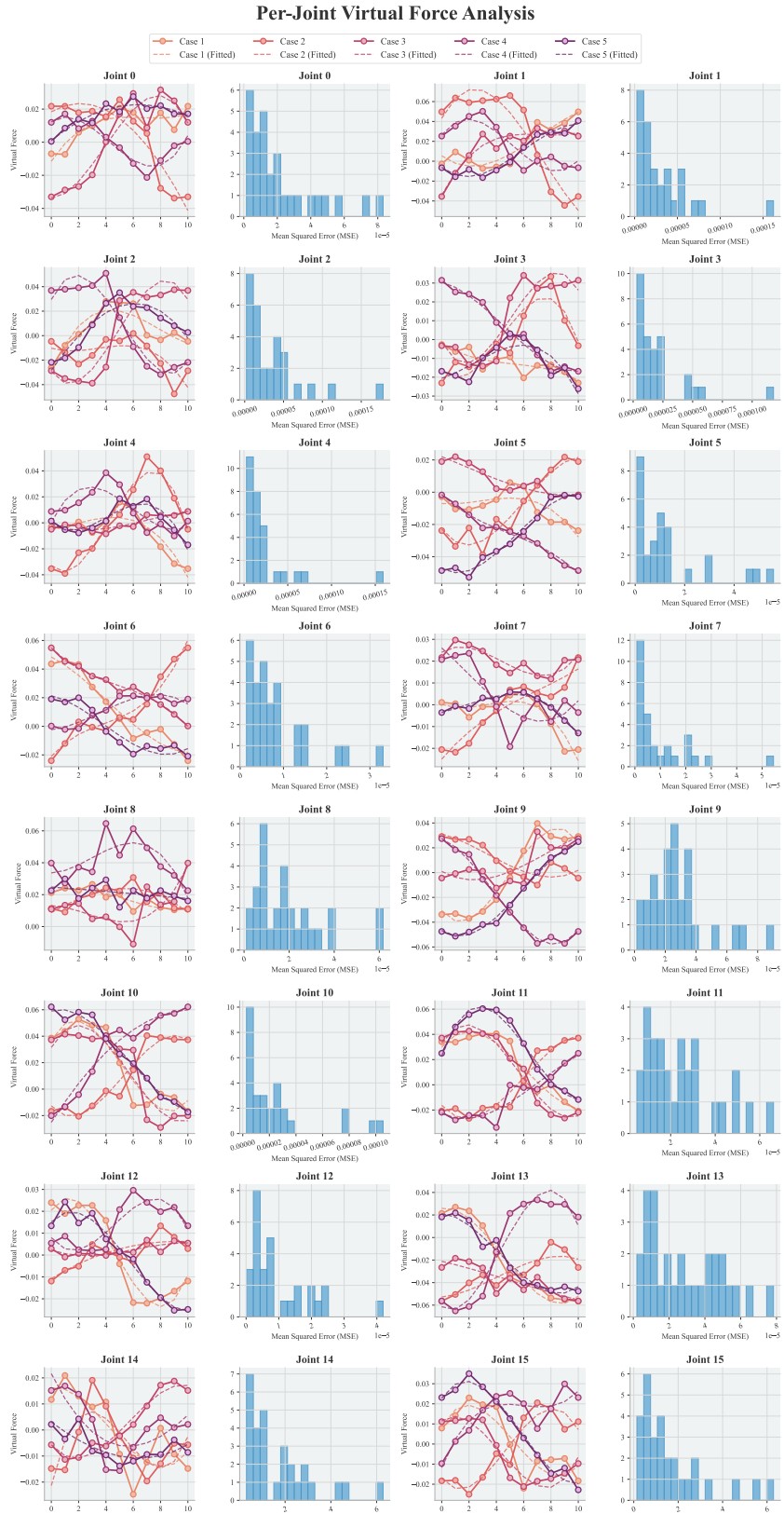

Figure 49: **Polynomial Fitting (order = 3) and Error Distribution of Per-Joint Virtual Force Sequences (window length = 10).** In each group with two subfigures, the left one draws the original data sequence and the fitted sequence using a three-order polynomial function while the right one shows the fitting error distribution.

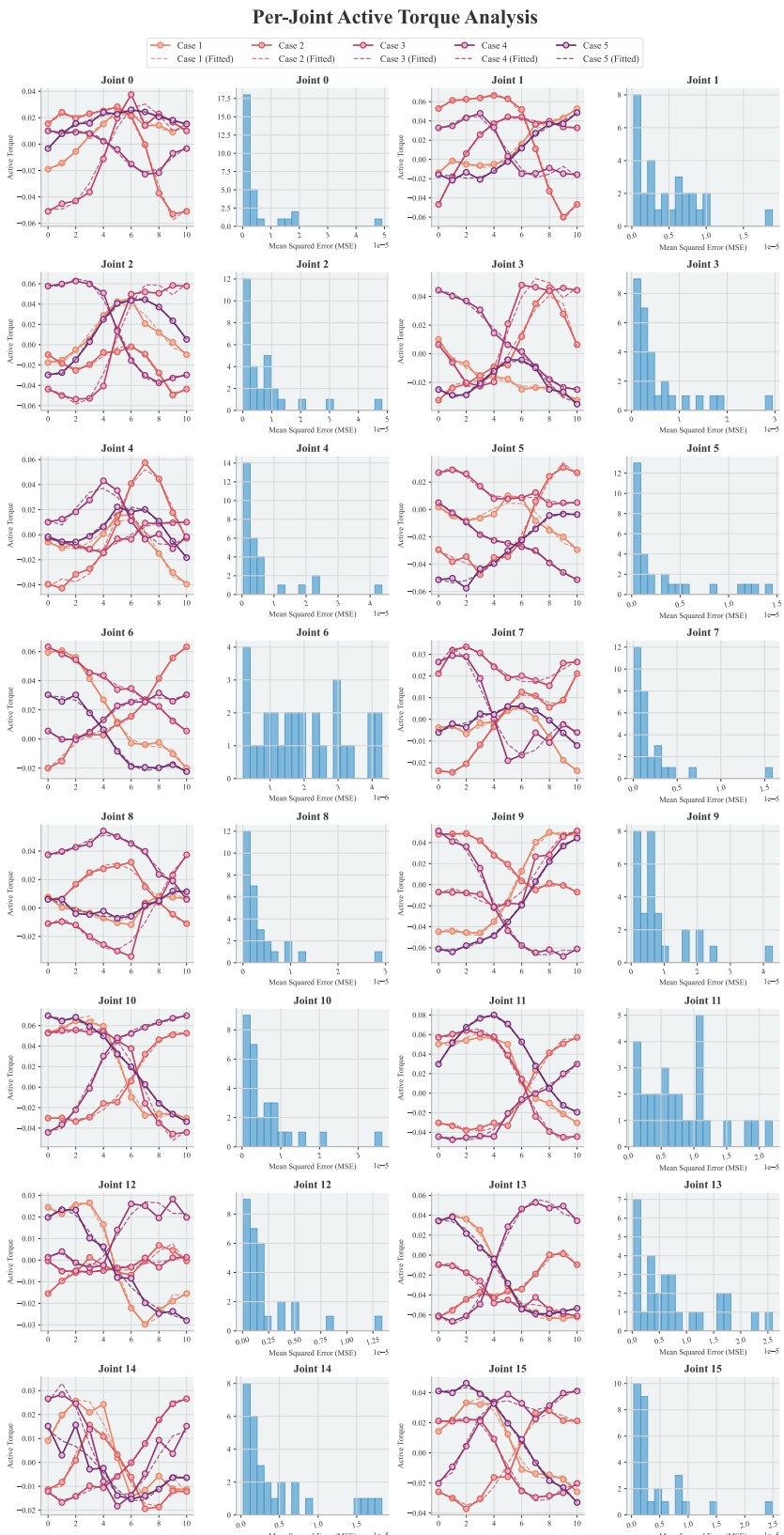

Figure 50: **Polynomial Fitting (order = 5) and Error Distribution of Per-Joint Virtual Force Sequences (window length = 10).** In each group with two subfigures, the left one draws the original data sequence and the fitted sequence using a five-order polynomial function, while the right one shows the fitting error distribution.

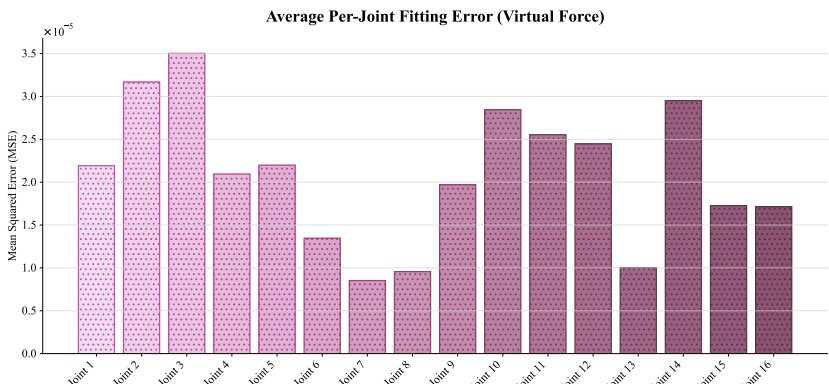

Figure 51: **Per-Joint Average Polynomial Fitting (order = 3) Error.**

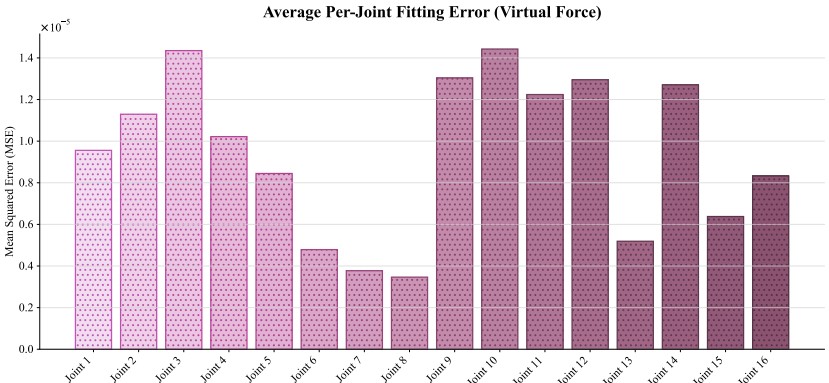

Figure 52: **Per-Joint Average Polynomial Fitting (order = 5) Error.**

