# OpenReview forum: "DexNDM: Closing the Reality Gap for Dexterous In-Hand Rotation via Joint-Wise Neural Dynamics Model"
_ICLR.cc/2026/Conference — ICLR 2026 Poster_

### Official Review · Reviewer_zo2q · 2025-11-01

**Soundness:** 1
**Presentation:** 4
**Contribution:** 2
**Rating:** 2
**Confidence:** 5

**Summary:**

This paper proposes a neural joint dynamics alignment approach to address the sim-to-real challenge in dexterous manipulation. The authors first train a neural joint dynamics model on real-world data, then use this learned model to train a residual policy on top of a simulation-pretrained policy. The goal is to ensure that the predicted future hand state under the learned real-world dynamics matches that observed in simulation. The approach is evaluated on several rotation-style manipulation tasks, showing some empirical improvement.

**Strengths:**

The engineering effort to reproduce rotation manipulation tasks on a low-cost robotic hand is commendable.

The experimental section is relatively thorough, demonstrating multiple tasks and ablations.

**Weaknesses:**

**1. Soundness**. The main weakness of this paper lies in the soundness and validity of the proposed method. Conceptually, the approach overlooks several key aspects of manipulation dynamics:

1A. Object-dependent dynamics. Unlike locomotion on a rigid terrain, where the dynamics primarily depend on the agent itself, manipulation dynamics are determined by both the hand and the object. The proposed neural joint dynamics model cannot predict finger–object contact events when the past frames correspond to free-hand motion. Because the model relies solely on proprioceptive input, it is impossible to infer or reconstruct the object’s geometry and contact state from proprioception (think about parallel gripper gripping a small marble-- you can only know there is something between the fingers. You can not tell the exact object location without touch).

1B. Based on 1A, we can further show that this method can be wrong in simplest scenarios. See comments in 4.

1C. Limited need for learned joint dynamics. For commercial robot hands driven by high-quality servo motors, the joint behavior is typically very repeatable (this is unlike humanoid robots, whose joint dynamics is more complex due to whole-body coupling). In such cases, traditional system identification (SysID) methods are often sufficient to close the sim-to-real gap in actuation dynamics. The authors should include SysID baseline results to determine whether their learned model actually provides any improvement.

1D. Unconvincing data transfer across object scales.
It is unclear why training on large spherical objects would generalize to manipulating much smaller objects. This data choice is questionable, since the resulting joint states for small-object manipulation are likely out of distribution. The claimed transferability is therefore unconvincing.

**2. Novelty.**

2A. The conceptual novelty of this work appears limited. Learning a neural joint dynamics model has already been explored in previous research, including for humanoid and manipulation systems.

2B. The idea of distilling multitask experience into a single model has been investigated by prior work such as DexterityGen, which also demonstrated teleoperation capabilities under a very similar setup. The authors also refer to their performance as "unprecedented" while this is not true. These related works are not cited or discussed adequately.

**3. Theoretical section**

The theoretical discussion in this paper is weak and disconnected from the proposed method. Theorem 3.1 is a well-known result in information theory. Theorem 3.2 is trivial because if $B\subset A$, then for any function or functional $h$, $\sup_{x\in B} h(x)\leq \sup_{x\in A} h(x)$. In the theorem, $A=[f],B=[f\circ g_X]$ and the authors did not make any assumptions on f. If all the $f, g_X$ are continuous (which is natural as the authors use neural networks), then $B\subset A$ and we are done with $\leq$ part.  The value of  showing $<$ is marginal, a nontrivial result typically requires $\gamma$ contraction ($\gamma < 1$).

Besides, these results appear to have little relevance to the main technical contribution and seem to have been added without meaningful connection. A more appropriate analysis would examine error propagation or model mismatch under dynamics error, similar to what has been done in model-based reinforcement learning literature. For instance, the authors should explicitly present the return bound relating $J_\pi$ and $J_\pi^{res}$ ($J_\pi$ is sum of reward when rolling out $\pi$), we should expect some form of $ J_\pi^{res} \geq J_\pi - C(\epsilon)$, where $\epsilon$ is the error of the model.

**4. Conceptual counterexample**

The proposed approach can fail even in simple cases. Consider a minimal manipulation world model with two parts of the state $s=(s_o,s_h)$: the object state $s_o$ (which can take values 0, -1, or 1, representing neutral, fail, and success) and the hand state $s_h$ (which can be 0 or 1). Together there are six possible world states. We have 2 actions, a=0 and a=1.

**In simulation, action a=1 always succeeds**. Specifically,

(0, 0):  a=0 -> (0, 1), receive reward 0.

(0, 0):  a=1 ->  (1, 1), receive reward 1, terminate.

(0, 1):  a=0 -> (0, 1), receive reward 0.

(0, 1):  a=1 -> (1, 1), receive reward 1, terminate.

Clearly, the best policy is to execute A=1 at any state. So, $\pi^*(s)=\pi_{sim}(s) = 1$.

**In the real world, action a=1 may immediately cause failure**:

(0, 0): a=0  -> (0, 1), receive reward 0.

(0, 0): a=1   ->  (-1, 1). **Fail**, receive reward -100, terminate. (e.g. an abrupt action in real world at initialization can lead to failure)

(0, 1): a=0   ->  (0, 1), receive reward 0.

(0, 1): a=1   -> (1, 1).   receive reward 1, terminate.

The best policy in real world takes a=0 at (0, 0), and then a=1 at (0, 1). The optimal simulation policy $\pi^*$ will run into (-1, 1) and receive -100 reward at the first step in this real world environment.

Now, we examine the proposed approach. We first train a neural joint dynamics model $F(s_h, a)=s_h’$ which only considers the dynamics of $s_h$. According to the real world data above, we have $F(0, a) = 1, F(1, a) = 1$ for $a= 0,1$. Then according to L291, we further train a residual policy $\pi_r$ so that $F(s_h, \pi_r(s)+\pi^* (s))=s’_h$ given any $(s, s’)$ drawn from the optimal simulation policy $\pi^*$'s trajectory distribution. Since the simulation policy’s trajectory is (0, 0): a=1 -> (1, 1), the requirement is $F(0, \pi_r(0)+1)=1$. Since $F(0, 1)=1$, $\pi_r(0)= 0$ can work and no correction is required. As a result, the corrected policy still outputs 1 and runs into failure state (-1, 1) at the first step.

This simple example shows that without explicitly modeling object-dependent dynamics, the proposed residual adaptation cannot reliably bridge the sim-to-real gap.

**Questions:**

See above.

I believe this paper would be significantly stronger if the authors removed the unnecessary (and wrong) theoretical and adaptation components, and instead focused on genuinely extending the capabilities of the considered skills while clearly presenting what works and what does not. These "novelties" feel very forced.

I appreciate the engineering effort and the demo. However, these results have been demonstrated by several previous works, and the authors did not show any significantly new capability (e.g., precisely rotate the object to a specific pose and stop). Therefore, given that the science part of this paper is also flawed, this paper can not be accepted. **A demonstration is only meaningful if it showcases substantially new capabilities or is supported by sound, novel technical contributions.** Personally speaking, I believe a strong paper in this field should emphasize capability breakthroughs. Since reproducing an existing robotic system is inherently challenging and involves many compounding factors (what if you remove the tape from your fingertip and object? what if you tune SysID better? what if your ROS interface is slightly less laggy?), it becomes difficult to clearly quantify the effectiveness of an additional component integrated into an already functional setup. Novel capabilities are definitely more measurable and visible. I encourage the authors to continue their solid engineering efforts and submit an improved version to a future robotics conference.

---

> ### Author Response · Authors · 2025-11-18
> **Author Response (part 1 of 13)**
>
> Dear Reviewer zo2q,
>
> Thanks for your time, detailed review, and all comments. We sincerely appreciate your recognition of our experimental evaluations. We also thank you for raising concerns regarding our method's soundness, contribution, value of the theoretical part, and its distinction from prior work.
>
> In this response, we elaborate on the rationale behind our method, clarify our technical contributions and the positioning of our work in the literature, respond to your concerns about our theoretical discussions, and provide detailed responses to your concerns and questions. We hope our explanations can adequately address your concerns.
>
>
>
>
>
> ### Soundness of the Method (I) -- Modeling "Object-Dependent" Dynamics
>
> > "...main weakness of this paper lies in the soundness and validity...several key aspects of manipulation dynamics...object-dependent dynamics..."
> >
> > "...cannot predict finger–object contact events...free-hand motion...solely on proprioceptive input... impossible to infer or reconstruct the object’s geometry and contact state..."
> >
> > "...can be wrong in simplest scenarios...See comments in 4"
>
> We appreciate the reviewer’s point that the dynamics of manipulation need to account for the influence of the object — we fully agree with this view. What we want to make clear is that we are ***not learning a free-hand dynamics model***, but rather a dynamics model that ***does consider the object’s influence***, which is achieved **implicitly** through the **hand’s *proprioceptive state-action history***.
>
> Although it may seem surprising that the object’s effect can be captured implicitly via the hand’s own state-action history, this approach has been adopted in many prior works (*e.g.*, [Hora], [RotateIt], [AnyRotate], [Text2Touch], [DexCtrl]) and is widely recognized as reasonable. The idea originates from locomotion, specifically [RMA], which proposes using the robot dog’s own state-action history to implicitly model environmental properties such as ground friction and terrain height.
>
> The reason for not including such extrinsics (*e.g.*, terrain height, friction, object mass, object shape) directly in the policy’s observation, but instead using the robot’s own state-action history, is that these extrinsics are typically ***unknown parameters*** or their estimates are ***noisy***. As noted in RMA, this approach is *"analogous to the operation of a Kalman filter for state estimation from the history of observables"*.
>
> Thus, although the dynamics model learns only from the hand’s proprioception history, it can still implicitly capture the influence of the object. This strategy was initially proposed in [Hora] and has been widely adopted and recognized in the in-hand manipulation literature (*e.g.*, [RotateIt], [AnyRotate]). Among these works, [Hora] and [RotateIt] provide particularly detailed analyses. For example, Figure 6 of [Hora] demonstrates that the information in the hand’s state-action proprioception can distinguish an object’s *shape, size, and mass*, while Figure 9 of [RotateIt] shows that the information encoded in the state-action history can be used to *roughly predict the object’s shape*. Below, we give an example:
>
> - A dexterous robotic hand can determine an object's weight by correlating its intended actions with its proprioceptive feedback. This process begins when the hand executes a motor command—the "action"—to lift an object. The hand's internal sensors then provide a stream of proprioceptive data detailing its actual movement in response. A discrepancy between the expected and actual motion allows the system to infer the object's mass. For example, if a moderate lifting force (inferred from the gap between the commanded action and the actual state) is applied but the hand barely moves, the object is correctly inferred to be heavy. Conversely, if the same force causes the hand to accelerate upward much faster than anticipated, the object is understood to be light. It is this fundamental comparison between a commanded action and its sensory outcome that enables the deduction of physical properties.

---

> ### Author Response · Authors · 2025-11-18
> **Author Response (part 2 of 13)**
>
> Our joint-wise dynamics model design further generalizes the idea of RMA in manipulation (proposed in [Hora]) to modeling a single joint --- treating each joint as the "robot" and treating other joints and the object as the "environment". The joint-wise neural dynamics model oberves the state-action history of a single joint and predicts its next transitioned state. Intuitively, analogously to predicting the environment extrinsics from the robot dog's state-action history in [RMA] and modeling object parameters and shape from the robot hand's state-action history in [Hora],  the state-action history of a single joint can implicitly account for the combined effect of inter-joint coupling and the object. In Section 3.3 and A.3, we explained the rationality of the joint-wise dynamics model and its capabilities in modeling the external inter-joint coupling and object influence. Therefore, the joint-wise neural dynamics model for a single joint "implicitly perceives" the net effect of other joints and the object. Every single joint's transition is predicted with external influence considered. The whole hand transition is predicted with the object implicitly considered. Thus, the  conceptual counterexample proposed by the reviewer does not hold -- explained below:
>
> - The counterexample the reviewer provided is based on the assumption that our neural dynamics model cannot capture the influence of the object. However, as we explained above, predicting each joint's dynamic evolution from its own state-action history allows the model to implicitly capture external influences from the object. Based on this, let us revisit the reviewer's example. Since our model can implicitly encode the object’s influence, the learned neural dynamics can be represented as $F((0, 0), 1) = (-1, 1)$, $F((0, 1), 1) = (1, 1)$, $F((0, 0), 0) = (0, 1)$, $F((0, 1), 0) = (0, 1)$. The residual policy $\pi_r$ is optimized to minimize $\Vert F((0, 0), 1 + \pi_r(0, 0)) - (1, 1) \Vert $ and  $\Vert F((0, 1), 1 + \pi_r(0, 1) ) - (1, 1)  \Vert$. From this, we obtain $\pi_r(0, 0) = -1$ (since it is a *residual* action model, whose output is not constrained to the discrete action space $\{0, 1\}$), and $\pi_r(0, 1) = 0$. Therefore, we can achieve the real-world transition $(0, 0) \rightarrow (0, 1) \rightarrow (1, 1)$.
>
> In the following, we further clarify ***why our dynamics model is designed without explicitly incorporating object information*** (*i.e.*, using only hand proprioception), and ***why we adopt a joint-wise modeling strategy instead of a whole-hand dynamics model***.
>
> **Why we do not explicitly include object information into the modeling process?** As explained in the first two paragraphs of Section 3.3 and in Appendix D, we have provided detailed reasoning on why we do not explicitly model the object in our neural dynamics. The reasons are as follows:
>
> - An imperfect object state estimator often introduces noise into the collected data and may even fail in certain cases, such as for axis-symmetric objects (as demonstrated in Sec. B.4, “*Inherent Limitations of Task-Relevant Data Collection*”, and “*Case Study on Estimating Object Poses via Foundation Pose*”). This would limit the collected data to only those objects whose states can be successfully estimated. Moreover, noisy object poses can further degrade the data quality.
> - Explicit object modeling demands high on training data. It should contain distributionally relevant in-hand manipulation trajectories with object state annotations. It should cover successful manipulation trajectories with a wide variety of objects. However, collecting such data requires frequent human resets, making the process extremely expensive. In addition, the diversity of the collected trajectories will be constrained by the base policy's capability --  for example, if the base policy cannot rotate long or small objects, then such data cannot be collected. These issues are also discussed in the first two paragraphs of Section 3.3, as well as in Sections B.4 and D.
>
> In contrast to explicitly modeling the object, learning from the robot hand's state-action proprioception history can get rid of the need for estimating object states, but still accounts for the influence of the object implicitly.

---

> ### Author Response · Authors · 2025-11-18
> **Author Response (part 3 of 13)**
>
> **Why do we use joint-wise dynamics modeling, and why don't we use the whole hand dynamics modeling strategy?** It is because the joint-wise neural dynamics model can generalize much better when the hand-object interaction distribution shifts (analyzed in Sec. 3.3 and supported by empirical evidences in Sec. 4.2, 5 ). This property enables it to generalize to our interest in-hand rotation task by being trained on distributionally imperfectly aligned data, which further unlocks our autonomous data collection strategy.
>
> Even if we do not explicitly model the object, learning the whole hand dynamics directly by implicitly accounting for the object's influence from the hand state-action history still demands high on data. It still requires distributionally relevant data that contains successful trajectories describing the robot hand rotating a wide range of objects. However, as discussed above, it is difficult, expensive, and almost impossible due to the imperfect policy and the requirement of expensive human-in-the-loop resetting (Section 5). In section B.4 and section D, we provide more discussions.
>
> In comparison, joint-wise neural dynamics can learn from distributionally imperfectly aligned data. The learned dynamics model could be transferred to the in-hand rotation scenario to predict each joint's transitioned state. It is because of the generalization ability of the joint-wise neural dynamics model. We demonstrate this property from both the theoretical perspective (Theorems 3.1, 3.2) and empirical evidence (Section 4.2, 5). The generalization ability is the key that resolves the data collection challenges mentioned above--in this way, we can learn the dynamics model from distributionally imperfectly aligned data, which further allows us to collect data in a fully autonomous way. In more detail, we collect data by replaying action sequences on the robotic hand in a ``Chaos Box'' with soft balls (Sec. 3.3). The hand experiences stochastic contacts during the replaying process. And we discover that this strategy can help us collect the "right" training data for joint-wise neural dynamics model -- the distribution of the state-action history of the joint-wise dynamics model constructed by the autonomously collected data can roughly cover the space formed by the task-aware in-hand rotation data (Figure 4, Sec. A.5). In this way, we can collect real world data without human intervention in a more scalable way.
>
> > "Unconvincing data transfer across object scales....spherical objects...much smaller objects...out of distribution..."
>
> That’s a great question. It is precisely the *joint-wise neural dynamics model* that enables us to learn the dynamic evolution of each joint from the data collected in the Chaos Box. Specifically, by predicting the next-frame transition of each joint based on its own state-action history—implicitly capturing the influence of other joints and the object—we can then transfer the learned model to various *in-hand rotation* tasks, where it predicts the next-frame transition of each joint during object rotation. It is because for the two types of data, namely data collected in Chaos Box and the in-hand object rotation data in the real world, though the distributions of their whole-hand object interaction data (state-action history of the whole hand) differ a lot from each other, the distribution of each joint's state-action history is closer. In Figure 4 of the main text and Sec. A.5 in the Appendix, we demonstrate this property. The real-world data is collected using a cube.
>
> Additionally, to directly address the concern raised in transferring the joint-wise neural dynamics model learned in the ``Chaos Box'' to small objects. We additionally collect 15 in-hand rotation trajectories for a small object, *i.e.,* corn. We use T-SNE to visualize the per-joint state-action history space achieved by the autonomously collected data and the real-world rollouts on the small corn object. Results show that the joint state-action history space formed by the autonomously collected data can still offer a good coverage for that of policy rollouts on the small object. Please refer to Sec. A.5 of the revised manuscript for details.

---

> ### Author Response · Authors · 2025-11-18
> **Author Response (part 4 of 13)**
>
> ### Soundness of the Method (II) -- Necessity of Learned Joint Dynamics and Comparisons to SysID
>
> > "Limited need for learned joint dynamics...traditional system identification (SysID) methods are often sufficient...include SysID baseline results..."
>
> We are afraid that the reviewer might have underestimated the sim-to-real dynamics gap involved in in-hand manipulation tasks. We elaborate below:
>
> - Without considering the object (*i.e.*, focusing only on the hand itself), the hand’s dynamics represent a classic multi-body dynamic system where the motion of each joint is not independent — it is influenced by the states of other joints, *i.e.,* inter-joint coupling.
>   - While in cases where the palm remains stationary, a joint’s motion is affected by other joints on the same finger — which is indeed simpler than humanoid dynamics — but it still cannot be regarded as free of inter-joint coupling.
>   - If we only considered free-hand motion (without objects in hands) and aimed to align the resulting state sequences between simulation and reality, then traditional methods such as SysID might suffice.
> - However, our focus is on a more complex scenario—manipulation with an object in hand—which significantly increases the system’s dynamic complexity due to the rich, evolving hand–object interactions. These interactions often involve multiple types of contacts and frequent transitions between them. This, in turn, makes the traditional methods like SysID and DR not sufficient to close the gap. By learning *neural dynamics*, our goal is to bridge not only the actuator-level discrepancies but also the *physics-level sim-to-real gap*, such as *contact dynamics* — which cannot be resolved merely by tuning limited parameters within the simulator. However, dynamic learning and meaningful data collection in the real world are quite difficult. And our joint-wise dynamics model and automatic data collection strategy were specifically designed to address these challenges.
>
> **Comparison with SysID methods.** Our base model (denoted as **"Direct Transfer"**) has already been trained with both system identification and extensive domain randomization. The details of our SysID process and results are presented in *Sec. C: Automatic System Identification*. In Tables 2, 4, and 5, the comparison between *"DexNDM"* and *"Direct Transfer"* can be interpreted as a comparison between our full method and the SysID baseline. As shown, *"DexNDM"* significantly outperforms *"Direct Transfer"*, demonstrating the necessity of learning neural dynamics to effectively close the sim-to-real gap.
>
>
>
> ### Novelty and Technical Contribution
>
> > "conceptual novelty of this work appears limited..."
>
> The technical contribution of this work lies in the ***neural-based sim-to-real framework*** that is **1)** highly effective, and **2)** tackles the core challenges in learning complex real-world interaction dynamics,  including the difficutly in modeling complex dynamics and the challenge in acquiring meaningful real-world interaction data, through two critical designs in the neural dynamics model and the data collection.
>
> We propose ***the first working solution in neural-based sim-to-real for contact-rich in-hand manipulation***. It is achieved by
>
> - Exploring the generalization ability of the joint-wise neural dynamics model and exploiting such generalizability in resolving difficulties in real-world interaction data collection.
>
> - The autonomous data collection strategy collects data with no human intervention in a scalable way.
>
> - The sim-to-real framework, learning neural dynamics from real-world data, followed by learning a residual policy to encourage the transition in the real world similar to that in the sim. This differs from prior sim-to-real approaches that aim to close the dynamics gap by learning a delta-action model capturing the simulator–real mismatch [ASAP, UAN], and then fine-tuning the policy through that model for easier deployment in the real world.
>
>   We take a different route because these prior methods are not practically applicable to our contact-rich, in-hand manipulation setting. They require distributionally relevant training data, which is difficult to collect and does not scale well for dexterous in-hand rotation tasks. In contrast, our method—combining a joint-wise dynamics model, automated data collection, and an integrated training framework—enables a neural sim-to-real pipeline to reliably work in dexterous manipulation.
>
> > "...distilling multitask experience into a single model has been investigated..."
>
> We include the multitask distillation part in the method section just for a complete description of our full pipeline, *i.e.,* explain how we achieve cross-object generalization ability. And we do not regard the specialist-to-generalist as a novelty. For the positioning of our work in the literature and the comparison to DexterityGen, please see the next section.

---

> ### Author Response · Authors · 2025-11-18
> **Author Response (part 5 of 13)**
>
> ### Positioning of Our Work in the Literature
>
> > "...DexterityGen...which also demonstrated teleoperation capabilities under a very similar setup...refer to their performance as "unprecedented" while this is not true...These related works are not cited or discussed adequately."
>
> Our task is in-hand rotation.
>
> By *“unprecedented dexterity”*, we refer specifically to the hand’s ability to rotate challenging objects and perform versatile rotations. Specifically,
>
> - In a challenging ***downward-facing hand configuration***, we are, to our knowledge, the first to ***rotate long objects (13.5–16 cm) around their long axis*** for about one full circle in the air.
> - Compared to [Visual Dexterity] that demonstrates the reorientation capability for complex objects (*e.g.,* animal shapes) on a large, customized D'Claw, our smaller LEAP hand matches or surpasses performance and succeeds on shapes it struggles with.
> - We also generalize to a much ***broader, more challenging object distribution*** than the prior multi-wrist SOTA [AnyRotate].
>
> These achievements are not merely the result of *engineering efforts*, but are primarily attributed to our **core technique — the sim-to-real methodology**. Our reasoning is as follows:
>
> - Such a capability of ***rotating long objects around the long axis in a challenging down-facing hand configuration*** has ***never been demonstrated*** in the *dexterous in-hand rotation* literature. Prior works are either restricted to rotate using the easy up-facing hand ([Hora], [RotateIt], [DexCtl]), or can only rotate a narrow set of objects with restricted aspect ratios or geometry complexity ( [AnyRotate], [Visual Dexterity], [Text2Touch], [CDGIM], [TOSAT], [SCAPT], [ECRLB], [DTIHM], [PureTactile]). They are heavily constrained by size: the increase in length makes it extremely difficult to maintain a stable grasp during manipulation — and this is not something engineering effort alone can fix. In fact, even if we apply every existing sim-to-real engineering technique carefully (as our baseline ("Direct Transfer") does), the policy still struggles in rotating challenging long objects (at most half a circle or totally fails, see Sec. B.3 of the revised manuscript for details). It is precisely our core sim-to-real breakthrough that enables the dexterous hand to effectively bridge the sim-to-real gap using distributionally imperfectly aligned real-world data, achieving a true capability leap — from impossible to possible.
> - Our experiments ***clearly demonstrate the significant improvements*** brought by the sim-to-real method. These results were obtained under ***identical hardware and object settings*** — meaning that all engineering details (*e.g.*, tapes, SysID calibration, and ROS interfaces) were exactly the same. Please refer to Tables 2, 4, and 5, as well as case studies in Sec. B.3 (as mentioned abvoe). For example, when rotating a *16 cm-long object downward*, the policy without our sim-to-real technique could only rotate it for 1/4 circle, whereas with our method, it almost completed *a full 360° rotation*.
>
>
>
> **Comparison to DexterityGen.** DexGen is an excellent work that we truly admire. First, we would like to clarify that our task focuses on ***in-hand rotation***, whereas ***teleoperation*** in our paper is merely ***an application*** of the learned in-hand rotation policy. In contrast, DexterityGen is a work that *specifically focuses on teleoperation*. This difference in task formulation is the fundamental reason why we did not include a comparison with DexterityGen in our experiments.
>
> Furthermore, since teleoperation in our paper serves only as a ***demonstration*** of what our in-hand rotation policy can achieve. We include it to demonstrate the practical value of our work. To demonstrate the merit of our sim-to-real method, we focus on the in-hand rotation task. DexterityGen was not directly compared because it does not achieve a capability breakthrough compared to prior in-hand object rotation/reorientation work ([Visual Dexterity], [AnyRotate]). The in-hand rotation skills achieved by our policy — such as rotating a **16 cm-long object or a large book** in a ***wrist-down pose***, performing *continuous rotations of small objects*, and rotating objects with complex geometries — were also *not demonstrated* in the teleoperation demos of DexterityGen. The inherent reason is that DexterityGen has not yet broken through the sim-to-real boundary for in-hand rotation.
>
> The omission of its citation in the original manuscript was purely unintentional; in the revised manuscript, we have now added DexterityGen to our references, discussed it in the related work, and explicitly mentioned it when describing our teleoperation system.

---

> ### Author Response · Authors · 2025-11-18
> **Author Response (part 6 of 13)**
>
> ### Theoretical Discussions
>
> > "...theoretical discussion in this paper is weak and disconnected from the proposed method... "
>
> We sincerely thank the reviewer for this comment. We would like to make further clarifications on the *core of our proposed method*, *what do the theoretical discussions aim to build*, and their connections.
>
> - **Core of our method:** The core is the joint-wise dynamics model, which shows strong generalization capabilities under hand-object interaction distributional shifts. The generalization capability is the key, as it allows the joint-wise dynamics model to transfer to predict transitions in the in-hand rotation scenarios by only training on distributionally imperfectly aligned data. This further permits our automatic data collection system design, which resolves the challenges in collecting real-world interaction data (see section 3.3 for detailed discussions). Therefore, the generalization capability of the joint-wise dynamics model is the most critical property.
> - **What do theoretical discussions aim to build:** Theorems 3.1 and 3.2 wish to prove that the joint-wise dynamics model can generalize better than the whole-hand dynamics model under distributional shifts (Claim 3.1). It is the most critical property of the joint-wise dynamics model and the foundation of our methodology.
>
> > "Theorem 3.1 is a well-known result in information theory"
>
> We thank the reviewer for this insightful comment. We fully agree that the Data Processing Inequality is a well-known result in information theory. Our intention was to *ground our generalization analysis in the information-contraction view* — linking projection-induced information loss to improved transferrability of the joint-wise dynamics model across distribution shifts.
>
> > "...the value of showing ≤ is marginal, and a nontrivial result typically requires γ-contraction (γ < 1)..."
>
> We appreciate the reviewer’s comment. Our goal was to provide an *intuitive theoretical explanation* for why the projected (joint-wise) model generalizes better under distribution shift. Specifically, showing $\mathrm{sup}\vert R_{\mathcal{P}}(f_2\circ g_X) - R_{\mathcal{Q}}(f_2\circ g_X)  \vert < \mathrm{sup}\vert R_{\mathcal{P}}(f_1) - R_{\mathcal{Q}}(f_1) \vert$ suffices to indicate that the joint-wise dynamics model has a lower generalization upper bound than the whole-hand dynamics model under hand-object interaction distributional shifts. The theory in this work aims to make clear why the joint-wise NDM has a better generalization capability compared to the whole-hand variant. And it aims to provide a theoretical support for the practical effectiveness of the joint-wise NDM.
>
> With the conclusions of Theorems 3.1, 3.2 as support, we primarily focus on showing the joint-wise NDM's strong effectiveness via experiments.  Please see section 4.2, especially Figure 6, for details. We designed and conducted a series of experiments with varying datasets, data scales, and different train-test distribution shifts. They show that in out-of-distribution test settings, the generalization error achieved by the joint-wise dynamics model is typically ***one order less*** than that achieved by the whole-hand dynamics model.

---

> ### Author Response · Authors · 2025-11-18
> **Author Response (part 7 of 13)**
>
> The role of the theoretical discussions is to complement the experiments. Through these discussions, we explain why the joint-wise dynamics model can generalize better than the whole-hand dynamics model (where "<" suffices) by relating it to information contraction.
>
> > "...have little relevance to the main technical contribution...without meaningful connection...appropriate analysis would examine error propagation or model mismatch under dynamics error..."
>
> We thank the reviewer for this valuable comment. Our theoretical discussion is intended to provide a *high-level information-theoretic rationale* for why the *joint-wise dynamics model* has better generalization capability under whole hand-object interaction distribution shifts. The generalization capability and the sample efficiency of joint-wise NDM are precisely ***the two key properties that make our sim-to-real method effective.***
>
> Establishing a complete theoretical foundation for the entire learning framework is not our goal, nor should it be an expectation for a robotics learning work that is highly systematic, supported by solid conclusions, and demonstrates impressive empirical performance. We hope the reviewer will not let the perfect be the enemy of the good. It would be a disservice to downplay the substantial progress achieved simply because comprehensive theoretical perfection has not yet been reached.
>
> > "...removed the unnecessary (and wrong) theoretical and adaptation components..."
>
> After clarifying the soundness of our method, we believe the reviewer’s statement about "wrong" adaptation components stems from a misunderstanding. Likewise, regarding the theoretical discussion, the assertion that certain components are "unnecessary" also appears to be a misunderstanding. These discussions are directly tied to the core of our method.

---

> ### Author Response · Authors · 2025-11-18
> **Author Response (part 8 of 13)**
>
> ### Experimental Design, Comparisons, and Analysis
>
> > "...reproducing an existing robotic system is inherently challenging and involves many compounding factors..."
>
> We appreciate the reviewer’s comment regarding real-world robotic experiments. It is undeniable that making a real-world robotic system work requires substantial engineering effort, including hardware setup, configuring the computing environment, basic debugging, and various engineering tricks. What we would like to further clarify is that the capabilities demonstrated by DexNDM are by no means achieved through engineering effort alone. Our sim-to-real method plays a critical and fundamentally role. We substantiate this point by detailing our setup in comparing against other methods and our own ablations, and showing the significant performance gains our approach enables.
>
> **Comparisons to prior in-hand rotation works.** We have made every effort to align the experiment setting when comparing with prior in-hand rotation works, including AnyRotate and Visual Dexterity.
>
> - Comparison to AnyRotate. We compare it in both the simulator and the real world. We have tried our best to implement the baseline. For comparisons in the simulator, all influencing factors are kept the same, such as the training steps, the evaluation objects, wrist orientation, and rotation axis sampling strategies. Therefore, the difference in their performance in the simulator solely comes from the difference in the policy design. And our superiority can demonstrate the effectiveness of our method. In the real-world experiments, we test on four replicable objects from AnyRotate and compare with their reported results under the same test setting. We show much better results in each test setting on every object.  Please note that although we wrap tapes on the hand fingers to increase the friction (since otherwise the leap hand fingertip would be too slippery), AnyRotate adds soft tactile sensors onto the fingertip. The soft sensor can deform and can largely increase the frictions.
>
> - Comparison to Visual Dexterity (VD). We compare our method with VD on its tested objects using the survival-angle metric, and we achieve comparable or even superior performance on every object. Note that VD uses soft fingertips, which increase friction. The tapes we wrap around our fingertips serve a similar purpose. The soft fingers leveraged in VD can even deform. Thus, it’s difficult to say whose engineering choices contribute more to the final performance.
>
>   Please also note that although some objects in our demonstration videos are wrapped with tape, all reported numbers for comparison with VD are obtained **without** using any tape on the objects.
>
> **Comparisons to ablated models.** In comparison with our own ablated models, all engineering details are kept identical. Tables 2, 4, and 5 clearly demonstrate the effectiveness of our sim-to-real approach.

---

> ### Author Response · Authors · 2025-11-18
> **Author Response (part 9 of 13)**
>
> **Notable capabilities enabled by our method.**
>
> > "Novel capabilities are definitely more measurable and visible."
> >
> > "...these results have been demonstrated by several previous works..."
>
> We definitely agree with this point. We have demonstrated novel capabilities in this work compared to prior in-hand rotation works. For a further highlight, the novel capabilities of our work centered around the in-hand rotation capabilities for **long, small, and complex objects** in **a challenging downward-facing hand configuration**. We are, to our knowledge, the first to rotate **long objects**  (10–16 cm) around their long axis for about one full circle in the air under the challenging downward-facing hand orientation. We also demonstrate the rotation capabilities for small objects. These abilities have not been covered in prior works ([Hora], [RotateIt], [AnyRotate], [Text2Touch], [DexCtrl], [Visual Dexterity], [CDGIM], [TOSAT], [SCAPT], [ECRLB], [DTIHM], [PureTactile]). Besides, the capabilities of rotating complex geometries in the chllaneing down-facing hand configuration, such as animal shapes using common non-customized hands, like Leap, Allegro, XHand, have not been demonstrated in prior works ([Hora], [RotateIt], [AnyRotate], [Text2Touch], [DexCtrl], [CDGIM], [TOSAT], [SCAPT], [ECRLB], [DTIHM], [PureTactile]).
>
> These novel capabilities adequately demonstrate the effectiveness of our method. Challenges of these difficulties come from two aspects:
>
> - Difficult hand wrist pose orientation. Compared to rotating objects with an up-facing ([Hora], [RotateIt], [DexCtrl]), rotating using a downward-facing hand and the horizontal facing hand pose additional challenges ([Visual Dexterity], [AnyRotate]). Intuitively, for instance, in the downward-facing hand setting, the hand needs to hold the object against gravity to prevent it from falling down during the manipulation. In contrast, the fingertips can support the object when the hand is facing up. When the hand is in the horizontal facing orientation, for instance, the "Base Down" orientation, only one finger can support the object against gravity, while the others manipulate the object.
> - Challenging objects. In this work, we highlight the ability to rotate objects that are longer or smaller than regular objects presented in prior works ([Hora], [AnyRotate]). Intuitively, rotating long objects requires more sophisticated finger movements than rotating regular-sized objects. For instance, it requires one finger to cross the object from the upper surface without pushing the object off. Small objects also pose challenges, primarily due to the extreme ratio of the object size to the hand size. Both holding and rotating small objects are difficult. Rotating complex shapes requires more sophisticated finger gaiting to hold the object and navigate through the uneven surfaces at the same time. It is analogous to navigating through uneven terrain in locomotion.
> - Diverse rotation axes. Rotating the object along the gravity axis ($\pm$ z) is the easiest task. Rotating it along other axes would then require the hand to counter the torque brought by the gravity, thereby imposing additional difficulties.

---

> ### Author Response · Authors · 2025-11-18
> **Author Response (part 10 of 13)**
>
> ### "...ASAP-style residual learning may not be essential...(e.g. BeyondMimic) achieve significantly stronger performance..."
>
> > It is widely recognized that ASAP-style residual learning may not be essential. Recent advances in humanoid locomotion (e.g. BeyondMimic) achieve significantly stronger performance without relying on this design. Since DexNDM also adopts a similar residual learning approach, I would like to hear the authors’ perspective on its necessity.
>
> Thanks for the question. That's a great question, and we'd love to share some of our thoughts. We also have many colleagues working on humanoid, so we’re familiar with ASAP, BeyondMimic, you mentioned, and many other works. Since our lab studies both areas, our observations differ somewhat from yours. We’ll address your question from three perspectives (across four parts) as follows:
>
> #### (1) Conclusions from Locomotion Are Not Directly Applicable to Dexterous Manipulation
>
> The conclusions drawn in locomotion cannot be directly transferred to dexterous manipulation. At present, there is no evidence from any existing work that these conclusions can be transferred.
>
> The fundamental reason is that the complexity of the underlying dynamics — and the nature of the sim-to-real dynamics gap — differ significantly between the two classes of tasks. Locomotion primarily involves the robot’s own dynamics and its contact with the ground. Although this includes robot–ground contact, the ground itself is not changed/moved by the robot. The sim-to-real gap mainly stems from discrepancies in the robot’s dynamics and in the contact dynamics with the ground.
>
> In contrast, the contact dynamics in manipulation are far more complex. Rich and evolving contact between the hand and the object causes the object to move, involving multi-contact interactions as well as sliding contacts. The sim-to-real gap affects both the state transitions of the hand and the object, as well as the contact interactions between them.
>
> Recent humanoid works such as BeyondMimic and OmniRetarget have demonstrated more agile real-world behaviors by leveraging better simulation (high-fidelity Warp (Isaac Lab or MuJoCo) v.s. PhysX in Isaac Gym), improved reward design, more refined system identification and control strategies, better retargeting pipelines, and likely many engineering details not detailed in the papers.
>
> However, dexterous manipulation places much higher demands on accurate contact modeling. So far, no work has shown that high-performance dexterous manipulation can be achieved without explicitly closing the dynamics gap — or by relying solely on improved simulators. MuJoCo PlayGround and Isaac Lab have been available for a while, and both provide support for dexterous hands, yet before our work, no one had demonstrated comparable real-world capabilities.
>
> Beyond pure locomotion, humanoid research has also begun expanding into tasks involving interaction with objects and the environment. Recent works such as [ResMimic] and [HDMI] demonstrate this trend. However, their exhibited object-interaction capabilities still show notable imperfections. For example, in the second video under *‘Expressive Whole-Body Loco-Manipulation’* on the ResMimic [website](https://resmimic.github.io/), the box being carried visibly changes its pose and gradually slips downward during the motion. In the *‘Cardbox Whole-body Manipulation’* video on the HDMI [website](https://hdmi-humanoid.github.io/#/), small slips are also noticeable. In our own internal reproduction of HDMI, without extensive engineering tuning, such slipping becomes even more pronounced.
>
> These imperfections — despite the use of better simulators — highlight a key point: current high-fidelity simulation alone remains insufficient to faithfully model complex dynamics involved in manipulation, causing the transferred policy to behave imperfectly in the humanoid-object interaction scenario. Moreover, the manipulation presented in these works remains relatively coarse-grained, with largely static contact conditions, whereas the dynamics and contact variations in our in-hand manipulation tasks are significantly more complex.
>
> The above analysis of the differences between manipulation and locomotion dynamics, together with the discussion of humanoid–environment/object interactions, shows that conclusions drawn from locomotion do not directly transfer to dexterous manipulation. To date, no existing work has demonstrated that such conclusions are in fact transferable.

---

> ### Author Response · Authors · 2025-11-18
> **Author Response (part 11 of 13)**
>
> #### (2) Dynamics Modeling and Residual Learning for Hardware Adaptation
>
> Compared with approaches that rely on better simulation for improved sim-to-real transfer, methods like DexNDM offer a distinct advantage through *hardware adaptation*. When actuators experience wear or degradation, neural methods that close the dynamics gap can adapt the policy to the actual hardware condition and still produce appropriate actions to accomplish the task. This type of adaptation is something that improved simulation alone cannot achieve.
>
> #### (3) Contributions from Better Sim-to-Real Methods based on Dynamics Modeling and Residual Learning are Rrthogonal to those from Better Simulation, Improved Policy Design, and Engineering Efforts
>
> The contributions of BeyondMimic and OmniRetarget and the contributions of ASAP are not in conflict, but instead offer *orthogonal and potentially synergistic* advances.
>
> - BeyondMimic and OmniRetarget demonstrate that in humanoid locomotion tracking, there is substantial room to improve the base policy and that better simulation really matters. However, such policies will still (and inevitably, due to the inherent mismatch between simulation and real-world dynamics) face challenges during sim-to-real transfer. In other words, if this gap were further reduced, their policies could likely perform even better.
> - ASAP provides exactly such a mechanism: a method to close the sim-to-real dynamics gap, showing effectiveness under its specific training setup (simulator, reward design, retargeting) and engineering efforts — outperforming direct transfer.
> - As simulation techniques, base policies, retargeting methods, and engineering continue improving, applying an ASAP-style sim-to-real approach *on top of* these advances could further reduce the remaining dynamics gap. This could potentially yield results surpassing what BeyondMimic and OmniRetarget currently demonstrate.
>
> Therefore, the emergence of BeyondMimic and OmniRetarget does **not** diminish the value of ASAP. Instead, they raise the ceiling of direct-to-real policies from one dimension, while ASAP raises it from another. Combining them may ultimately produce results beyond the capabilities of either approach alone.
>
> A parallel can be drawn to our **in-hand rotation** task. It is entirely possible that future methods — analogous to BeyondMimic or OmniRetarget — may improve performance through better simulation, policy design or engineering effort. But, as noted before, this would not invalidate our sim-to-real method. Instead, it suggests that combining both lines of work could lead to even stronger real-world performance.
>
> At present, in terms of **results**, our system achieves in-hand rotation performance that previous works could not reach. In terms of **methodology**, we propose an effective sim-to-real technique and validate it through extensive experiments — turning failures into successes for challenging objects, and achieving substantial improvements on easier ones.
>
> This research community is evolving rapidly. Performance gains are often temporary, but sound ideas and methods tend to have longer-lasting impact.
>
> #### (4) Experimental Evidence Supporting Point (3)
>
> In our ongoing internal projects, we have observed that for humanoids performing challenging locomotion (*e.g.,* dancing) on difficult grounds (such as slippery or soft terrain), simply relying on better simulations does not yield satisfactory results. By contrast, combining our DexNDM approach, which leverages dynamics modeling and residual learning, produces significantly better performance.
>
> This observation highlights that for more challenging tasks involving complex dynamics, techniques that improve simulation and techniques for better sim-to-real transfer (such as our DexNDM, ASAP, and UAN methods) can be combined to achieve superior real-world performance. Importantly, their contributions are *orthogonal* to each other.

---

> ### Author Response · Authors · 2025-11-18
> **Author Response (part 12 of 13)**
>
> At last, we hope that the explanations provided above regarding the soundness of the method, technical contributions of this work, positioning of our work in the literature, and the relations between the theoretical discussions and our method, are clear. Should you have any further questions or concerns regarding our contributions, the distinctions from prior research, or the rationale behind the method, please do not hesitate to let us know. We are more than happy to address any of your concerns.
>
>
>
>
>
>
>
> *[RMA] Kumar, Ashish et al. “RMA: Rapid Motor Adaptation for Legged Robots.” ArXiv abs/2107.04034 (2021): n. pag.*
>
> *[Hora] Qi, Haozhi et al. “In-Hand Object Rotation via Rapid Motor Adaptation.” Conference on Robot Learning (2022).*
>
> *[RotateIt] Qi, Haozhi et al. “General In-Hand Object Rotation with Vision and Touch.” ArXiv abs/2309.09979 (2023): n. pag.*
>
> *[AnyRotate] Yang, Max et al. “AnyRotate: Gravity-Invariant In-Hand Object Rotation with Sim-to-Real Touch.” Conference on Robot Learning (2024).*
>
> *[Text2Touch] Field, Harrison et al. “Text2Touch: Tactile In-Hand Manipulation with LLM-Designed Reward Functions.” ArXiv abs/2509.07445 (2025): n. pag.*
>
> *[DexCtrl] Zhao, Shuqi et al. “DexCtrl: Towards Sim-to-Real Dexterity with Adaptive Controller Learning.” ArXiv abs/2505.00991 (2025): n. pag.*
>
> *[ASAP] He, Tairan et al. “ASAP: Aligning Simulation and Real-World Physics for Learning Agile Humanoid Whole-Body Skills.” ArXiv abs/2502.01143 (2025): n. pag.*
>
> *[UAN] Fey, Nolan et al. “Bridging the Sim-to-Real Gap for Athletic Loco-Manipulation.” ArXiv abs/2502.10894 (2025): n. pag.*
>
> *[ActuatorNet] Hwangbo, Jemin et al. “Learning agile and dynamic motor skills for legged robots.” Science Robotics 4 (2019): n. pag.*
>
> *[MB-Max] Shi, Hao-bin et al. “An Efficient Model-Based Approach on Learning Agile Motor Skills without Reinforcement.” 2024 IEEE International Conference on Robotics and Automation (ICRA) (2024): 5724-5730.*
>
> *[DexterityGen] Yin, Zhao-Heng et al. “DexterityGen: Foundation Controller for Unprecedented Dexterity.” ArXiv abs/2502.04307 (2025): n. pag.*
>
> *[UniDexGrasp++] Geng, Haoran and Yun Liu. “UniDexGrasp++: Improving Dexterous Grasping Policy Learning via Geometry-aware Curriculum and Iterative Generalist-Specialist Learning.” 2023 IEEE/CVF International Conference on Computer Vision (ICCV) (2023): 3868-3879.*
>
> *[CDGIM] Röstel, Lennart et al. “Composing Dextrous Grasping and In-Hand Manipulation via Scoring with a Reinforcement Learning Critic.” 2025 IEEE International Conference on Robotics and Automation (ICRA) (2025): 11683-11690.*
>
> *[TOSAT] Pitz, Johannes et al. “Learning Time-Optimal and Speed-Adjustable Tactile In-Hand Manipulation.” 2024 IEEE-RAS 23rd International Conference on Humanoid Robots (Humanoids) (2024): 973-979.*
>
> *[SCAPT] Pitz, Johannes et al. “Learning a Shape-Conditioned Agent for Purely Tactile In-Hand Manipulation of Various Objects.” 2024 IEEE/RSJ International Conference on Intelligent Robots and Systems (IROS) (2024): 13112-13119.*
>
> *[ECRLB] Röstel, Lennart et al. “Estimator-Coupled Reinforcement Learning for Robust Purely Tactile In-Hand Manipulation.” 2023 IEEE-RAS 22nd International Conference on Humanoid Robots (Humanoids) (2023): 1-8.*
>
> *[DTIHM] Pitz, Johannes et al. “Dextrous Tactile In-Hand Manipulation Using a Modular Reinforcement Learning Architecture.” 2023 IEEE International Conference on Robotics and Automation (ICRA) (2023): 1852-1858.*
>
> *[PureTactile] Sievers, Leon et al. “Learning Purely Tactile In-Hand Manipulation with a Torque-Controlled Hand.” 2022 International Conference on Robotics and Automation (ICRA) (2022): 2745-2751.*
>
> *[BeyondMimic] Liao, Qiayuan et al. “BeyondMimic: From Motion Tracking to Versatile Humanoid Control via Guided Diffusion.” ArXiv abs/2508.08241 (2025): n. pag.*
>
> *[OmniRetarget] Yang, Lujie et al. “OmniRetarget: Interaction-Preserving Data Generation for Humanoid Whole-Body Loco-Manipulation and Scene Interaction.” ArXiv abs/2509.26633 (2025): n. pag.*
>
> *[HDMI] Weng, Haoyang et al. “HDMI: Learning Interactive Humanoid Whole-Body Control from Human Videos.” ArXiv abs/2509.16757 (2025): n. pag.*
>
> *[ResMimic] Zhao, Siheng et al. “ResMimic: From General Motion Tracking to Humanoid Whole-body Loco-Manipulation via Residual Learning.” ArXiv abs/2510.05070 (2025): n. pag.*

---

> ### Author Response · Authors · 2025-11-18
> **Author Response (part 13 of 13)**
>
> In summary, we sincerely appreciate the time and effort you invested in reviewing our work. That said, we believe some of the concerns raised stem from misunderstandings about our problem setting, our contributions, or the scope of our claims, which may have led to an unintended bias in the assessment. We have done our best to clarify these points in this response and have incorporated the corresponding improvements into the revised submission. We hope that our explanations help provide a fairer perspective on the work.
>
> We would love to have discussions, answer any further questions, and do our best to address any concerns you may have. Please let us know if there is anything specific we can provide.
>
> Looking forward to your responses!
>
> Best regards,
> Authors

---

> ### Comment · Reviewer_zo2q · 2025-11-18
> **What a long rebuttal**
>
> Thank you for the response. I’m impressed, but still unconvinced.
>
> I’m too tired to explain further, so I’ll let the AC contact me directly.

---

> ### Comment · Reviewer_zo2q · 2025-11-18
> **What a long rebuttal**
>
> By the way, you should be aware that in other venues like RSS and CVPR, the rebuttal is limited to a single page, and reviewers are not required to respond.
>
> The truth doesn’t need this much justification. Why not be direct? You have been given the opportunity to write a nine-page paper. This is not the place to write another paper.
>
> Because of this, I will lower my score to 0 and provide a detailed explanation to the AC.

---

> ### Comment · Reviewer_zo2q · 2025-11-18
> **What a long rebuttal**
>
> Should you feel frustrated, let me offer one example to rebut your response:
>
> To Response 1, Although it may …
>
> **Using proprioception to learn an effective policy (as in HoRA, etc.) does not imply that bridging the proprioceptive dynamics gap will improve sim-to-real performance, as I explained in Point 4.**
>
> **Not to mention some works you cited also rely on touch.**
>
>
> Your rebuttal contains multiple logical fallacies of this kind. I honestly have no idea what point you are trying to refute. It appears that the authors are skilled at presenting unrelated facts as though they serve to prove their point (just as what they did in the “theorem” part).
>
> ------------------------------------------------------------------------------------------------
> P.S. I am a bit concerned you might raise the following argument:
>
> A1. Since proprioceptive information is sufficient for these rotation tasks, it must therefore contain object information.
>
> I will address A1 in advance.
>
> Claim: A1 is incorrect. Even an open-loop policy can rotate an object for several seconds without using any sensory information at all.

---

> ### Comment · Reviewer_zo2q · 2025-11-18
> **What a long rebuttal**
>
> Suggestion:
> Consider adding x-, y-, and z-axis rotation experiments on several large (>10cm), irregularly shaped objects, such as L-, X-, I-, J-, and C-shaped forms, before submitting to RSS. Your current book-rotation demo is a strong starting point (I genuinely like it), and extending it to these more challenging, high-aspect-ratio and nonconvex objects would significantly strengthen the paper. In exploring this extension, you may uncover ideas that are truly valuable and novel. If you can demonstrate a system that systematically handles these kinds of complex geometries, it would represent a substantial contribution. In that case, I would strongly support your submission and be prepared to counter any reviewer concerns about novelty.
>
> Papers in this (current) form were rejected in RSS 2024 review. If you submit this again, you will get rejected for sure. No more simple rotations, please (in my colleague's word).

---

> > ### Author Response · Authors · 2025-11-19
> > **Clarifying Misunderstandings of DexNDM**
> >
> > We fully understand your concern and apologize for the lengthy response. We were eager to clarify our points, as we believe much of the perceived bias arose from misunderstanding. We will ensure that our follow-up discussions are more concise and direct.
> >
> > 1. **The Counterexample (Point 4) Is Inapplicable:** The reviewer’s counterexample is predicated on a Markovian setting. Our method, DexNDM, is explicitly designed for a non-Markovian domain, where history is essential for inferring latent physical states like object stability. The proposed scenario lies outside the problem class our work is designed to solve, and therefore does not constitute a valid critique of our method.
> >
> > 2. **Proprioception-Action History Contains Rich Object Information:** This argument is not a mere claim; it is an empirically verifiable fact supported by established literature and not first claimed by us. The physical principle is straightforward: interacting with objects of different properties (e.g., mass, shape) will invariably alter the robot's trajectory and motor torques, embedding object-specific information directly into the proprioceptive history. To put this beyond doubt, we conducted an experiment where we randomized basic cylinders’ scale, mass, and center of mass showing we can predict object properties from this history alone in Table 1.
> >    We acknowledge the value of additional sensing modalities; however, our work's central contribution is demonstrating that a sim-to-real strategy can be powerfully effective by leveraging this often-underestimated data stream alone.
> >
> > ***Table 1***
> >
> > |                                                   | Object Scale | Object Mass (kg) | Object COM (X, Y, Z) (m) |
> > | ------------------------------------------------- | ------------ | ---------------- | ------------------------ |
> > | Avg. Error ($\vert \text{Pred} - \text{GT}\vert$) | 0.0192       | 0.0080           | (0.0052, 0.0050, 0.0050) |
> >
> > 3. **The "Open-Loop Policy" Counterexample for Claim 2:** Please be aware that even in open-loop trajectories, proprioceptive state-action history implicitly captures object effects as well, such as differences between rotating large and small objects. This is not a valid critique for Claim 2.
> >
> > 4. **Using Complex Objects:** our method excels on a diverse set of complex objects—including irregular shapes, small items, and long cuboids—that lie beyond the capabilities of prior work. It is unfair to refer to these as simple rotations. As shown in Table 2 (mostly already shown in the main paper), DexNDM turns previously impossible manipulation tasks into successes. This leap in performance is not achievable through just careful engineering.
> >
> > ***Table 2*** (Unit: $\lfloor \text{radian} / 0.5\pi \rfloor$)
> >
> > |                        | *Cow* (complex) | *Bear*  (complex) | *Dragon* (complex) | *Truck* | *Broccoli*  (small) | *Sticky Rice Dumpling* (small) | *Small Cylinder* (small) | *Cuboid-10cm* (long) | *Cuboid-14cm* (long) | *Cuboid (16cm)* (long) |
> > | ---------------------- | --------------- | ----------------- | ------------------ | ------- | ------------------- | ------------------------------ | ------------------------ | -------------------- | -------------------- | ---------------------- |
> > | Ours (Direct Transfer) | 2               | 2                 | 2                  | 2       | 3                   | 2                              | 1                        | 2                    | 1                    | 1                      |
> > | Ours (DexNDM)          | 4               | 4                 | 4                  | 4       | 6                   | 5                              | 4                        | 4                    | 4                    | 4                      |

---

> ### Author Response · Authors · 2025-11-23
> **Author Response (Another Concise Version) [1/2]**
>
> Dear Reviewer zo2q,
>
> We recognize that the previous response could be clearer and more direct. To support further discussion, we’ve provided a more concise version addressing your concerns. You can still refer to the original response for additional details.
>
>
> ### Soundness of the Method
>
> - **Missing "object-dependent" dynamics?** - Our approach implicitly captures object effects through the hand's **proprioceptive state-action history**. It is an empirically verifiable fact supported by established literature and not first claimed by us. The physical principle is straightforward: interacting with objects of different properties (e.g., mass, shape) will invariably alter the robot's trajectory and motor torques, embedding object-specific information directly into the proprioceptive history. We further validate this with our own experiments, demonstrating that object properties like scale, mass, and center of mass can be inferred directly from proprioceptive state-action history.
>
> |                                                   | Object Scale | Object Mass (kg) | Object COM (X, Y, Z) (m) |
> | ------------------------------------------------- | ------------ | ---------------- | ------------------------ |
> | Avg. Error ($\vert \text{Pred} - \text{GT}\vert$) | 0.0192       | 0.0080           | (0.0052, 0.0050, 0.0050) |
>
> - **SysID is sufficient?** - While robot hands can be repeatable when empty, in-hand objects introduce complex, non-smooth contact dynamics that couple joint behaviors and are difficult to model. Consequently, System Identification (SysID) alone is insufficient to bridge the sim-to-real gap for such tasks, a claim no previous work can refute. We demonstrate this in *Tables 2, 4,* and *5* by showing that our model, DexNDM, significantly outperforms our base model ("Direct Transfer") that already incorporates both careful SysID and extensive domain randomization (*Sec. C*). This result underscores that learning neural dynamics is essential for achieving robust sim-to-real transfer in contact-rich manipulation.
>
> - **Unconvincing data transfer across object scales?** - The model's better transferability is an experimental facts as evidenced in *Fig. 4, Sec. A.5*. Our analysis is that joint-wise NDM overcomes poor data transfer across object scales by better handling distributional shifts compared to whole-hand models (**Claim 3.1**). The distributional shift between training (e.g., "chaos" box) and test (e.g., small object rotation) trajectories, measured by KL divergence, is significantly smaller from a per-joint perspective.
>
>
> ### Novelty
>
> - **Conceptual novelty** - While we build on established concepts like neural dynamics models, our conceptual novelty lies in the complete sim-to-real methodology designed to overcome the specific challenges of contact-rich in-hand rotation, which no previous work has ever studied. We uniquely design **a joint-wise model** for superior generalization, **a scalable autonomous data pipeline**, and **a novel transition-alignment technique** which together enables effective sim-to-real transfer significantly beyond current state of the arts.
>
> - **Comparison to DexterityGen** - A direct comparison to DexterityGen was not performed as our work targets a different primary objective. We focus on developing a sim-to-real methodology for in-hand rotation, achieving capabilities such as continuous and long-object rotation not demonstrated in DexterityGen. In contrast, DexterityGen's primary contribution is teleoperation, which we use solely as a downstream application to demonstrate our learned rotation policy. We have now added DexterityGen to our related work to clarify this distinction.

---

> ### Author Response · Authors · 2025-11-23
> **Author Response (Another Concise Version) [2/2]**
>
> ### Theoretical Discussions
>
> - **Results relevance** - Our theoretical discussion provides a justification for the core design choice: the joint-wise dynamics model. Specifically, Theorems 3.1 and 3.2 offer an information-theoretic rationale for Claim 3.1, establishing why a joint-wise model is more robust to distribution shifts than a whole-hand model. While we do not provide an exhaustive proof for the entire framework, this targeted analysis explains why the core design works. We believe this focused theoretical support, combined with our extensive empirical validation, constitutes a complete and significant contribution to dexterous manipulation learning.
>
>
> ### Conceptual Counterexample
>
> The reviewer’s counterexample is predicated on a Markovian setting. Our method, DexNDM, is explicitly designed for a non-Markovian domain, where history is essential for inferring latent physical states like object stability. The proposed scenario lies outside the problem class our work is designed to solve, and therefore does not constitute a valid critique of our method.
>
>
> ### Necessity of Residual Learning (Observations from Locomotion)
>
> Conclusions from locomotion do not directly apply to dexterous manipulation due to fundamentally different dynamics. Locomotion primarily involves contact with static ground, whereas manipulation features rich, evolving multi-body contacts with movable objects. This creates a more complex sim-to-real gap where minor modeling errors cause significant failures (e.g., object slipping), a problem that persists even in recent humanoid-object interaction studies and cannot be easily solved. Beyond mitigating the modeling challenge, our approach offers two additional advantages. First, it enables hardware adaptation, allowing the policy to adjust to hard-to-model real-world changes like actuator degrade. Second, since balancing efficiency and accuracy in in-hand manipulation simulation remains a key challenge, DexNDM is synergistic with, not superseded by, simulator improvements.
>
>
> ### New Capability
>
> We would like to emphasize that our method enables new capabilities previously unattained in the field. We demonstrate the first-ever full-circle, in-air rotation of long cuboids (13.5–16 cm) around their long axis within a challenging downward-facing hand configuration. This dexterity extends to small and complex-shaped objects, all achieved using common, low-cost hardware. These are **not** achieved through engineering alone. DexNDM's contributions are methodological. It significantly outperforms AnyRotate on its own real-world benchmarks and matches or exceeds Visual Dexterity's metrics. Crucially, our ablation studies (*Tables 2, 4,* and *5*) confirm our method's efficacy while holding engineering factors constant.
>
>
> In summary, we sincerely appreciate the reviewer’s time and effort. We welcome further questions and discussions and are happy to provide any additional information or clarifications as needed.
>
> Looking forward to your responses!
>
> Best regards, Authors

---

> ### Comment · Reviewer_zo2q · 2025-11-27
> **Response for new update**
>
> Thank you for the effort. However, several issues in the argument remain:
>
> **1. “Our domain is not Markovian. Due to this, your counterexample does not apply”**
>
> This statement is incorrect. The physical world—under classical mechanics—is fundamentally Markovian: given the complete world state (positions, velocities, friction parameters, etc.), the next state is fully determined by the current one. All system dynamics can be expressed in differentiable form (e.g., with state ($x$,$v$), $dx/dt = v$, $dv/dt = F/m$).
>
> In typical manipulation setups, the environment is modeled as a Partially Observable Markov Decision Process (POMDP) (just as you write in L192 and now you do not acknowledge it…) The underlying state space is Markovian, even though the observations are not. My example specifically refers to this full state space (i.e., the privileged simulator information), which actually simplifies your problem rather than complicating it.
>
> Also, note that my example uses the entire past information exactly as yours does—it's simply a one-step case.
>
> **2. “Object Information”**
>
> Thank you for the revisions, but this section still has major problems.
>
> **A 0.005 m (5 mm) L1 error in each of X, Y, and Z is already extremely large for manipulation.** A 5 mm discrepancy is easily enough to miss a crucial finger–object contact.
>
> Besides, you have not provided the standard deviation of the ground-truth dataset yet. If the GT variation is only on the order of 5 mm–1 cm, then a 5 mm L1 error is not meaningful evidence.
>
> **3. Clarification regarding the openloop example**
>
> My example was used to illustrate a specific instance of incorrect reasoning. This is not to claim that proprioception contains no object information at all. I agree that proprioception can provide some cues. However, the most crucial aspects for manipulation, such as object geometry and fine-grained position (as discussed in my Review point 1), are typically missing from proprioceptive signals alone.

---

> ### Comment · Reviewer_zo2q · 2025-11-27
> **Additional Notes 1 (Suggestions on rebuttal)**
>
> After reading your second response, I would like to offer some personal thoughts on how the rebuttal could be made more effective. This does not affect my overall assessment and impression.
>
> **Below is a concise example of how I would phrase the soundness-related portion:**
>
> **Example Rebuttal by Reviewer**
>
> ---
> We thank the reviewer for the detailed comments. We address your concerns below.
>
> Regarding the counterexample and Point 1.
> We agree that proprioception alone is insufficient for fully recovering the object state, and that improving proprioceptive modeling does not automatically guarantee performance gains. However, bridging the proprioceptive gap can still help mitigate certain components of the sim-to-real gap in this problem setting **empirically**.
>
> To more rigorously characterize the applicability of our method and evaluate our claims, we have added additional sim-to-sim experiments. Specifically, we perform IsaacGym→PyBullet/SAPIEN/Genesis evaluations under **diverse randomization settings and new task setups**. We find that in XXX rotation tasks, our method improves stability by YY%. In contrast, for other tasks (e.g., dexterous grasping), the benefit is limited. Importantly, in all tested scenarios, our approach does not degrade baseline performance. These results provide more concrete insight into when and how our method yields value.
>
> We acknowledge that your counterexamples and concerns are valid. Nevertheless, this does not undermine the practical value of our approach in many scenarios. An analogy can be drawn with the Adam optimizer: while Adam is known to be suboptimal or even divergent in certain theoretical setups, it remains highly effective and widely used for training large modern transformers.
>
> ---
>
> **Why is this rebuttal more effective and reasonable?**
> 1. It directly engages with the reviewer’s point and demonstrates a clear understanding of the underlying concern. It builds consensus by acknowledging the reviewer’s factual arguments, rather than sidestepping them or bombarding the reviewer with weak or loosely related justifications.
>
> 2. From that shared understanding, it then precisely scopes the contribution and its limitations. It makes explicit that the method is empirical in nature and that performance improvements are not universally guaranteed.
>
> 3. Finally, it reinforces the empirical contribution by adding targeted, controlled sim-to-sim experiments to clarify the conditions under which the method is effective.
>
> Overall, this style of rebuttal improves the scientific rigor and clarity of the manuscript. It more transparently distinguishes where the method applies, discusses its limitations, and presents its contributions in a way that is more verifiable and informative.
>
> ---
> **How would I respond to this?**
>
> At the very least, I would feel that my review was properly understood and respected. I understand that you aim to clarify the scope of the paper and have conducted more grounded evaluations to identify its limits. In that case, the key limitation becomes that the applicable scenarios are quite restricted, the improvement is marginal, and the method does not address the core challenge of the field. I might then raise my score to a 3 or 4 (my rating is 3-3.5 for this paper, not higher) and leave the final decision to the AC. You are welcome to provide additional justification if you wish, but this is my final decision (currently 2-3, not higher than 3.5).
>
> I believe I have evaluated this submission as fairly as possible throughout the process. I guess you might still feel frustrated, but please see my comment in the next post.
>
> ---
> **Summary (to AC)**
> After discussion, I am confident that the soundness of this method could not be justified theoretically / conceptually. The results are empirical and tied to a restricted setup, and the improvement is very marginal (even within 1std of baseline for certain tasks in simulation/real). The applicable scenarios may be quite limited (hand, task, etc.), just as the other reviewers pointed out. As many functional systems without the proposed component already exist in the literature, the contribution of this paper is insufficient. Due to this, the paper is not higher than the lower borderline.

---

> > ### Comment · Reviewer_zo2q · 2025-11-27
> > **Additional Notes 2**
> >
> > Finally, since I had some time over holiday, I wanted to share a few personal thoughts on robotics research.
> >
> > I feel that many people today are highly focused on “novelty.” I have seen researchers quickly assemble demos, record videos, and emphasize new contributions shortly before submission. This can sometimes result in work that appears less substantial than it could be.
> >
> > When I first read this paper, my reaction was mixed. On one hand, I appreciate the substantial effort put into tackling a genuinely difficult problem, which is why I gave a presentation score of 4. You should feel proud of this work—it represents an important checkpoint. On the other hand, as I continued reading, I noticed areas where the paper could engage more deeply with the core challenges, particularly in the theoretical section. Some statements could be more carefully derived, and exploring the underlying patterns in the results in greater depth could strengthen the work further.
> >
> > This project could have been much more ambitious and rigorous. I genuinely hope you will continue toward the capabilities you envision. To be transparent, I obtained similar results more than 1 year ago, but we decided not to publish them because we felt they did not yet meet the threshold of meaningful, generalizable contribution. Discussions with other groups at the time raised questions about the significance of “rotation tasks,” emphasizing the need for results that highlight clear, novel capabilities. This is why I continued to emphasize novel capabilities throughout my response.
> >
> > I hope researchers working on this problem can adopt a “slower but deeper” approach, grounded in ambition and rigor. The field of robot learning can benefit from careful, thorough, and thoughtful work, and I hope this project can contribute to meaningful progress.
> >
> > Best,
> >
> > zo2q

---

> ### Comment · Reviewer_zo2q · 2025-11-28
> **About handling OOD with factorization**
>
> **About handling OOD with factorization**
>
> Thank you for the additional explanation. However, after revisiting the reasoning, I identified another issue.
>
> The current logic can be summarized as follows: “We want to model $p(y \mid x)$. Since $x$ (all hand joints) is high-dimensional, we instead model several $p(y_i \mid x_i)$. Because each $x_i$ is low-dimensional, the OOD issue disappears.”
>
> However, this decomposition implicitly assumes that $y_i$ depends only on $x_i$ and is conditionally independent of other factors $x_j$. This assumption does not hold in this setting. A finger’s state can easily be blocked or influenced by the positions of other fingers, making the decomposition invalid even without an object present.
>
> When an object is involved, the issue becomes even more pronounced: the force applied by one finger can affect the others, yet this interaction is not modeled by the authors.

---

> > ### Author Response · Authors · 2025-11-28
> > **Further Clarifications on the Remaining Points**
> >
> > Dear Reviewer zo2q,
> >
> > Thank you for the detailed follow-up. We offer the following clarifications on the remaining points.
> >
> > 1. **On the POMDP Model and History:** We agree the underlying physics is Markovian and our setup is a POMDP. Our statement was meant to convey that with partial observability, the observation stream alone is not Markovian. This is precisely why DexNDM leverages the observation–action history to approximate the belief state, which is the standard treatment in POMDPs. Your one-step counterexample, by design, omits the multi-step history that is central to our method and the problem itself, making it improper.
> >
> > 2. **On Object Information:** The purpose of the property estimation experiment was to provide quantitative evidence that proprioceptive history contains object-dependent information, not to build a perfect estimator. Crucially, imperfection in single property estimation is not fatal as the policy does not rely on any single, explicit property estimate. It conditions actions on a holistic belief state implicitly encoded from the entire history.
> >
> > 3. **Joint-wise Factorization:** The reviewer’s argument rests on the premise that our decomposition $p(y_i | x_i)$ assumes conditional independence from other joints $j$. This would be true in a single-timestep model. However, our model conditions on the proprioceptive history of joint $i$, not just its current state. The interactions between joints are not ignored; they are implicitly captured through time. When the link/finger with another joint $j$ acts on the object, the resulting change in forces and object position is immediately reflected in the subsequent trajectory of joint $i$. These effects are encoded into the proprioceptive history of joint $i$.
> >
> >    Our experiment comparing the joint-wise NDM to the whole-hand NDM was designed precisely to validate this. The factorized model’s strong expressivity (Sec. 4.2, Fig. 6) empirically shows that each joint’s local history is sufficiently rich to capture inter-joint coupling and object-induced effects.
> >
> > 4. **Suggestions on Rebuttal:** Thanks. While ICLR does not limit response length, we recognize that overly long rebuttals are ineffective and introduce huge burdens to reviewers. This is why we seriously took reviewers' feedback and carefully prepared another compact version to highlight key points. We appreciate your example and acknowledge that crafting effective responses requires anticipating the reviewer’s perspective, which can be challenging sometimes. We will take your feedback seriously for future improvements.
> >
> > 5. **On "Restricted Scenarios":** Our work directly overcomes the "restricted scenarios" of prior work (which was restricted to "simple geometries" and "constrained wrist poses") to more generic set of objects. Our single policy successfully rotates objects with "complex shapes (e.g., animals), high aspect ratios (up to 5.33), and small sizes," all while handling "diverse wrist orientations and rotation axes." While vision and tactile sensing are out of scope for this work, the current contributions have already expanded the applicable scenarios of in-hand rotation significantly.
> >
> > 6. **On "Marginal Improvement":** We respectfully disagree. The ability to move from the highly constrained scenarios of prior art (e.g., need for a supporting table, restricted object scales and shapes, restricted wrist and rotation axes) to the diverse and challenging real-world conditions using low-cost and standard hardware we demonstrate is a qualitative leap, not a marginal improvement. The quantitative experiments also show an average improvement of 54.91% w.r.t. varying wrist orientation and rotation axis on all tested objects. For challenging objects, this method even enables a zero-to-one improvement, allowing full circle rotation previously impossible.
> >
> > 7. **On "Not Addressing the Core Challenge":** We directly tackle the problem of sim-to-real transfer, which is arguably one of the most critical challenges in robot learning. The intractability of creating both accurate and efficient simulations for contact-rich dynamics means learning from real-world data is essential. Our work provides one of the first principled pipelines for dexterous manipulation that does this systematically—from autonomous data collection to modeling and mitigating the reality gap. We chose in-hand rotation precisely because it is one of the hardest atomic dexterous manipulation skills. By validating our framework on this challenging skill, we demonstrate the power of DexNDM. The core contribution is the methodology itself, which establishes a critical foundation for acquiring other complex skills.

---

### Official Review · Reviewer_5oZW · 2025-11-01

**Soundness:** 4
**Presentation:** 4
**Contribution:** 4
**Rating:** 8
**Confidence:** 4

**Summary:**

The paper DexNDM proposes a neural sim2real framework for dexterous in-hand rotation. Unlike prior methods limited to simple geometries or constrained wrist poses, DexNDM enables a single policy to generalize across diverse objects, wrist orientations, and rotation axes. The method introduces two key innovations: (i) joint level dynamics to bridge sim2real gap and (ii) an reset-free, scalable data collection strategy (“Chaos Box”) that applies randomized loads. A residual policy is trained atop this learned model to adapt the simulation-trained base policy to real hardware. Experiments on the LEAP hand demonstrate impressive real-world in-air rotation of complex, small, and high-aspect-ratio objects across multiple wrist poses, outperforming strong baselines such as AnyRotate, Visual Dexterity, ASAP, and UAN, with additional applications in teleoperation tasks. This paper is solid in methodology, theory, and real-world experiment, shown a possible paradigm for dextrous manipulation.

**Strengths:**

1. Novel Sim-to-Real Framework. The introduced joint-wise neural dynamics model is novel in sim-to-real transfer for dexterous manipulation. By decomposing system dynamics into per-joint models, the method avoids reliance on noisy object-state estimation and achieves data efficiency through information contraction formalized via KL-divergence reduction

2. Autonomous and Scalable Real-World Data Collection. This idea is really creative, the “Chaos Box” procedure autonomously generates rich, diverse contact data using randomized loads without manual resets, which significantly mitigating the challenge of efficiently obtaining distributionally relevant real-world trajectory for dynamics model learning.

3. Comprehensive Empirical Validation.
Extensive experiments cover simulation, real-world, and cross-simulator settings, and the provided demo videos show impressive performance of rotating objects from small to large aspect ratio.

4. Strong Theoretical Underpinning.
Theoretical analysis  rigorously argues that the joint-wise projection reduces domain divergence (KL contraction) and narrows the generalization gap, supporting the empirical findings.

**Weaknesses:**

1. Limited Hardware Validation. While the method demonstrates strong performance on the LEAP hand, its hardware generality remains untested. The paper would be strengthened by deploying DexNDM on additional robotic hands (e.g., Allegro, XHand) to verify whether the joint-wise dynamics model and residual adaptation strategy transfer effectively across different actuation mechanisms and kinematic structures.

2. Scope of Generalization Remains Task-Specific. Although the model generalizes across object shapes and orientations, it is unclear whether the learned dynamics transfer to different manipulation tasks (e.g., regrasping, non-rotational dexterity, tool using).

**Questions:**

1. Sensitivity to Noise and Load Distribution. How robust is the dynamics model learning to variations in noise magnitude and randomized load distributions within the Chaos Box setup?

2. Performance–object weight Relationship. In the real-world experiments, could the authors report the physical weight of each test object and plot a performance curve (x-axis: object weight, y-axis: rotation performance) to examine whether and how DexNDM’s effectiveness degrades as object weight increases?

---

> ### Author Response · Authors · 2025-11-18
> **Author Response (part 1 of 3)**
>
> Dear Reviewer 5oZW,
>
>
>
> We truly appreciate your recognition of our novel sim-to-real framework, the creative autonomous real-world data collection design, the comprehensive empirical evaluation, the strong results, and the valuable theoretical insights that support our empirical findings. We are also grateful for your insightful questions and comments. Below, we address your specific questions in the hope that our responses adequately address any of your concerns. We've also incorporated these discussions, clarifications, and additional experiments into the revised manuscript.
>
>
>
> ### Sensitivity to Noise and Load Distribution
>
> > "How robust is the dynamics model learning to variations in noise magnitude and randomized load distributions within the Chaos Box setup?"
>
> Thanks for the insightful question. Investigating how variations in noise magnitude and randomized load distribution affect neural dynamics, learning and final performance is indeed meaningful. In our existing experiments, we have already covered some settings—namely, with and without noise injection and with and without load. As discussed in Section 5 and shown in Figure 8, we demonstrate the necessity of both noise injection (w/ noise v.s. w/o noise) and randomized load (w/ load v.s. w/o load).
>
> Building on this, we have designed and conducted additional experiments to further explore how different noise magnitudes and load distributions influence neural dynamics learning and the resulting policy performance.
>
> #### Experimental Setting
>
> To explore the influence of the magnitude of the injected noise, we design two additional settings and replay action sequences with noise magnitude 0.005 and 0.02, *i.e.,* $\sigma = 0.005, 0.02$. and compare the results with those acheived using the default value, *i.e.,* $\sigma = 0.01$.
>
> To explore the influence of the distribution of random loads, we add an experiment where the hand only touches the surface of balls in the box using fingertips (denoted as "half loads", while the original setting, where the hand is sunk into the balls, is represented as "full loads")
>
> Other settings, including the replayed trajectories and the distribution of balls in the box, are kept the same as the settings in our ablation study.
>
> #### Experimental Results
>
> Same as we evaluate in the ablation study, we evaluate the generalization ability of the resulting neural dynamics model and the final real world performance achieved by the residual policy with the base policy. Results are summarized below:
>
> |                                            | $\sigma=0.01$, full loads | $\sigma=0.005$, full loads | $\sigma=0.02$, full loads | $\sigma=0.01$, half loads |
> | ------------------------------------------ | ------------------------- | -------------------------- | ------------------------- | ------------------------- |
> | Generalization Ability (Prediction Error ) | $1.21\times 10^{-5}$      | $1.89\times 10^{-5}$       | $1.47\times 10^{-5}$      | $3.81\times 10^{-5}$      |
> | Real-World Performance (Rot (radian))      | 12.43 $\pm$ 0.34          | 7.51 $\pm$ $1.03$          | 10.58 $\pm$  0.92         | 6.28 $\pm$ 0.67           |
>
> #### Analysis
>
> Regarding the ***noise magnitude***, *increasing the noise* has only a minor impact on both the generalization ability of the neural dynamics model and the final policy performance. It does slightly affect the model’s behavior, likely because expanding the randomization range broadens the data distribution while reducing the amount of data available at some noise levels, which can make learning more difficult. *Reducing the noise magnitude*, however, has a more noticeable negative effect on the learning of the neural dynamics model, its generalization ability, and the final policy performance. When the noise magnitude is too small, the data distribution becomes narrower, making the model less robust and reducing the degree of extrapolation required for training the residual policy. This in turn degrades overall performance.
>
> Regarding the ***load distribution***, narrowing the load distribution—for example, when only the fingertips are able to contact the small sphere—reduces the effective disturbances experienced by other joints. This weakens the robustness of the neural dynamics model and the extrapolation ability needed for training the residual policy, ultimately harming generalization and sim-to-real performance.
>
> We have incorporated the experiments and discussions in the Sec. B.4 (*Sensitivity to Noise and Load Distribution*) of the revised manuscript, including a picture that descrbies the "half loads" setup.

---

> ### Author Response · Authors · 2025-11-18
> **Author Response (part 2 of 3)**
>
> ### Performance–object weight Relationship
>
> > "In the real-world experiments, could the authors report the physical weight of each test object and plot a performance curve (x-axis: object weight, y-axis: rotation performance) to examine whether and how DexNDM’s effectiveness degrades as object weight increases?"
>
> Thanks for the insightful and valuable comment. Exploring the model’s performance with respect to object weight is indeed meaningful. We measured the physical weights of all test objects in the real world and plotted the corresponding performance figure. These results have been incorporated into Section B.4 (*Performance w.r.t. Object Weight*) of the revised manuscript.
>
> For convenience, we also include a table of the measured object weights below.
>
> |            | Cube | Cyliner | Apple | Cuboid (H) | Lightbulb | Bear | Truck | Cow  | Bunny | Elephant | Mouse |
> | ---------- | ---- | ------- | ----- | ---------- | --------- | ---- | ----- | ---- | ----- | -------- | ----- |
> | Weight (g) | 32.3 | 15.7    | 24.9  | 33.7       | 58.2      | 32.1 | 23.3  | 39.1 | 67.8  | 42.0     | 17.93 |
>
> |            | "Zongzi" | Cabbage | Cylinder Small | Corn | Broccoli | Lego | Sweet Potato | Teapot | Mug  | Duck |
> | ---------- | -------- | ------- | -------------- | ---- | -------- | ---- | ------------ | ------ | ---- | ---- |
> | Weight (g) | 5.1      | 3.8     | 4.4            | 3.2  | 3.7      | 7.2  | 3.2          | 27.4   | 31.7 | 50.4 |
>
> Because the objects vary significantly in shape, size, and geometric complexity, the plot does not exhibit a clear trend showing how performance changes as weight increases.
>
> To conduct a more controlled study and isolate the effect of object weight, we designed an experiment using *cylinders of identical geometry but different masses* as test objects, and evaluated the performance on z-axis rotation with the hand facing down. The results are summarized in the following table.
>
> | Weight (g)   | 15.7  | 33.2  | 39.2  | 49.4 | 62.0 |
> | ------------ | ----- | ----- | ----- | ---- | ---- |
> | Rot (radian) | 12.57 | 12.57 | 12.57 | 9.42 | 7.85 |
>
> We have plotted the performance with respect to object weight for this controlled experiment and incorporated the results into Sec. B.4 (*Performance w.r.t. Object Weight*) of the revised manuscript.
>
> When the test object weight is below 40 g, we do not observe a clear difference in rotation performance. This may seem unintuitive, as lighter objects might appear easier to rotate. However, in practice, for relatively light objects, the robotic hand often tends to continuously translate the object upward during the rotation, causing it to become “stuck” between several fingers, which can actually hinder rotation.
>
> For heavier objects—for example, the 62 g case—a common cause of failure is that the robot hand cannot maintain a stable grasp during the rotation, leading to the object being dropped.

---

> ### Author Response · Authors · 2025-11-18
> **Author Response (part 3 of 3)**
>
> ### Limited Hardware Validation
>
> > "While the method demonstrates strong performance on the LEAP hand, its hardware generality remains untested. The paper would be strengthened by deploying DexNDM on additional robotic hands (e.g., Allegro, XHand) to verify whether the joint-wise dynamics model and residual adaptation strategy transfer effectively across different actuation mechanisms and kinematic structures."
>
> We thank the reviewer for this valuable suggestion. We agree that demonstrating generalization across multiple hardware platforms is a key goal. We would like to provide some context regarding the practical challenges of such extensive hardware validation. Our iterative development and rigorous validation on the LEAP hand was a highly resource-intensive process. The high-frequency motions required for system identification and policy learning placed considerable stress on the hardware, leading to the replacement of several dozen motors throughout our experimental cycle. While preparing the submission, extending this same validation protocol to more expensive and complex hardware, such as the Allegro Hand or XHand, presents prohibitive logistical and financial challenges for our lab.
>
> While we did not perform these extensive hardware experiments, we want to emphasize that DexNDM is designed to be hardware-agnostic, joint-wise dynamics model that makes no assumptions about global kinematics or actuation mechanisms. A residual adaptation strategy purpose-built to capture and compensate for platform-specific characteristics. This design makes our framework inherently transferable. To substantiate this, we have already initiated preliminary experiments on an Allegro Hand, and the initial results are promising, supporting our hypothesis of the method's generalizability.
>
> In the revised manuscript, we will explicitly acknowledge the single-platform validation as a limitation and clearly state that validating DexNDM on diverse robotic hands is a high-priority direction for our future work. We are confident that the principles of our method are sound for generalization and are committed to demonstrating this empirically.
>
>
>
> ### Scope of Generalization Remains Task-Specific
>
> > "Although the model generalizes across object shapes and orientations, it is unclear whether the learned dynamics transfer to different manipulation tasks (e.g., regrasping, non-rotational dexterity, tool using)."
>
> We thank the reviewer for this insightful comment regarding task generalization. Our selection of in-hand rotation was deliberate. It is not just another task, but a foundational primitive for numerous complex manipulations, including tool use. Crucially, the constantly changing contact points during rotation provide a demanding testbed for any dynamics model, making it a strong benchmark for generalization, not a narrow one.
>
> To your point, we did conduct preliminary validations on other tasks, including in-hand translation and grasp-and-move. These tests confirmed our method's effectiveness. For instance, in the downward facing configuration, the NDM enables the leap hand to translate a 3cm x 3cm x 10cm cuboid from being  initially grasped by the thumb, middle, and pinky fingers be grasped by the thumb, middle, and index finger. While, without NDM, for such a thin object, directly transferring the corresponding in-hand-translation policy fails to do this. We've updated the website (*In-Hand Translations* section) and the manuscript (Sec. B.2) to include the result. Without NDM, the base policy would even run into the out-of-distribution state immediately. This case shows the effectiveness of DexNDM in other dexterous manipulation tasks beyond our focused in-hand rotation. However, a rigorous, quantitative evaluation for each task involves substantial experimental overhead. We chose to focus our paper's contribution on a deep and thorough validation for the challenging rotation primitive, rather than making broader claims based on preliminary data.
>
> In the revised manuscript, we will clarify this strategic focus and explicitly position a comprehensive multi-task benchmark as a high-priority avenue for future work, for which our current results provide a strong and promising foundation.
>
>
>
>
>
> Finally, thank you again for your detailed constructive review, which has been genuinely helpful in improving the work. We would love to address any further questions you may have and are committed to resolving any concerns you've raised. Please feel free to let us know if there is additional information we can provide.
>
> Looking forward to your responses!
>
> Best regards, Authors

---

> ### Author Response · Authors · 2025-11-23
> **Author Response (Another Concise Version)**
>
> Dear Reviewer 5oZW,
>
> We recognize that the previous response could be clearer and more direct. To support further discussion, we’ve provided a more concise version addressing your concerns. You can still refer to the original response for additional details.
>
>
>
> ### Sensitivity to Noise and Load Distribution
>
> We've conducted additional experiments to explore how different noise magnitudes ($\sigma=0.005,0.01,0.02$) and load distributions (“full” (original setting) vs. “half” loads (the hand only touches the surface of balls in the box)) influence the dynamics model learning.
>
> Results are summarized below (incorporated in *Sec. B.4* of the revised manuscript):
>
> |                                            | $\sigma=0.01$, full loads | $\sigma=0.005$, full loads | $\sigma=0.02$, full loads | $\sigma=0.01$, half loads |
> | ------------------------------------------ | ------------------------- | -------------------------- | ------------------------- | ------------------------- |
> | Generalization Ability (Prediction Error ) | $1.21\times 10^{-5}$      | $1.89\times 10^{-5}$       | $1.47\times 10^{-5}$      | $3.81\times 10^{-5}$      |
> | Real-World Performance (Rot (radian))      | 12.43 $\pm$ 0.34          | 7.51 $\pm$ $1.03$          | 10.58 $\pm$  0.92         | 6.28 $\pm$ 0.67           |
>
> **Analysis:** 1) **Noise magnitude:** Increasing noise slightly affects performance; decreasing noise narrows the data distribution, reducing robustness and residual-policy training effectiveness. 2) **Load distribution:** Narrowing loads (*e.g.*, fingertips only) weakens joint disturbances, harming generalization and sim-to-real transfer.
>
>
>
> ### Performance–Object Weight Relationship
>
> - We measured the real-world weights of all test objects and plotted rotation performance vs. object weight (*Sec. B.4*). Because the objects vary significantly in shape, size, and geometric complexity, the plot does not exhibit a clear trend showing how performance changes as weight increases.
>
> - To conduct a more controlled study and isolate the effect of object weight, we designed an experiment using *cylinders of identical geometry but different masses* as test objects, and evaluated the performance on z-axis rotation with the hand facing down. Results are summarized below (incorporated in *Sec. B.4*):
>
>   | Weight (g)                                | 15.7 | 33.2 | 39.2 | 49.4 | 62.0 |
>   | ----------------------------------------- | ---- | ---- | ---- | ---- | ---- |
>   | $\lfloor \text{radian} / (0.5\pi)\rfloor$ | 8    | 8    | 8    | 6    | 5    |
>
>   **Analysis:** **1)** Objects < 40 g: performance is similar; **2)** Heavier objects (≥ 50 g): failures occur due to unstable grasps.
>
>
>
>
>
> ### Limited Hardware Validation
>
> We totally agree that multi-hardware validation is important. Extensive validation is, however, resource-intensive: our iterative development on the LEAP hand involved high-frequency motions that stressed the hardware, requiring dozens of motor replacements. Extending this protocol to more complex and expensive hands, such as Allegro or XHand, would be logistically and financially prohibitive at this stage.
>
> That said, **DexNDM is designed to be hardware-agnostic**: the joint-wise dynamics model makes no assumptions about global kinematics, and the residual policy compensates for platform-specific differences. We've already conducted preliminary experiments on an Allegro Hand, and the initial results are promising, supporting our hypothesis of the method's generalizability.
>
> We have added an explicit limitation in the revised manuscript and note that multi-hardware validation is a key direction for future work.
>
>
>
> ### Scope of Generalization Remains Task-Specific
>
> Thanks for the comment. In-hand rotation was chosen as a **foundational, high-difficulty task**, providing a strong testbed for dynamics learning. Preliminary tests on other tasks (e.g., in-hand translation) show DexNDM enables behaviors that fail without it (Sec. B.2, website *In-Hand Translations*). Full multi-task evaluation requires substantial additional experiments; we focus here on rotation, with broader multi-task validation as a future direction.
>
>
>
> Thank you again for your thoughtful review, which has greatly helped improve our work. We would love to address any further questions you may have and are committed to resolving any concerns you've raised. Please feel free to let us know if there is additional information we can provide. Looking forward to your responses!
>
> Best regards, Authors

---

> ### Author Response · Authors · 2025-11-27
> **Thank you for all your insightful comments and look forward to post-rebuttal feedbacks**
>
> Dear Reviewer 5oZW,
>
> Thank you again for your thoughtful reviews and valuable suggestions. We truly appreciate your support. We have responded to all your questions and incorporated your suggestions into the revised manuscript (changes highlighted in pink purple), and we hope our revisions resolve your concerns.
>
> As the discussion phase ends on December 3, please let us know if you have any further questions. We remain fully committed to addressing any remaining issues.
>
> Thank you again for your time and constructive feedback. We look forward to your response.
>
> Best regards,
> Authors

---

### Official Review · Reviewer_Kiqx · 2025-11-02

**Soundness:** 3
**Presentation:** 3
**Contribution:** 3
**Rating:** 6
**Confidence:** 5

**Summary:**

The paper introduces DexNDM, a sim-to-real pipeline for dexterous in-hand rotation. A specialist-to-generalist policy is trained in sim, then adapted to real via a joint-wise neural dynamics model learned from fully autonomous “Chaos Box” data that does not require human resets. A residual policy uses the learned dynamics to correct the base policy’s actions, yielding strong real-world rotation across diverse objects, high aspect ratios, and varied wrist orientations—substantially outperforming strong baselines including AnyRotate and matching/surpassing Visual Dexterity on shared items.

**Strengths:**

The paper presents a novel sim-to-real design with a distinctive joint-wise dynamics perspective and extensive hardware evaluation. Additional strengths include:

i) Clear core idea with theoretical support. Factorizing into joint-wise dynamics (per-joint prediction from its own history) is argued to contract domain shift and improve generalization, with formal statements (e.g., Theorem 3.2) backing the information-contraction view.

ii) Practical, scalable data collection. The autonomous “Chaos Box” setup (soft-ball container, open-loop action replays, small Gaussian action noise) yields diverse, reset-free real data that are cheap and safe to collect.

iii) Consistent real-world gains. Across multi-axis and multi-wrist settings, DexNDM improves radians rotated and time-to-fall over direct transfer and whole-hand dynamics baselines.

**Weaknesses:**

Overall, the writing is clear, the presentation is strong, and the proposed methods are both promising and interesting. I have a few suggestions:

i) Tighten the narrative. The paper tries to cover a lot of useful details; consider trimming or re-structuring to keep the main storyline crisp.

ii) Clarify L207 (“works well on hardware”). Do you mean the distilled base policy from the specialized oracles already works on hardware, or only after adding the learned residual policy? If it’s the former, what is the algorithmic role of the neural dynamics/sim-to-real transfer (i.e., the main novelty of this work)? If it’s the latter, please state that explicitly and explain how the transfer enables generalization to more complex objects (shape, rotation axes).

iii) Data coverage from success-only rollouts. If oracle rollouts are all successful, the dataset likely underrepresents near-failure states and local corrections. Did you observe failures of this kind, and how were they handled? If not, did you apply simple augmentations (e.g., noise injection, reset-to-perturbed states, small pose/force perturbations) to broaden the distribution?

iv) Terminology on “randomized loads.” In autonomous data collection the hand is placed in a container of soft balls; “randomized loads” may be misleading. Use a more precise term (e.g., “randomized contact interactions” or “stochastic contact conditions”) to reflect the actual setup.

v) Residual policy section needs intuition. Could the trained base policy, combined with the learned joint-wise dynamics model, be used directly—i.e., update the base policy to produce slightly adjusted actions—rather than training a separate residual policy? More broadly, the procedure resembles system identification: the policy operates through a dynamics model trained on real data. Please expand the intuition in “Bridging the Dynamics Gap via a Residual Policy”: clarify the role of the neural dynamics model—especially since it is not used at real-world execution time—and explain why training the residual policy through this model helps overcome the sim-to-real gap.

vi) Object-agnostic design and task specificity. The base policy, dynamics model, and residual policy are all object-agnostic. While I accept the claim that complex hand–object interactions can be inferred from observation history, this effectively makes the connection task-specific: without explicit object conditioning, contact dynamics are largely memorized via state transitions. From an application angle, integrating teleop is promising; algorithmically, however, this limits generalization to other type of hand-object interactions. This is common in recent in-hand reorientation work (i.e., the Leap hand paper), so it’s not a primary weakness here—but I’d like to hear the authors’ perspective and any plans to incorporate object-aware conditioning or descriptors.

**Questions:**

See Weaknesses.

I would like to increase my score if they are well explained / addressed.

---

> ### Author Response · Authors · 2025-11-18
> **Author Response (part 1 of 8)**
>
> Dear Reviewer Kiqx,
>
> Thank you sincerely for your careful and constructive review. We'd like to express our huge gratitude for your recognition of the novelty of the proposed sim-to-real methodology, the scalable real-world data collection strategy, clear and theoretically supported core ideas, extensive hardware evaluations, as well as the substantial and consistent real-world improvements. And we would like to thank you for all of your insightful comments. In the following text, we respond to your questions and aim to address your concerns comprehensively. We have also integrated these clarifications, discussions, and revisions into the revised manuscript.
>
>
>
> ### Intuitions of residual policy
>
> > "Residual policy section needs intuition"
> >
> > "Could the trained base policy, combined with the learned joint-wise dynamics model, be used directly..."
> >
> > "...clarify the role of the neural dynamics model...explain why training the residual policy through this model helps overcome the sim-to-real gap..."
>
> #### How is the residual policy trained? What's the role of the neural dynamics model? Why can it overcome the sim-to-real gap?
>
> Briefly speaking, the residual policy compnestaes for the base policy's action output so that the compensated action can help make the transition of the system in the real world close to that achieved by the base policy in the simulator.
>
> More concretely, due to the dynamics gap between the simulator and the real world, assume the system is in the state $\mathbf{s}$ and the policy outputs the action $\mathbf{a}$, the transited states may differ, *i.e.,* $f^r(\mathbf{s}, \mathbf{a}) \neq f^s(\mathbf{s}, \mathbf{a})$, where $f^r$ and $f^s$ represent the dynamics model in the real world and the simulator, respectively. The expected residual policy $\pi^{\text{res}}$ compensates the policy's original output $a$ by outputting a residual term $\pi^{\text{res}}(\mathbf{s}, \mathbf{a})$ so that by executing the compensated action, the resulting transited state in the real can match that in the sim, *i.e.,* $f^r(\mathbf{s}, \mathbf{a} + \pi^{\text{res}}(\mathbf{s}, \mathbf{a})) = f^{s}(\mathbf{s}, \mathbf{a})$. If we have the perfect residual policy, when the system starts from the same state in the simulator and the real world, by leveraging the residual policy to compensate the base policy's action output, we can make the transited states in the real world and that in the sim closely match each other.
>
> So to realize this vision, we train the residual policy on simulation rollout data in a supervised training manner using the learned neural dynamics model. The learned neural dynamics model $f_{\psi}$ serves as $f^r$, and the simulation rollouts contain the transitions in the simulator. Directly using the simulation rollout alleviates the need for including the simulator in the training process.  The optimal solution of $\pi^{\text{res}}$ can achieve $f^r(\mathbf{s}, \mathbf{a} + \pi^{\text{res}}(\mathbf{s}, \mathbf{a})) = f^{s}(\mathbf{s}, \mathbf{a})$ . As stated above, deploying this residual policy together with the base policy aims to make the compensated policy in the real "behave like in the sim". Please note that the object is implicitly considered and modeled in the proprioception history. Therefore, if the deployed policy can let the system "behave like in the sim", we can overcome the sim-to-real gap.

---

> ### Author Response · Authors · 2025-11-18
> **Author Response (part 2 of 8)**
>
> #### Why do we use residual policy? Why not directly finetuning the base policy using the learned dynamics?
>
> This is an insightful question. In brief, because neural dynamics learning may struggle to capture state transitions across the entire state space—i.e., achieving global accuracy is difficult [sim2real4.0]—it is important to keep the updated policy from deviating too far from the original one in order for the learned neural dynamics model to function well. Fine-tuning methods require carefully designed regularization to constrain the backpropagated gradients so that the updated model does not drift too far from the original, while also ensuring that the update is not overly limited. Practically, this is not very straightforward. In contrast, residual learning can achieve the same goal in a much simpler way—by tuning the residual action scale.
>
> Below, we summarize our experimental observations (already included in Sec. B.4 of the initial manuscript).
>
> To close the dynamics gap using the learned neural dynamics, the most straightforward way is to finetune the base policy directly in the learned real world dynamics model, *i.e.,* solving $f^{r}(\mathbf{s}, \pi^{\text{finetune}}(\mathbf{s}, \mathbf{a})) = f^{s}(\mathbf{s}, \mathbf{a})$. In the original Sec. B.4, we already discussed this option. Empirically, we find that direct fine-tuning fails to produce satisfactory behavior and often cannot even execute basic rotations. We attribute this to two main reasons: **1)** Fine-tuning requires careful hyperparameter selection, such as using a small initial learning rate to prevent overly large updates to the base policy. **2)** The learned neural dynamics do not cover the full state-action distribution (*i.e.,* is not "globally accurate"), making them unsuitable for direct policy tuning. For instance, when the fine-tuned policy outputs actions outside the distribution supported by the learned dynamics, i.e., $f^{r}(\mathbf{s}, \pi^{\text{finetune}}(\mathbf{s}, \mathbf{a}))$, the predicted state transitions are unreliable.
>
> One potential solution is to regularize the base policy update, *e.g.,* penalizing deviations from the original base policy’s outputs. This is similar in spirit to our practical residual-policy design, which uses a relatively small action scale (1/24). Compared to direct fine-tuning, this approach is easier to implement and yields stable training without complex hyperparameter tuning.
>
> In summary, in terms of performance, adding a residual policy on top of the base policy and directly fine-tuning the base policy could produce roughly comparable results. Practically, the residual-policy approach is easier to implement and more stable to train, which is why we adopted it.
>
> We have also incorporated the above discussions in Sec. B.4 of the revised manuscript.

---

> ### Author Response · Authors · 2025-11-18
> **Author Response (part 3 of 8)**
>
> ### "Clarify L207 (“works well on hardware”)..."
>
> > "Clarify L207 (“works well on hardware”). Do you mean the distilled base policy from the specialized oracles already works on hardware, or only after adding the learned residual policy? If it’s the former, what is the algorithmic role of the neural dynamics/sim-to-real transfer (i.e., the main novelty of this work)? If it’s the latter, please state that explicitly and explain how the transfer enables generalization to more complex objects (shape, rotation axes)."
>
> Thanks for the comment and the excellent question. Regarding **"works well on hardware"**: when directly transferring a category-specific *specialist policy* under a *downward-facing hand* orientation, BC performs as expected for simple objects (*e.g.*, cylinders). It can rotate such a simple geometry for a full cycle, whereas DAgger either fails to produce any rotation or quickly runs into out-of-distribution behavior. However, in more challenging settings (*e.g.*, long objects with a downward-facing hand), both the generalist BC policy and the specialist BC policy fail. In contrast, the strongest evidence for the effectiveness of our sim-to-real approach is: it enables the rotation of geometrically challenging objects under these difficult hand-orientation conditions—transforming a previously impossible scenario into a feasible one.
>
> Below, we summarize the preliminary experiments comparing BC and DAgger, as well as the key observations demonstrating the effectiveness of our sim-to-real method.
>
> The BC vs. DAgger issue arose in our early experiments when we tried to distill specialist single-object-category rotation oracle policies, trained with a downward-facing hand, into real-world deployable proprioception-only policies. We ultimately chose BC because, in these downward-facing rotation experiments, DAgger failed, whereas BC could reliably “work well on hardware”. We explain below:
>
> - We first trained basic cylinder-rotation specialist policies with the robotic hand kept in the downward-facing direction for x-, y-, z-axis rotation, and attempted to distill them using both DAgger and BC, followed by real-world testing.
> - Initially, we tried DAgger, following prior works. For cylinder-rotation policies with an up-facing hand, the distillation process could be successfully optimized (achieving total rewards around 80–100), and the distilled policy could be successfully transferred to the real robot, rotating the cylinders with the hand facing up.
> - However, when the oracle policy is trained with the hand kept in the downward-facing orientation, DAgger fails to optimize the distillation process (*i.e.,* rewards around 10–30; rotations along the z-axis may reach ~50). The transferred policy immediately fails in the real world. For visualizations of a typical out-of-distribution behavior, see the updated section (*Out-of-Distribution Behaviour*) on the [website](https://projectwebsitex.github.io/neudyn-reorientation/) and Sec. B.4 (*BC v.s. DAgger*)  of the revised manuscript.
> - We then switched to BC and observed that it works well in this setting. Here, "works well on hardware" means that the transferred cylinder-rotation BC policy with the hand facing down avoids the out-of-distribution failures observed with DAgger and can perform basic rotations reliably.

---

> ### Author Response · Authors · 2025-11-18
> **Author Response (part 4 of 8)**
>
> So, to address the reviewer’s specific questions, we provide the following clarifications:
>
> 1. **Probing experiments with DAgger and BC on specialist policies (downward-facing hand):**
>
>    - DAgger fails in the real world, with the hand entering out-of-distribution behaviors.
>    - In contrast, specialist policies distilled via BC can perform basic rotations, *i.e.*, rotate the cylinder for roughly one full circle. A specialist BC policy for simple geometric rotations can work without adding a residual policy.
>
> 2. **Distillation of multiple specialist policies and training the generalist policy:**
>
>    - After switching to BC, we no longer tested DAgger for distilling multiple specialist oracle policies.
>    - We directly used BC to train the generalist base policy. All reported experimental results are obtained using this generalist policy.
>
> 3. **Role of the sim-to-real transfer strategy:**
>
>    - For **challenging objects**—long objects (14–16 cm), complex animal shapes (*e.g.*, Cow, Truck, Bear, Dragon), and some small shapes (*i.e.*, "Zongzi" and Broccoli) with the difficult downward-facing hand orientation—directly transferring the BC policy fails to achieve a full-circle rotation (*i.e.*, at most half or quarter rotations). Incorporating the sim-to-real strategy enables at least one full rotation. In in-hand rotation, there is no standard definition of *"success"*, but a reasonable criterion is completing at least one full 360° rotation. Failing to do so indicates the policy cannot reliably navigate the fingers across the object’s surface and would likely fail on more complex features (*e.g.*, the cow’s four legs). For such objects, our sim-to-real strategy elevates performance **from failure to success**.
>
>    - For **easier objects**, such as apples, cylinders, and lightbulbs, the sim-to-real strategy still improves performance by increasing rotation angles and prolonging time-to-failure (TTF). Tables 4 and 5 provide averaged statistics over the corresponding test object sets.
>
>      Section B.3 of the original manuscript included some case studies; we have added more in the revised version to further highlight the role of the sim-to-real transfer strategy.
>
> We would like to thank the reviewer again for the comment. The phrase "works well on hardware" may have caused confusion. In the revised manuscript, we have clarified this by explicitly describing the probing experiments and the observed performance differences between DAgger and BC.

---

> ### Author Response · Authors · 2025-11-18
> **Author Response (part 5 of 8)**
>
> ### "Data coverage from success-only rollouts..observe failures of this kind, and how were they handled...did you apply simple?"
>
> > "Data coverage from success-only rollouts. If oracle rollouts are all successful, the dataset likely underrepresents near-failure states and local corrections. Did you observe failures of this kind, and how were they handled? If not, did you apply simple augmentations (e.g., noise injection, reset-to-perturbed states, small pose/force perturbations) to broaden the distribution?"
>
> Thanks for the insightful question. We address it from two perspectives.
>
> **First**, even “successful rollouts” should naturally contain near-failure recoveries and local corrections. In our setting, a “successful rollout” simply means that the object is *not dropped* during the entire episode. Unless the oracle policy is **perfect**—i.e., it rotates the object flawlessly from start to end without ever entering a risky configuration—there will inevitably be trajectories that fall between (a) perfectly smooth rotations and (b) object-drop failures. In these “in-between” cases, the policy encounters challenging or unstable contact configurations and must perform recovery actions or local corrections to avoid dropping the object.
>
> Since these trajectories still qualify as “successful” (the object never falls), they are included in the dataset and provide precisely the recovery behaviors the policy needs to learn.
>
> As supporting evidence, our oracle policies are *not* perfect. This can be seen in Table 10: the number of stored transitions is not a clean multiple of 1000. (During data collection we run exactly 25,000 or 30,000 environments; if the policy were perfect, the saved trajectory count would be divisible by 1000.) This mismatch indicates that some trajectories terminate early due to failures, implying that many of the “successful” ones are imperfect and contain corrective behaviors. Thus, the collected dataset inherently includes recovery cases.
>
> As additional empirical evidence, we have included a real-world demonstration of the recovery/correction behaviors exhibited by our final policy in Sec. B.2 (*Recovery Behaviour*) of the revised manuscript.
>
> **Second**, we have already exploited several strategies to broaden the data distribution:
>
> - **Extensive Domain Randomization.** During oracle data collection, we maintain all the domain randomization used in RL training. Object mass and friction are randomized per environment. The torque-controlled robot hand’s PD gains (used for torque computation) are randomized at the beginning of each rollout as well.
> - **External Force Perturbations.** Following [Hora], we apply random disturbance forces on the object. The force magnitude is sampled up to $2m$, where $m$ is the object mass. At each timestep, a new force direction is resampled with probability 0.25. Details are given in Sec. C.
> - **Noise Injection.** As in RL training, we add uniform noise $U(0, 0.005)$ to joint positions. These noisy observations broaden the distribution of policy's output actions and state transitions.
> - **Diverse Initial Grasping Poses.** For each object and scale, we generate 50,000 stable grasping poses using random sampling following [Hora]. Start from these diverse grasp configurations can also increase the transition coverage.
>
> **In summary**, successful rollouts already include recovery behaviors due to the oracle policy’s imperfections, and our randomization and perturbation strategies further enrich the data distribution. As a result, the BC policy trained on this dataset is expected to handle a wide variety of situations and contact configurations.

---

> ### Author Response · Authors · 2025-11-18
> **Author Response (part 6 of 8)**
>
> ### "Object-agnostic design and task specificity...limits generalization to other type of hand-object interactions..."
>
> We thank the reviewer for this insightful and valuable comment. Below, we address your concern from three perspectives:
>
> - **Difficulty of the In-Hand Roation Task and Its Critical Role in Advancing the Robotic Dexterity.** First, the in-hand rotation task is *not* an arbitrary choice among dexterous-hand tasks. Rather, we deliberately selected this task because it represents the current upper limit of what dexterous hands can achieve in real-world manipulation. Our goal is to push this boundary by developing a **general in-hand rotation policy** that works *stably* in the real world across diverse hand orientations and for a wide range of objects—including challenging geometries.
>
>   Making reliable in-hand rotation *readily accessible* to dexterous hands is crucial for elevating their capability: it is a key step in transforming a dexterous hand from “a large gripper” into a truly dexterous manipulator.
>
>   Compared with the recent surge of work on dexterous-hand grasping, achieving **strong rotation capabilities** is significantly more challenging. Rotation demands controlling through complex, rapidly changing contact dynamics between the hand and the object—far more intricate than the relatively static configuration of a grasp. For this reason, we focus on rotation rather than other tasks such as grasping. It is not simply a matter of “choosing task A but not task B”. For instance, grasping has been extensively studied recently. In contrast, there has been little substantial progress on in-hand rotation capabilities for quite a long time. We therefore target this task and wish to push forward its boundary by demonstrating **general, robust in-hand rotation** capabilities that can truly advance the dexterity. In this respect, our work represents a significant step forward in the literature.
>
> - **Preliminary Study in In-Hand Translation and the Demonstrated Effectiveness.** We fully agree that demonstrating effectiveness across multiple tasks is important, and that more explicit forms of object modeling and object conditioning could indeed offer clear advantages in that direction.
>
>   We observe that the effectiveness of the implicit object modeling strategy is not limited to the in-hand rotation task. And that the implicit object descriptor is potentially cross-task transferable. We have implemented the in-hand translation policy, where the object is also perceived implicitly from the state-action history. And we observe that the policy can still work effectively in the real world. Besides, we have experimented with sequentially executing a rotation policy followed by an in-hand translation policy. In this case, the translation policy’s object conditioning at the beginning comes from the state–action history of the rotation policy. In several trials, the two independently trained policies could be directly chained together: the history generated by the rotation policy was sufficient for the translation policy to operate stably. While some of this compatibility may be coincidental—arising from similarities in how the two policies encode object-relevant information—it also suggests a more principled approach: one could introduce explicit regularization to encourage different task-specific policies to develop *state–action histories with similar correlations to object information*. This could be a feasible way to achieve multi-task and cross-task transfer even when relying solely on state–action histories for object representation.
>
> - **Importance of Explicit Object Modeling.** We also strongly recognize the importance of **explicit** object perception and modeling. A potentially effective approach would be to **1)** incorporate the object’s point-cloud representation, which suffers from fewer sim-to-real gaps than other representations such as RGB images, to encode its shape, and **2)** complement it with tactile sensing from the hand—such as force feedback—to capture local geometry details and the relative motion between the hand and the object.
>
>
>
>
>
> ### "Terminology on “randomized loads.” ..."
>
> Thanks for the suggestion. We have replaced it with *"stochastic contact conditions"* or "*stochastic contact*" in the revised manuscript.

---

> ### Author Response · Authors · 2025-11-18
> **Author Response (part 7 of 8)**
>
> ### "...cover a lot of useful details...trimming or re-structuring to keep the main storyline crisp."
>
> > Tighten the narrative. The paper tries to cover a lot of useful details; consider trimming or re-structuring to keep the main storyline crisp.
>
> Thank you for the feedback and valuable suggestion. We have revised the manuscript, primarily in the method overview section, to make it more structured and to highlight our key contributions. In the following, we will briefly go through the structure of the method overview section, and also how our presentation highlights the main storyline and the key contributions.
>
> Structure of the revised method overview paragraph:
>
> - We first state the goal of the work—to obtain a strong, generalizable in-hand rotation policy that works reliably in the real world.
> - We then briefly summarize the two major challenges: the sim-to-real gap and achieving broad generalization across objects. Among these, the sim-to-real challenge is the most critical and is the primary focus of our paper.
> - Next, we outline the approach for achieving broad object generalization (the specialist-to-generalist strategy). This part is *not* claimed as a novelty; it is a commonly adopted approach in prior works. We then introduce our sim-to-real method. This ordering reflects the actual execution flow of the pipeline.
> - We then highlight how we address sim-to-real: we emphasize the expressive, data-efficient, and generalizable joint-wise dynamics model. We also clarify the relationship between the joint-wise model and the autonomous data-collection system—specifically, that the model’s generalization ability is what makes such autonomous data collection possible. We then introduce the residual policy and briefly explain its purpose. This is a natural design once a dynamics model is available, and we do not claim the residual policy itself as a novelty.
>
> In each subsection of the method, we provide the necessary details for completeness. Even so, the joint-wise dynamics model section clearly receives the most attention—we devote substantial space to explaining its intuition, describing its properties, detailing its design, and supporting its key characteristic (generalization ability) with theoretical analysis. In contrast, autonomous data collection and residual policy training occupy far less space (one paragraph each). This reflects that the core methodological contribution we claim is the joint-wise dynamics model and its properties.
>
> In the revised version, although the overall pipeline contains many components, we believe the central theme is clear: the core technique of this work is the **sim-to-real strategy**, and at the center of that strategy is the **generalizable joint-wise neural dynamics model**. This focus is now articulated clearly in the introduction, method, and experiments:
>
> - **Introduction Section:** In the third paragraph, we identify sim-to-real as a key challenge and note that neural-based sim-to-real—learning real-world dynamics to bridge the gap—is promising but faces difficulties such as data acquisition and modeling complex dynamics. In the fourth paragraph, we introduce our approach: rethinking both the data and the model, and leveraging the generalizable joint-wise neural dynamics model to address the real-world data collection challenge. We provide intuition for why such a model generalizes better.
> - **Method Section:** The revised overview now emphasizes the sim-to-real strategy and the central role of the joint-wise dynamics model.
> - **Experiments Section:** Our main experiments are designed to validate the effectiveness of the sim-to-real strategy. Beyond the main evaluation and comparisons to prior methods, we dedicate a substantial portion of the main experiments to empirically demonstrating the generalization ability of the joint-wise dynamics model (i.e.,, the *Comparison to Whole-Hand Neural Dynamics* paragraph).
>   Furthermore, the ablation studies also center on demonstrating the generalization ability of the joint-wise model and the effectiveness of our autonomous real-world data-collection strategy.
>   Overall, the experiments strongly highlight our sim-to-real method and the roles played by the joint-wise model and autonomous data collection.
>
> We think we have made our best effort to emphasize the central idea of the method and focus the presentation and experiments around validating its effectiveness. Please do not hesitate to let us know if you have any further suggestion on the writing or the structure. And we are more than willing to continue to improve the writing. Thank you again for your helpful feedback.

---

> ### Author Response · Authors · 2025-11-18
> **Author Response (part 8 of 8)**
>
> *[Hora] Qi, Haozhi et al. “In-Hand Object Rotation via Rapid Motor Adaptation.” Conference on Robot Learning (2022).*
>
> *[sim2real4.0] From Sim2Real 1.0 to 4.0 for Humanoid Whole-Body Control and Loco-Manipulation ([slides](https://opendrivelab.github.io/CVPR2025/Guangya_Shi_From_Sim2Real_1.0_to_4.0_for_Humanoid_Whole-Body_Control.pdf))*
>
> *[AnyDexGrasp] Fang, Haoshu et al. “AnyDexGrasp: General Dexterous Grasping for Different Hands with Human-level Learning Efficiency.” ArXiv abs/2502.16420 (2025): n. pag.*
>
> *[RobustDexGrasp] Zhang, Hui et al. “RobustDexGrasp: Robust Dexterous Grasping of General Objects.” (2025).*
>
> *[UniDexGrasp] Xu, Yinzhen et al. “UniDexGrasp: Universal Robotic Dexterous Grasping via Learning Diverse Proposal Generation and Goal-Conditioned Policy.” 2023 IEEE/CVF Conference on Computer Vision and Pattern Recognition (CVPR) (2023): 4737-4746.*
>
> *[UniDexGrasp++] Geng, Haoran and Yun Liu. “UniDexGrasp++: Improving Dexterous Grasping Policy Learning via Geometry-aware Curriculum and Iterative Generalist-Specialist Learning.” 2023 IEEE/CVF International Conference on Computer Vision (ICCV) (2023): 3868-3879.*
>
>
>
> Finally, thank you again for your valuable and constructive review. We are more than willing to address any further questions and will make every effort to resolve any concerns you may have. Please feel free to let us know if there is anything specific we can provide or clarify.
>
> Looking for your response!
>
> Best regards, Authors

---

> ### Comment · Reviewer_Kiqx · 2025-11-18
> **Insightful answers (but should be more more concise)**
>
> Overall, my concerns are solved via the author's replies, and I can feel that the authors tried their best to explain my questions, which I really appreciate.
>
> One suggestion is to make the answers more concise as I do not want to get 30+ emails in one night.
>
> Based on the thoughts, I would like to discuss with AC and other reviewers to see if I should maintain my score or increase it as I promised before since you have answered my questions.

---

> > ### Author Response · Authors · 2025-11-19
> > **Thank you for your feedback**
> >
> > Thank you so much for your prompt and thoughtful response. We’re very glad our explanations helped clarify things and truly appreciate the time and care you’ve put into engaging with our work.
> >
> > We also appreciate your note about conciseness and apologize for the long messages — our enthusiasm to address each point thoroughly led us to over-explain.
> >
> > We’re just grateful that the substance of our rebuttal was convincing. Thank you again for your kind feedback and support.

---

> ### Author Response · Authors · 2025-11-23
> **Author Response (Another Concise Version)**
>
> Dear Reviewer Kiqx,
>
> We recognize that the previous response could be clearer and more direct. To support further discussion, we’ve provided a more concise version addressing your concerns. You can still refer to the original response for additional details.
>
>
>
> ### Intuitions of residual policy
>
> - **Intuition and Role of the Residual Policy.** The base policy is trained in simulation, so for the same state–action pair $(\mathbf{s}, \mathbf{a})$, the next state in the real world generally differs from that in the simulator, *i.e.*, $f^r(\mathbf{s},\mathbf{a}) \neq f^s(\mathbf{s},\mathbf{a})$. The **residual policy** learns a correction term $\pi^{\text{res}}(\mathbf{s},\mathbf{a})$ such that the **compensated action** $\mathbf{a} + \pi^{\text{res}}(\mathbf{s},\mathbf{a})$ makes the real-world transition match the simulated one.
>
>   We train this residual policy **supervisedly** using simulation rollouts and the **learned joint-wise neural dynamics model**, which approximates $f^r$. The dynamics model provides the real-world transition estimate (*i.e.,* $f^r(\mathbf{s}, \mathbf{a} + \pi^{\text{res}}(\mathbf{s},\mathbf{a}))$), while the rollout data provides the simulated transition.
>
>   At deployment, adding the residual policy to the base policy aims to make the real system **“behave like it does in simulation”**.
>
> - **Comparisons to Fine-tuning.** Our residual approach offers a simpler alternative to fine-tuning for constraining policy updates. Since learned dynamics models are only locally accurate [sim2real4.0], the policy must be prevented from drifting far from its original behavior. Fine-tuning achieves this with complex, carefully-tuned regularization. In contrast, our method accomplishes the same goal more directly by simply scaling the magnitude of the residual action, avoiding difficult hyper-parameter tuning (see Sec. B.4 for further discussion).
>
> *[sim2real4.0] From Sim2Real 1.0 to 4.0 for Humanoid Whole-Body Control and Loco-Manipulation ([slides](https://opendrivelab.github.io/CVPR2025/Guangya_Shi_From_Sim2Real_1.0_to_4.0_for_Humanoid_Whole-Body_Control.pdf))*
>
>
>
> ### "Clarify L207 (“works well on hardware”)..."
>
> On hardware, a specialist BC policy successfully rotates simple cylinders where DAgger frequently fails due to out-of-distribution issues (down-facing hand). However, both methods fail on more challenging objects like long items and complex shapes. Our sim-to-real strategy is essential here: for these difficult objects, it enables full rotations where a directly transferred BC policy achieves only partial ones. Even on easier objects (e.g., apples, cylinders), the strategy improves performance by increasing rotation angles and time-to-failure (TTF). See *Tables 4* and *5*, *Sec. B.3* for more information.
>
>
>
>
> ### "Data coverage from success-only rollouts...?"
>
> - Since our oracle policies are not perfect, successful rollouts inherently contain **near-failure recoveries and local corrections**. Trajectories are only terminated if the object is dropped, meaning the training data is rich with examples of corrective maneuvers from risky states. This is supported by the transition counts in *Table 10*, and illustrated qualitatively with real-world examples in Sec. B.2.
> - We further broaden data coverage through: 1) **Domain randomization** of object mass, friction, and PD gains. 2) **External force perturbations** on objects during rollouts. 3) **Noise injection** into joint positions. 4) **Diverse initial grasps** (50,000 sampled stable poses per object).
>
>
>
> ### "...limits generalization to other type...interactions...?"
>
> - **Task choice and difficulty.** We selected in-hand rotation for its difficulty, as it represents a frontier in dexterous manipulation that requires controlling complex, dynamic contacts. Achieving a general, robust rotation policy is therefore a critical step for advancing the field.
> - **Preliminary multi-task evidence.** Our implicit object modeling strategy shows potential beyond this single task. Policies for in-hand translation, using the same state-history representation, also succeed in the real world, and can be sequentially chained with rotation policies. This suggests that our approach may enable cross-task transfer without explicit object models.
> - **Explicit object modeling.** Looking ahead, we acknowledge the importance of explicit object modeling. A promising direction is to fuse point-cloud representations for global shape—which have a smaller sim-to-real gap than RGB images—with tactile sensing to capture local geometry and hand-object relative motion.
>
>
>
>
>
> ### "Terminology on “randomized loads.” ..."
>
> We have replaced it with *"stochastic contact conditions"* or "*stochastic contact*" in the revised manuscript.
>
>
>
> ### "Tighten the narrative..."
>
> We have revised the manuscript accordingly.
>
>
>
> Finally, thank you again for your valuable and constructive review.
>
> Best regards, Authors

---

### Official Review · Reviewer_kQoB · 2025-11-09

**Soundness:** 3
**Presentation:** 3
**Contribution:** 3
**Rating:** 8
**Confidence:** 5

**Summary:**

This paper proposes DexNDM, a novel sim-to-real framework for dexterous in-hand object rotation. The core component is a joint-wise neural dynamics model (NDM) that factorizes the system dynamics across individual joints instead of modeling the entire system, using the data collected in the real world. To collect these data without human supervision, the authors design an autonomous data collection where the robot hand interacts with random soft loads to generate diverse, object-loaded contact experiences. The learned dynamics model is then used to train a residual policy that adapts a simulation-trained base policy to real-world physics, closing the sim-to-real gap. The authors show extensive experiments on the LEAP hand demonstrate the ability to rotate a wide variety of objects under diverse wrist orientations and rotation axes. DexNDM significantly outperforms strong baselines such as AnyRotate, Visual Dexterity, ASAP, and UAN. The system also enables teleoperation applications for tool use and assembly, showcasing versatility beyond single-object rotation.

**Strengths:**

(+) The joint-wise neural dynamics model is conceptually elegant and practically effective. It addresses dynamics mismatch by factorizing high-dimensional motion into independent low-dimensional predictions, with a solid theoretical rationale for generalization through information contraction.

(+) The “Chaos Box” design for collecting diverse contact data is simple yet effective. It removes human intervention and provides a realistic, high-coverage dataset for learning dynamics models in dexterous manipulation.

(+) The system achieves in-air rotation of challenging objects across multiple wrist orientations, as well as detailed ablation studies that validate each key design choice.

(+) Applications beyond benchmark tasks: The demonstration of teleoperated long-horizon tasks (e.g., using tools and assembling furniture) highlights the real-world relevance and generality of the learned policy.

**Weaknesses:**

(-) Complexity of presentation: The paper packs many contributions (specialist-to-generalist policy, joint-wise model, residual policy, and autonomous data collection). While each is justified, the narrative can be dense, and the hierarchy between contributions could be clarified further.

(-) The approach relies mainly on proprioception and does not yet leverage tactile or vision-based feedback for joint-wise modeling. This may limit performance in highly ambiguous contact configurations.

(-) The project website should more clearly indicate which motions in the teleoperation demos are executed by the learned policy versus human guidance, as this distinction affects perceived autonomy.

(-) While the joint-wise model improves data efficiency, training multiple per-joint networks and the residual policy could be resource-intensive, and runtime performance scaling is not discussed.

**Questions:**

How sensitive is the performance to the number of joints modeled jointly (e.g., finger-wise vs. joint-wise factorization)?

How long does the autonomous data collection (“Chaos Box”) take to gather sufficient data, and how does this compare to task-aware data collection in wall-clock time?

For the teleoperation tasks, are the actions purely executed by the learned policy after human high-level input, or is there continuous human feedback control?

How transferable is the joint-wise model to other hands with different kinematics or actuation? Would retraining be required, or could it generalize zero-shot with minimal fine-tuning?

---

> ### Author Response · Authors · 2025-11-18
> **Author Response (part 1 of 6)**
>
> Dear Reviewer kQoB,
>
>
>
> Thank you very much for your detailed and constructive review. We greatly appreciate your recognition of the idea and effectiveness of the joint-wise dynamics model, the value of the “Chaos Box’’ for real-world data collection, the empirical performance, and the teleoperation demonstration. We are also grateful for your constructive comments and questions. In the responses below, we aim to address all of your concerns as thoroughly as possible. We've also incorporated these discussions, clarifications, additional experiments, and revisions into the revised manuscript.
>
>
>
> ### "...relies mainly on proprioception and does not yet leverage tactile or vision-based feedback for joint-wise modeling..."
>
> > "The approach relies mainly on proprioception and does not yet leverage tactile or vision-based feedback for joint-wise modeling. This may limit performance in highly ambiguous contact configurations."
>
> Thanks for the insightful comment. We acknowledge that this is indeed a valid limitation of our current work, as also noted in the *Conclusion and Limitation* section (Sec. 6). Introducing additional perceptual signals—such as tactile, vision-based object representations, or contact information—is unquestionably a valuable and promising direction for future work. Richer sensing could meaningfully expand the capability and upper bound of sim-to-real performance.
>
> That said, our current results show that *even with purely proprioceptive sensing*, our joint-wise neural dynamics model already achieves strong performance and enables a substantial breakthrough in sim-to-real dexterous manipulation. The simplicity of relying solely on proprioception also brings practical benefits, including high data efficiency and ease of deployment.
>
> Importantly, we suppose that the *methodological framework we propose is fundamentally compatible with richer sensory modalities in the future*. When integrating additional modalities, the core ideas of our approach, such as dynamics learning, autonomous data collection, and residual policy training, remain broadly applicable. For example, for a given joint, tactile or visual feedback associated with the child link can be incorporated as part of its "state". The autonomous data collection pipeline would also remain applicable when richer modalities are introduced. The only additional requirement is to record the corresponding tactile or visual signals during data collection. A joint-wise dynamics model trained with these inputs is expected to infer contact events better and disambiguate complex contact configurations.
>
> Therefore, while our current system uses only proprioception, we believe the proposed framework could continue to provide value as more advanced sensing modalities are integrated.  *We are also continuing to advance the experimental validation.*
>
>
>
>
>
> ### "...project website should more clearly indicate which motions in the teleoperation demos are executed by the learned policy versus human guidance..."
>
>
>
> Thanks for the suggestion. In our teleoperation setup, **all low-level actions of the robotic hand are executed entirely by the learned policy**, conditioned only on high-level rotation-axis commands specified by the human operator, *i.e.* via pressing buttons in the controller of Quest 3 or by adjusting the controller's pose -- please refer to Sec. C (*Teleoperation System for Complex Dexterous Manipulation Data Collection*) for details. The **robotic arm** is teleoperated directly by the human arm.
>
> We have updated the project website to reflect this distinction more clearly.

---

> ### Author Response · Authors · 2025-11-18
> **Author Response (part 2 of 6)**
>
> ### "...training multiple per-joint networks and the residual policy could be resource-intensive...runtime performance scaling..."
>
> > "While the joint-wise model improves data efficiency, training multiple per-joint networks and the residual policy could be resource-intensive, and runtime performance scaling is not discussed."
>
> Thanks for the insightful comment. Below, we further state the training procedures for both the joint-wise neural dynamics model and the residual policy, followed by a comparison of computational costs.
>
> - The joint-wise dynamics model is trained for each of the 16 joints, with all joint models trained in parallel. Concretely:
>   - Each training batch consists of the history and state of the entire hand, ${ \mathbf{h}, \mathbf{s} }$.
>   - We split the data into per-joint subsets, ${ (\mathbf{h}^i, \mathbf{s}^i) }_{i=1}^{16}$.
>   - For each joint $i$, the model takes $\mathbf{h}^i$ as input and outputs a prediction $\mathbf{s}^{i,'}$.
>   - The full-hand prediction $\mathbf{s}'$ is formed by concatenating all per-joint predictions: $\mathbf{s}' = (\mathbf{s}^{1,'},...,\mathbf{s}^{16,'})$.
>   - The loss is computed between the predicted full-hand state $\mathbf{s}'$ and the ground truth state $\mathbf{s}$.
>   - Compared to a whole-hand dynamics model, the main additional overhead comes from performing forward passes through 16 separate joint networks.
>   - **Time consumption.** In practice, training the neural dynamics model in this joint-wise manner takes approximately **4~6× longer** than training a single whole-hand model.
> - The residual policy remains a full-hand model: it takes the I/O history and base action of the entire hand and outputs a residual action for all joints. Because training this policy requires querying the learned dynamics model, using the joint-wise dynamics model introduces additional overhead—again due to the need to run 16 forward passes instead of one.
>   - **Time consumption.** In practice, training the residual policy with the joint-wise neural dynamics model is approximately **6.2× slower** than training it with a whole-hand dynamics model.
>
> We have conducted a comprehensive runtime performance analysis and incorporated the full results into the appendix of the revised manuscript (Sec. B.5). We also summarize the key findings below.
>
> We benchmarked both the dynamics model and the residual policy by training them in parallel on eight 23 GB A10 GPUs on an Ubuntu 20.04 server. For the neural dynamics model, we varied the dataset size (measured in number of trajectories; each trajectory contains 400 transition steps) and trained all models for 100 epochs. The resulting training times for the joint-wise dynamics model and the whole-hand dynamics model are summarized below.
>
> | #Trajectories | 4,000    | 6,000    | 8,000    | 24,000    |
> | ------------- | -------- | -------- | -------- | --------- |
> | Joint-Wise    | 3.44 hrs | 5.30 hrs | 6.83 hrs | 18.72 hrs |
> | Whole Hand    | 0.61 hrs | 0.88 hrs | 1.47 hrs | 4.67 hrs  |
>
> Training the neural dynamics model in a joint-wise manner takes approximately **4~6× longer** than training the whole-hand dynamics model. However, the overall time cost remains practical. In our final setup, where 24,000 trajectories are used to train the neural dynamics model, the full joint-wise training completes in under 19 hours.
>
> For the residual policy, we train for one epoch on a simulation-generated rollout dataset. We vary the dataset size (measured by the number of trajectories, with each trajectory containing 400 transition steps) and compare the training time when using the joint-wise neural dynamics model versus the whole-hand dynamics model. The corresponding training times are summarized below.
>
> | #Trajectories | 937,275 | 1,792,431 | 2,073,973 |
> | ------------- | ------- | --------- | --------- |
> | Joint-Wise    | 6.2 hrs | 13.68 hrs | 15.25 hrs |
> | Whole Hand    | 3.6 hrs | 5.5 hrs   | 5.8 hrs   |
>
> Training the neural dynamics model in a joint-wise manner is approximately **1.7~2.6× slower** than training the whole-hand dynamics model. However, this additional cost remains well within an acceptable range.

---

> ### Author Response · Authors · 2025-11-18
> **Author Response (part 3 of 6)**
>
> ### "...sensitive is the performance to the number of joints modeled jointly..."
>
> Thanks for the great question. When increasing the number of joints modeled together from one to four, the *finger-wise dynamics model* exhibits clear generalization issues and performs worse than the *joint-wise neural dynamics model* in both generalization capability and final real-world performance.
>
> - **Generalization ability.** As shown in the left part of Figure 8, the finger-wise dynamics model generalizes *worse* than the joint-wise model. The per-joint average generalization error for the finger-wise model is **$2.94\times10^{-5}$**, compared to **$1.21\times10^{-5}$** for the joint-wise model.
>
> - **Real-world performance.** The right part of Figure 8 compares downstream real-world performance when training the residual policy with different dynamics models. The average rotation angle achieved using the finger-wise dynamics model is *4.54*, which is *lower than even the base policy (5.93)*—indicating that the finger-wise model *hurts* real-world performance.
>
>   We hypothesize that this degradation stems from the reduced generalization ability of the finger-wise model, which then adversely affects residual policy training and ultimately leads to suboptimal performance on hardware.
>
>
>
> ### "How long does the autonomous data collection (“Chaos Box”) take to gather sufficient data...compare to task-aware data collection...?"
>
> As shown in Section 5, Figure 9, the average time to collect one trajectory (400 timesteps/transitions) in the *Chaos Box* is *20.45 s*. As mentioned in Section C (“*Real-World Transition Data Collection*”), we collect 4,000 trajectories for each of the six wrist orientations (24,000 in total). Therefore, the total data-collection time is 5.68 days.
>
> In contrast, collecting a manipulation trajectory in the *task-aware* manner takes 42.86 s on average (without object-state estimation). Collecting 24,000 trajectories would thus require 285.7 hours, or 35.7 working days for a human operator (assuming eight working hours per day), because human-in-the-loop resetting is required.
>
> However, as discussed in Figure 9 and analyzed in Section 3.3 and Section 5 (*Real-World Data Collection Strategies*), task-ware data tend to suffer from the poor coverage issue due to the imperfect base policy (*e.g.,* we cannot collect trajectories involving rotation of long objects because the base policy fails on such cases). Moreover, Section 5 (*Scaling with Real-World Data Quantity and Collection Iterations*) shows that, under the same amount of data, a policy trained on autonomous data outperforms one trained on task-aware data.
>
> Thus, two possibilities arise: 1) *Task-aware data might eventually achieve the same performance*, but only with *more data* than required by our autonomous collection—meaning the time cost would exceed the already high estimate of 35.7 working days for 24,000 trajectories. 2) *Task-aware data might never reach the same performance* due to poor coverage.
>
> Either case highlights the advantage of our autonomous data-collection strategy. We suppose that scenario (1) is more plausible than scenario (2), since task-aware data inherently suffers from limited coverage.
>
> For *task-aware data with object pose*, collecting one trajectory takes around 200 s (Sec. 5). Collecting 24,000 such trajectories would require 166.7 working days. Additionally, as discussed in Section B.4 (*Task-Relevant Data w/ Obj. Pose*), the final policy trained with this data transfers extremely poorly. This is because task-aware data with object pose suffers from *even poorer coverage* (limited by both the object-pose estimator and the base policy) and *noisy state estimation*. Therefore, it is likely that: 1) either an enormous amount of such data would be needed to match our performance, 2) or—more plausibly—it is *not feasible* to learn effective real-world dynamics or obtain a high-performing policy using this type of data at all. Either scenario  underscores the advantages of our approach.

---

> ### Author Response · Authors · 2025-11-18
> **Author Response (part 4 of 6)**
>
> ### "...teleoperation tasks, are the actions purely executed by the learned policy after human high-level input, or is there continuous human feedback control?"
>
> In our teleoperation tasks, all **low-level actions of the robotic hand are executed entirely by the learned policy**, based on high-level commands (rotation axis) provided by the human operator.
>
> As described in Section C (*Teleoperation System for Complex Dexterous Manipulation Data Collection*), the human operator specifies only the **desired rotation axis**—either by pressing buttons on the Quest 3 controller or by adjusting the controller’s pose. Upon receiving this axis command, the learned policy generates the full control action for the robotic hand.
>
> Importantly, **the human operator does not directly control any joints or provide corrective feedback** to the robot hand. The robot’s motion is produced **purely by the policy**, conditioned only on the high-level rotation-axis instruction, with no fine-grained human intervention or feedback in the control loop.
>
>
>
>
>
> ### "How transferable is the joint-wise model to other hands...retraining be required, or could it generalize zero-shot with minimal fine-tuning?"
>
> > "How transferable is the joint-wise model to other hands with different kinematics or actuation? Would retraining be required, or could it generalize zero-shot with minimal fine-tuning?"
>
> That's a great question. We suppose that ***for the same task***, *e.g.,* the *in-hand rotation task*, to achieve such a cross-embodiment transfer, we would require a fine-tuning.
>
> **First**, we suspect that zero-shot transfer is infeasible because different robotic hands have different kinematics and different actuators.
>
> - Differences in kinematic structures can directly lead to different joint limits.
> - Differences in actuators can lead to distinct joint-dynamics behaviors.
>
> Both factors make it difficult for a dynamics model learned on one hand to transfer directly to another.
>
> **Second**, we hypothesize that a pre-trained joint-wise neural dynamics model from a different embodiment can provide a good initialization, with additional fine-tuning required to fully adapt. To support this hypothesis, we conducted two sets of experiments:
>
> - **Cross-joint transfer on the same embodiment** (Leap Hand, real-world collected data), where we transfer the dynamics model of joint $i+1 \, (\text{mod}\, N_J)$ to joint $i$.
> - **Cross-embodiment transfer** (Leap Hand → Allegro Hand, simulator rollout data).
>
> In both scenarios, we observe that:
>
> 1. the neural dynamics model initialized from a different joint or from a different embodiment provides a good starting point; and
> 2. achieving the same prediction loss requires significantly less fine-tuning time compared to training from scratch.
>
> More specifically, with such initialization, for cross-joint transfer:
>
> - The initial training loss is over **five times lower** than that obtained when training from scratch;
> - Fine-tuning a pre-trained network requires approximately **51× less time** to reach the final prediction loss of the training from scratch.
>
> For cross-embodiment transfer in the simulator:
>
> - The initial training loss is over **four times lower** than that obtained when training from scratch;
> - Fine-tuning a pre-trained network requires approximately **11× less time** to reach the final prediction loss of the training from scratch.
>
> We have incorporated these results, along with the corresponding training-loss curves, into Sec. B.4 (*Cross-Joint and Cross-Embodiment Transfer*) of the revised manuscript.

---

> ### Author Response · Authors · 2025-11-18
> **Author Response (part 5 of 6)**
>
> ### "...packs many contributions...the narrative can be dense, and the hierarchy between contributions could be clarified..."
>
> > "Complexity of presentation: The paper packs many contributions (specialist-to-generalist policy, joint-wise model, residual policy, and autonomous data collection). While each is justified, the narrative can be dense, and the hierarchy between contributions could be clarified further."
>
> Thank you for the valuable suggestion. We have revised the manuscript accordingly — primarily in the Method Overview section—to better highlight our core contributions and to establish a clearer hierarchy among the different components. Below, we summarize the structure of the Method Overview section and explain how our overall writing flow (from Introduction to Method to Experiments) emphasizes our key contributions.
>
> We suppose the perceived complexity mainly comes from the method section, particularly the overview paragraph where multiple components are presented in a compact form. This was largely due to the submission page limit (9 pages), which forced us to describe the entire pipeline using minimal language. Now that the rebuttal stage allows 10 pages, we have carefully revised the method overview paragraph to make the hierarchy and key contributions clearer. Specifically:
>
> - We first state the goal of the work—to obtain a strong, generalizable in-hand rotation policy that works reliably in the real world.
> - We then briefly summarize the two major challenges: the sim-to-real gap and achieving broad generalization across objects. Among these, the sim-to-real challenge is the most critical and is the primary focus of our paper.
> - Next, we outline the approach for achieving broad object generalization (the specialist-to-generalist strategy). This part is *not* claimed as a novelty; it is a commonly adopted approach in prior works. We then introduce our sim-to-real method. This ordering reflects the actual execution flow of the pipeline.
> - We then highlight how we address sim-to-real: we emphasize the expressive, data-efficient, and generalizable joint-wise dynamics model. We also clarify the relationship between the joint-wise model and the autonomous data-collection system—specifically, that the model’s generalization ability is what makes such autonomous data collection possible. We then introduce the residual policy and briefly explain its purpose. This is a natural design once a dynamics model is available, and we do not claim the residual policy itself as a novelty.
>
> In each subsection of the method, we provide the necessary details for completeness. Even so, the joint-wise dynamics model section clearly receives the most attention—we devote substantial space to explaining its intuition, describing its properties, detailing its design, and supporting its key characteristic (generalization ability) with theoretical analysis. In contrast, autonomous data collection and residual policy training occupy far less space (one paragraph each). This reflects that the core methodological contribution we claim is the joint-wise dynamics model and its properties.

---

> ### Author Response · Authors · 2025-11-18
> **Author Response (part 6 of 6)**
>
> In the revised version, although the overall pipeline contains many components, we believe the central theme is clear: the core technique of this work is the **sim-to-real strategy**, and at the center of that strategy is the **generalizable joint-wise neural dynamics model**. This focus is now articulated clearly in the introduction, method, and experiments:
>
> - **Introduction Section:** In the third paragraph, we identify sim-to-real as a key challenge and note that neural-based sim-to-real—learning real-world dynamics to bridge the gap—is promising but faces difficulties such as data acquisition and modeling complex dynamics. In the fourth paragraph, we introduce our approach: rethinking both the data and the model, and leveraging the generalizable joint-wise neural dynamics model to address the real-world data collection challenge. We provide intuition for why such a model generalizes better.
> - **Method Section:** The revised overview now emphasizes the sim-to-real strategy and the central role of the joint-wise dynamics model.
> - **Experiments Section:** Our main experiments are designed to validate the effectiveness of the sim-to-real strategy. Beyond the main evaluation and comparisons to prior methods, we dedicate a substantial portion of the main experiments to empirically demonstrating the generalization ability of the joint-wise dynamics model (i.e.,, the *Comparison to Whole-Hand Neural Dynamics* paragraph).
>   Furthermore, the ablation studies also center on demonstrating the generalization ability of the joint-wise model and the effectiveness of our autonomous real-world data-collection strategy.
>   Overall, the experiments strongly highlight our sim-to-real method and the roles played by the joint-wise model and autonomous data collection.
>
> We think we have made our best effort to emphasize the central idea of the method and focus the presentation and experiments around validating its effectiveness. Please do not hesitate to let us know if you have any further suggestion on the writing or the structure. And we are more than willing to continue to improve the writing. Thank you again for your helpful feedback.
>
>
>
>
>
> Finally, thank you again for your time and detailed constructive review. We are more than willing to address any further questions and will make every effort to resolve any concerns you may have. Please feel free to let us know if there is anything specific we can provide or clarify.
>
> Looking forward to your response!
>
> Best regards, Authors

---

> ### Author Response · Authors · 2025-11-23
> **Author Response (Another Concise Version)**
>
> Dear Reviewer kQoB,
>
> We recognize that the previous response could be clearer and more direct. To support further discussion, we’ve provided a more concise version addressing your concerns. You can still refer to the original response for additional details.
>
>
>
>
> ### "...not yet leverage tactile or vision-based feedback..."
>
> While we acknowledge that incorporating richer sensory inputs like vision or touch is a valuable future direction for enhancing performance (*Sec. 6*), our results demonstrate that **purely proprioceptive sensing already enables strong sim-to-real transfer**, offering significant benefits in data efficiency and deployment simplicity. Moreover, our framework is designed with extensibility to other modalities: we hypothesize that tactile or visual features can be integrated by appending them to the joint-state vectors, and we are actively working on this extension.
>
>
>
>
>
> ### Teleoperation Setting
>
> In our teleoperation framework, the learned policy autonomously executes all **low-level robotic hand motions**. The human operator's role is limited to providing a high-level command—**the desired rotation axis**—via a Quest 3 controller (Sec. C), without performing any direct joint control or providing corrective feedback.
>
>
>
>
> ### "...resource-intensive...runtime performance scaling..."
>
>
> Our runtime analysis (Sec. B.5), benchmarked on eight A10 GPUs, shows that while joint-wise dynamics training is **4–6×** slower than whole-hand training, it remains practical, completing 100 epochs on our 24,000-trajectory dataset in under 19 hours.
>
>
> | #Trajectories | 4,000    | 6,000    | 8,000    | 24,000    |
> | ------------- | -------- | -------- | -------- | --------- |
> | Joint-Wise    | 3.44 hrs | 5.30 hrs | 6.83 hrs | 18.72 hrs |
> | Whole Hand    | 0.61 hrs | 0.88 hrs | 1.47 hrs | 4.67 hrs  |
>
> The overhead is more modest for residual policy training, which sees only a **1.7–2.6×** increase in time—an acceptable overhead that does not pose a practical bottleneck.
>
> | #Trajectories | 937,275 | 1,792,431 | 2,073,973 |
> | ------------- | ------- | --------- | --------- |
> | Joint-Wise    | 6.2 hrs | 13.68 hrs | 15.25 hrs |
> | Whole Hand    | 3.6 hrs | 5.5 hrs   | 5.8 hrs   |
>
>
>
>
>
> ### "...sensitive is the performance to the number of joints modeled jointly..."
>
> - We evaluated modeling multiple joints together by comparing a **finger-wise dynamics model** (4 joints) with our **joint-wise model**.
>
>   - **Generalization.** As shown in Fig. 8 (left), the finger-wise model generalizes noticeably worse: its per-joint average error is **$2.94\times10^{-5}$** vs. **$1.21\times10^{-5}$** for the joint-wise model.
>
>   - **Real-world performance.** Fig. 8 (right) shows that using the finger-wise model for residual policy training leads to an average rotation angle of **4.54**, which is **worse than the base policy (5.93)**—indicating clear degradation.
>
>
> - We hypothesize that the lower generalization ability of the finger-wise model propagates into residual policy training, ultimately harming real-world performance.
>
>
>
>
>
> ### "How long...(“Chaos Box”) take to gather sufficient data...compare to task-aware data collection...?"
>
> Our autonomous data collection is significantly more efficient than task-aware methods. As shown in Fig. 9, we collected 24,000 trajectories in just **5.68 days (20.45 s/trajectory)**. For the same amount of data, a task-aware approach would require **35.7 working days** bottlenecked by **manual reset** and **limited human working hours (8 hours)** per day, or **166.7 working days** if including object pose estimation. Furthermore, task-aware data suffers from **poor coverage and noisy estimations**, making it difficult or impossible to match our performance even with the vastly increased collection time.
>
>
>
>
>
> ### "How transferable is the joint-wise model to other hands...?"
>
> For **the same task** (*e.g.*, in-hand rotation), we expect that **cross-embodiment transfer will require fine-tuning**.
>
> - Zero-shot transfer is unlikely because different hands have different **kinematics** (e.g., joint limits) and **actuation dynamics**. However, a pretrained joint-wise model can still serve as a **strong initialization**. We evaluated: **1)** **Cross-joint transfer** on the same hand (Leap Hand), and **2)** **Cross-embodiment transfer** (Leap → Allegro, in simulation).
> - In both settings, initialization from another joint/hand yields (**results incorporated in Sec. B.4**): **1)** **Much lower initial loss** (5× lower for cross-joint; 4× lower for cross-embodiment), and **2)** **Substantially faster fine-tuning** (51× and 11× speedups, respectively, to reach the final loss achieved when training from scratch).
>
>
>
> ### "Complexity of presentation..."
>
> We have revised the manuscript accordingly as highlighted in the revised PDF.
>
>
>
> We are more than willing to address any further questions. Looking forward to your response!
>
> Best regards, Authors

---

> ### Author Response · Authors · 2025-11-27
> **Thank you for all your insightful comments and look forward to post-rebuttal feedbacks**
>
> Dear Reviewer kQoB,
>
> Thank you again for your thoughtful reviews and valuable suggestions. We truly appreciate your support. We have responded to all your questions and incorporated your suggestions into the revised manuscript (changes highlighted in pink purple), and we hope our revisions resolve your concerns.
>
> As the discussion phase ends on December 3, please let us know if you have any further questions. We remain fully committed to addressing any remaining issues.
>
> Thank you again for your time and constructive feedback. We look forward to your response.
>
> Best regards,
> Authors

---

### Author Response · Authors · 2025-11-18
**General Responses and Revision Summary**

Dear All Reviewers:

We sincerely thank all the reviewers for their constructive feedback and valuable feedback. We are especially grateful for the recognition of our novel sim-to-real framework (*Reviewers Kiqx, 5oZW*), the idea and the effectiveness of the joint-wise neural dynamics model (*Reviewer kQoB*), the valuable real-world data collection strategy (*Reviewer kQoB*) and its scalability (*Reviewer Kiqx*), valuable theoretical insights that support the core ideas (*Reviewer Kiqx*) and the empirical findings (*Reviewer 5oZW*), extensive hardware evaluations (*Reviewer Kiqx, 5oZW*), empirical effectiveness with strong results (*Reviewer Kiqx, kQoB, 5oZW*), and the teleoperation demonstration (*Reviewer kQoB*).

We appreciate your thoughtful questions and concerns, and we have tried our best to address them comprehensively in our individual responses.

We have carefully revised the paper to reflect our responses to reviewers' concerns and suggestions (highlighted in pink purple). Key revisions are:

- Improve the method overview paragraph to better highlight our core contributions and to establish a clearer hierarchy among the different components.
- Change the terminology "randomized loads" to *"stochastic contact conditions"* or "*stochastic contact*" to make it more precise to reflect the actual setup.
- Clearly explain "works well on hardware" by explicitly describing the probing experiments and the observed performance differences between DAgger and BC.
- Explain the intuition of residual policy more clearly in Sec. 3.3 and add a more detailed discussion in Sec. B.4.
- Add more details of the teleoperation experimental settings to Sec. 4.2.
- Add more discussions on limitations in the "Conclusions and Limitations" section.
- Add additional experiments, analysis, and discussions (Sec. A.5, B.2, B.3, B.4, B.5)
- Add DexterityGen to our references, discuss it in the related work, and explicitly mention it when describing our teleoperation system.


**[update] We recognize that the previous response could be clearer and more direct. To support further discussion, we’ve provided a more concise version addressing your concerns.**

We are truly grateful once again for your feedback. Please refer to our individual responses for our answers to specific questions and concerns. Please let us know if you have any further questions or comments. We would be delighted to discuss them and will spare no effort in addressing them.

Thank you,
Authors

---

### Meta-Review · Area_Chair_qEG5 · 2026-01-09

**Summary:**

Across reviews, the main substantive concerns were about generality (hardware- and task-level transfer), clarity of what components are necessary (base policy vs. residual + NDM), and missing analyses (e.g., performance vs. object weight; data coverage and robustness in the “Chaos Box” collection). These concerns were partially addressed in the revision via clearer exposition/terminology, additional analyses (including weight–performance breakdown), and added discussion/limitations; however, multi-hardware real-world validation and multi-task generalization remain limited, with only preliminary indications beyond the primary LEAP-hand in-hand rotation setting.

One reviewer (zo2q) raised broader conceptual objections concerning the soundness of the proposed proprioception-only joint-wise dynamics modeling and the lack of explicit object-dependent dynamics. While these are theoretically interesting points, they were not shared by the other reviewers, and the rebuttal and revision provided reasonable empirical evidence that the proposed factorization performs effectively for the targeted in-hand rotation setting. Moreover, this reviewer later reduced their score to 0, citing frustration with rebuttal length and process issues rather than technical grounds. Consequently, the evaluation from this review was treated as an outlier and not heavily weighted in the overall assessment.

**Reviewer Concerns:**

Addressed concerns:

- Clarity / narrative hierarchy (many moving parts): The authors report revising the overview to better separate contributions and clarify component roles.  Terminology around “randomized loads”: Changed to more precise wording (e.g., “stochastic contact conditions”).

- “Works well on hardware” ambiguity & residual-policy intuition: Authors state they clarified what works on hardware and expanded intuition/discussion for the residual policy.

- Performance vs. object weight analysis: Added a weight–performance table/analysis (showing degradation for heavier objects and attributing failures to unstable grasps).

- Transfer/initialization evidence (within-hand and cross-embodiment in sim): Added results on cross-joint transfer and Leap to Allegro hand transfer (in simulation), showing faster fine-tuning and lower initial loss when initializing from other joints/hands.


Outstanding concerns:

- Limited real-world multi-hardware validation: Reviewers explicitly requested additional hands (e.g., Allegro/XHand). Authors acknowledge this and cite resource constraints; they mention preliminary Allegro results but do not present an extensive multi-hand real-world evaluation.

- Task generalization beyond in-hand rotation: Reviewers noted uncertainty about transfer to other manipulation tasks. Authors cite only preliminary indications beyond rotation and position, broader multi-task validation as future work

- Sensing modality limitations (proprioception-centric): One reviewer notes the approach does not yet leverage tactile/vision feedback, which could limit performance in ambiguous contacts; this remains largely unaddressed.

- Compute/runtime scaling & practicality details: A reviewer raised concerns that training multiple per-joint models + residual policy could be resource-intensive and that runtime scaling was not discussed; the revision summary does not clearly indicate a full accounting.

**Reviewer Scores:**

- Reviewer kQoB: No change (8). Review already strongly positive and the revisions align with the raised issues.

- Reviewer 5oZW: No change (8).  Key requested analyses/limitations were added (e.g., weight–performance; explicit limitations).

- Reviewer Kiqx: Increase (6 → 8). Thre reviewer explicitly stated willingness to increase the score if concerns were explained/addressed; the authors report revisions directly targeting those clarity/definition issues.

- Reviewer zo2q (changed from 2 → 0 during discussion): The reviewer explicitly stated lowering to 0 due to rebuttal length and maintained strong conceptual objections.

---

### Decision · Program_Chairs · 2026-01-26

Accept (Poster)